# Secondary vectors of Zika Virus, a systematic review of laboratory vector competence studies

**Marina Bisia[1], Carlos Alberto Montenegro-Quinoñez[2,3], Peter Dambach[2], Andreas Deckert[2], Olaf Horstick[2], Antonios Kolimenakis[1], Valérie R. Louis[2], Pablo Manrique-Saide[4], Antonios Michaelakis[1], Silvia Runge-Ranzinger[2], Amy C. Morrison [5]***

**1** Laboratory of Insects and Parasites of Medical Importance, Scientific Directorate of Entomology and Agricultural Zoology, Benaki Phytopathological Institute, Athens, Greece, **2** Heidelberg Institute of Global Health (HIGH), Faculty of Medicine and University Hospital, Heidelberg University, Heidelberg, Germany, **3** Instituto de Investigaciones, Centro Universitario de Zacapa, Universidad de San Carlos de Guatemala, Zacapa, Guatemala, **4** Unidad Colaborativa para Bioensayos Entomológicos (UCBE), Universidad Autónoma de Yucatán, Mérida, México, **5** Department of Pathology, Microbiology, and Immunology, School of Veterinary Medicine, University of California Davis, Davis, California, United States of America

* amy.aegypti@gmail.com

**Data Availability Statement:** All relevant data are within the paper and its Supporting information files.

## Abstract

### Background

After the unprecedented Zika virus (ZIKV) outbreak in the western hemisphere from 2015–2018, *Aedes aegypti* and *Ae. albopictus* are now well established primary and secondary ZIKV vectors, respectively. Consensus about identification and importance of other secondary ZIKV vectors remain. This systematic review aims to provide a list of vector species capable of transmitting ZIKV by reviewing evidence from laboratory vector competence (VC) studies and to identify key knowledge gaps and issues within the ZIKV VC literature.

### Methods

A search was performed until 15th March 2022 on the Cochrane Library, Lilacs, PubMed, Web of Science, WHOLIS and Google Scholar. The search strings included three general categories: 1) "ZIKA"; 2) "vector"; 3) "competence", "transmission", "isolation", or "feeding behavior" and their combinations. Inclusion and exclusion criteria has been predefined and quality of included articles was assessed by STROBE and STROME-ID criteria.

### Findings

From 8,986 articles retrieved, 2,349 non-duplicates were screened by title and abstracts,103 evaluated using the full text, and 45 included in this analysis. Main findings are 1) secondary vectors of interest include *Ae. japonicus*, *Ae. detritus*, and *Ae. vexans* at higher temperature 2) *Culex quinquefasciatus* was not found to be a competent vector of ZIKV, 3) considerable heterogeneity in VC, depending on the local mosquito strain and virus used in testing was observed. Critical issues or gaps identified included 1) inconsistent

**Funding:** This study was a byproduct of original funding by the Special Programme for Research and Training in Tropical Diseases of the World Health Organization (WHO/TDR, contract number WCCPRD5295811 2017/700816 to OH, SRR, ACM, and PMS). MB, AK, and AM received salary support from the LIFE CONOPS (LIFE12 ENV/GR/000466 to AK) project co-funded by the European Commission and the participating Organizations in the framework of a LIFE + Environment Policy and Governance project (www.conops.gr). ACM received salary support from the National Institute of Allergy and Infectious Diseases of the National Institutes of Health (U01AI151814 to ACM). The funders had no role in study design, data collection and analysis, decision to publish, or preparation of the manuscript.

**Competing interests:** The authors have declared that no competing interests exist.

definitions of VC parameters across the literature; 2) equivalency of using different mosquito body parts to evaluate VC parameters for infection (mosquito bodies versus midguts), dissemination (heads, legs or wings versus salivary glands), and transmission (detection or virus amplification in saliva, FTA cards, transmission to neonatal mice); 3) articles that fail to use infectious virus assays to confirm the presence of live virus; 4) need for more studies using murine models with immunocompromised mice to infect mosquitoes.

## Conclusion

Recent, large collaborative multi-country projects to conduct large scale evaluations of specific mosquito species represent the most appropriate approach to establish VC of mosquito species.

## Author summary

The mosquitoes *Aedes aegypti* and *Ae. albopictus* are known to transmit Zika virus (ZIKV) but it is important to identify other potential secondary vectors. We conducted a systematic review of the literature to answer this question. We searched four databases (PubMed, Lilacs, Cochrane Library Web of Science), WHOLIS and Google Scholar using different combinations of Zika, *Aedes*, *Culex*, vector, or competence in the title/abstract, up to March 2022. Most of the studies reviewed were of high quality methodologically, but the methods were different making it hard to compare them. There is a need for standardization to better interpret these studies and make appropriate recommendations. Secondary vectors of ZIKV with evidence of low transmission rates comparable to primary vectors are *Ae. japonicus*, *Ae. detritus* and *Ae. vexans* at higher temperatures. *Culex quinquefasciatus* was not found to be a competent vector of ZIKV. Future research should focus on well-defined and established experimental approaches (midguts/bodies for infection, legs+wings/heads for dissemination and the use of murine models/other artificial feeding systems). Importantly, development of large collaborative multi-country projects are needed to conduct large scale evaluations of specific mosquito species with common protocols to appropriately address the inherent geographic variation in both mosquito and ZIKV strains.

## Introduction

The emergence of Zika virus (ZIKV) transmission and associated neurological and congenital consequences in the western hemisphere in 2015 resulted in a World Health Organization declaration of a "Public Health Emergency of International Concern" for most of 2016 [1,2]. This emergency exposed significant knowledge gaps, not only about vector competence (VC) for "known" vectors *Aedes aegypti* and *Ae. albopictus*, but also on potential "unknown" secondary vectors. Following the Zika outbreak several review studies employing different search methodologies reported ZIKV isolations and laboratory-based VC studies for known and suspected mosquito vectors. In a scoping review Wadell and Grieg [3] included studies on 45 mosquito species of which 18 were positive for ZIKV from 1956 to 2015 in Africa and Asia. Epelboin et al. [4] included studies of 53 mosquito species across eight genera in a systematic review focused on vectors that included work published through August 2017. While these two

studies were published about 18 months apart, the number of species tested for VC increased from 8 to 22. In a 2018 expert opinion review, Boyer et al. [5] summarized laboratory studies and cataloged the mosquito species from which Zika virus was isolated.

Vector competence is defined as the ability of a mosquito to become infected, allow virus amplification, and subsequently transmit a pathogen to another vertebrate host [6,7]. It represents all the intrinsic factors (genetic, physical, physiological, and immunological) underlying virus propagation in the mosquito; a virus' journey through the mosquito that includes successful replication of the virus in the mosquito's midgut epithelium, navigation across the midgut wall, dissemination to the salivary gland cells, and secretion into saliva. The success of a vector to transmit a pathogen also includes additional factors like longevity and host preferences, as well as extrinsic factors associated with behavioral and ecological characteristics of the species which increase exposure to mosquito bites; which together interplay and are defined as vectorial capacity [6]. Thus, vector competence is a necessary but not sufficient requirement to characterize definitively a vector species as epidemiologically relevant and that contributes significantly to the natural maintenance and transmission of a specific virus. For example, *Ae. aegypti* has many characteristics that enhance its vectorial capacity: larval development sites and adult resting sites strongly associated with human habitats in urban areas, highly anthropophilic biting behavior, females taking multiple blood meals during a single gonotrophic cycle, and diurnal feeding behavior [8–10].

Vector competence experiments generally include the following steps: 1) exposure to virus via an artificial blood feeder, or feeding on an infected live host (for Zika an immunocompromised mouse model) (number of engorged mosquitoes that were tested [#tested]), 2) testing of whole mosquito bodies, midguts, or carcasses to measure infection (the number of mosquitoes testing positive for virus or viral RNA are the #infected [#inf.]) 3) testing heads, legs+wings, salivary glands/ovaries to measure dissemination (the number samples testing positive for virus or viral RNA are mosquitoes with viral dissemination [#dissem.]), and 4) testing saliva or exposure of live animals to virus infected mosquitoes to measure transmission (number of samples testing positive for virus or viral RNA are mosquitoes that were able to transmit [#transm.]) [7]. Researchers present rates using two approaches where different denominators are employed: stepwise and cumulative [7]. Infection rate is the same for both approaches, #inf./#tested. Cumulative rates, the most informative for evaluating VC are #dissem./#tested and #transm./#tested for dissemination and transmission, respectively. Stepwise rates reveal where potential genetic barriers exist use smaller denominators for dissemination (#dissem./#inf) and transmission (#transm/#dissem). Terminology for these rates varies among authors and overtime but for the purposes of this review we use these recently proposed terms designed to establish minimum reporting standards for VC experiments [7]. The cumulative transmission rate alone implicates a species as a vector.

The challenges associated with assessment of VC studies for ZIKV are similar to those identified for dengue virus VC studies e.g., primarily the observation of significant variation in competence among geographically isolated vector populations [11,12]. The combination of virus and mosquito strains tested, assays used to detect virus (RNA compared to infectious virus), blood feeding and experimental methods, and parameters assessed (both stepwise and cumulative infection, dissemination, transmission rates) are all critical for assessment of VC [4,7,13].

Interest in ZIKV vectors has been characterized by an urgent response to outbreaks, first in the South Pacific [14–17] and then in the western hemisphere as described above. Prior to that, a forest cycle between ZIKV-non-human primates- canopy mosquitoes had been recognized since the first ZIKV was isolated in Uganda from a rhesus monkey and *Ae. africanus* in the 1940s [18]. Thus, much of our knowledge pre-dating the South Pacific epidemics

originated from forest-based studies directed primarily at yellow fever (YF) [19–29]. During the peak of the ZIKV outbreak in the western hemisphere from 2015–2018, there was an increase of VC studies (particularly short communications) testing an array of mosquito species often from insectary-readily available colonies. During this period, some publications [30–32] implicated *Cx. quinquefasciatus*, a cosmopolitan species often associated with wastewater and abundant in urban settings, as a vector of ZIKV with potentially dramatic epidemiological consequences. Also important was the assessment of a variety of *Aedes* species, including the invasive *Ae. japonicus* and others with more restricted geographic distributions. After 2017, ZIKV transmission has decreased world-wide, and a period where VC studies with more complex and through study designs emerged.

After over 5 years since the peak of the Zika virus outbreak, there is an opportunity to evaluate and update all the evidence for possible secondary ZIKV vectors. There is now ampler and well-established evidence that *Ae. aegypti* and *Ae. albopictus* are vectors for ZIKV to humans [4,5,13], but still, no clear consensus about other possible vectors. We have assessed the latter in this systematic review and focused on laboratory VC studies, because they are a requirement to evaluate if a potential vector species can transmit an arbovirus, with the overall aim to identify key gaps/issues in the current literature to provide a concrete list of vector species capable of transmitting ZIKV.

## Methods

### Search strategy, databases, and search terms

Our research question formulated was "Is there evidence of vector competence for ZIKV for mosquito species in addition to *Ae. aegypti* and *Ae. albopictus*?". Our review followed the Preferred Reporting Items for Systematic Reviews and Meta-Analyses (PRISMA) statement [33]. Articles were identified by searching electronic databases until the 15[th] of March 2022. The articles were extracted from: four databases, Cochrane Library (https://www.cochranelibrary.com/), Latin American and Caribbean of Health Sciences Information System- LILACS (https://lilacs.bvsalud.org/en/), PubMed (https://pubmed.ncbi.nlm.nih.gov/) and Web of Science (https://www.webofknowledge.com); one register, World Health Organization (WHO) library catalogue (WHOLIS, https://asksource.info/resources/wholis); and Google Scholar (https://scholar.google.com/). The search strings included three general categories: 1) "ZIKV"; 2) "vector"; 3) "competence" and their combinations using free text terms and medical subject headings (MeSH) terms when applicable (e.g., PubMed) for: Zika, *Aedes*, *Culex*, vector, or competence in the title/abstract. After piloting our initial search string and determined that some relevant articles based on previous reviews were not retrieved, we added the following search terms: transmission, isolation, and feeding behavior. Our principal search strings using MeSH terms are as shown in Box 1 and were adjusted to each search engine.

We used Zotero (https://www.zotero.org/) to identify duplicate articles, and all the extracted articles were processed using the Rayyan platform (www.rayan.com) where two members of the research team (MB, CAMQ) screened each article title/abstract to identify ZIKV VC studies that represented primary research. The search was performed in English. All articles were examined by both researchers before inclusion and a third researcher and subject expert (ACM) was asked to assist with the final decision for articles without consensus. Exclusion criteria (Fig 1) were:

- Systematic or literature reviews

- Opinion papers

Box 1. Search string utilized to extract articles in Cochrane Library, Latin American and Caribbean of Health Sciences Information System (LILACS), PubMed, Web of Science, World Health Organization library catalogue (WHOLIS) and Google scholar. Medical subject headings (MeSH) terms were used when applicable (e.g., PubMed).

Zika*[Title/Abstract] AND [[Aedes*[Title/Abstract] OR [Aedes*[MeSH Terms]] OR [[Culex*[Title/Abstract] OR [Culex*[MeSH Terms]] OR Vector*[Title/Abstract] AND Competence*[Title/Abstract] OR Competence*[MeSH Terms]

Zika*[Title/Abstract] AND [[[Aedes*[Title/Abstract] OR [Aedes*[MeSH Terms]] OR [[Culex*[Title/Abstract] OR [Culex*[MeSH Terms]] OR Vector*[Title/Abstract] AND Transmission*[Title/Abstract]

Zika*[Title/Abstract] AND [[Aedes*[Title/Abstract] OR [Aedes*[MeSH Terms]] OR [[Culex*[Title/Abstract] OR [Culex*[MeSH Terms]] OR Vector*[Title/Abstract] AND Isolation*[Title/Abstract]

Zika*[Title/Abstract] AND [[Aedes*[Title/Abstract] OR [Aedes*[MeSH Terms]] OR [[Culex*[Title/Abstract] OR [Culex*[MeSH Terms]] OR Vector*[Title/Abstract] AND [[Feeding behavior*[Title/Abstract] OR [behavior, feeding*[MeSH Terms]]

Zika*[Title/Abstract] AND [[Aedes*[Title/Abstract] OR [Aedes*[MeSH Terms]] OR [[Culex*[Title/Abstract] OR [Culex*[MeSH Terms]] OR Vector*[Title/Abstract] AND Competence*[Title/Abstract] OR Competence*[MeSH Terms] OR Transmission*[Title/Abstract] OR Isolation*[Title/Abstract] OR [Feeding Behavior*[Title/Abstract] OR [behavior, feeding*[MeSH Terms]]

- Studies focused on mathematical modelling

- Vaccine development

- Case reports

- Field-based vector incrimination studies (isolation of virus from field-collected mosquitoes)

- Papers regarding workshops

- Meeting results

  Inclusion criteria (Fig 1) were:

- Laboratory evaluation of vector competence

- Included vector species other than *Ae. aegypti* and *Ae. albopictus*

- Published in a peer reviewed journal

Data were extracted (MB) and entered into extraction forms, including author, title, journal, publication date, and study design. Additional sections broadly followed content analysis methods, using categories as these emerged during analysis of the results [34], such as

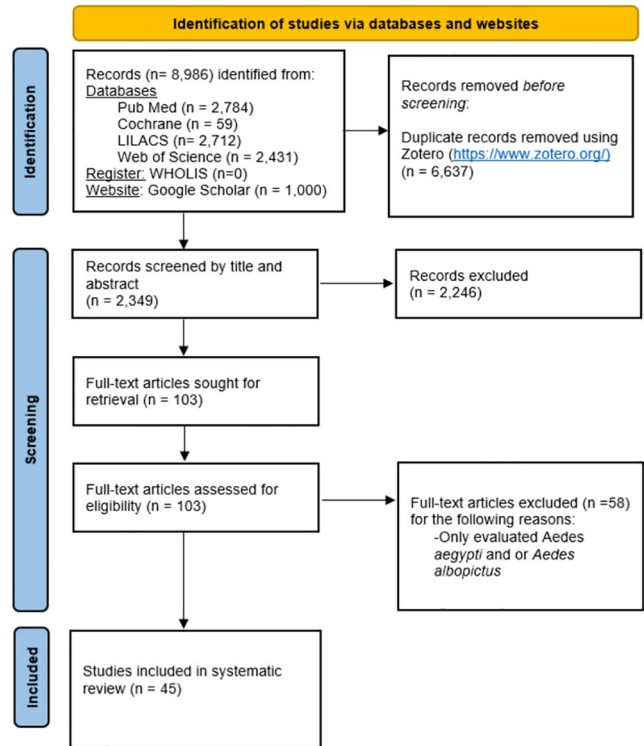

**Fig 1. PRISMA flow diagram (adapted from Page et al. [33]) describing identification of articles examining vector competence of mosquito species other than *Aedes aegypti* and *Aedes albopictus* for Zika virus transmission.** The electronic search started on 10 November 2021 and article selection was finalized on 15 March 2022.

mosquito species and/or strains studied, type of mosquito used (colony or field), virus strains and doses administered, infection method used (blood feeding device, direct feeding on mouse), assays used to detect virus or RNA, strategy used to measure mosquito infection (positive bodies or midguts), dissemination (positive heads, legs, wings, salivary glands, ovaries, etc.), and most importantly, ability to transmit virus (saliva testing or transmission to mouse), as well as the number of days post-exposure that mosquitoes were processed or analyzed, results, limitation mentioned by authors, and conclusions. The data extraction forms were reviewed and checked for accuracy by ACM and CAMQ.

## Quality assessment

We developed a grading tool to assess the quality of our evaluated articles, using a checklist developed from Reporting of Observational Studies in Epidemiology (STROBE) [35] and Strengthening of the Reporting of Molecular Epidemiology for Infectious Diseases (STROME-ID) [36] criteria (S1 Table). The STROBE checklist was enriched by the authors to be in accordance with the procedures followed in the papers attempting to include score categories for all relevant STROBE and STROME-ID criteria (S1 Table). The checklist was finally composed of 33 items with a possible maximum of 38 points. The evaluation, however, was heavily weighted on 20 items evaluating study methodology and a "methodology score" was additionally computed that comprised a maximum of 24 possible points. The items included a clear description of the virus and mosquito strains used in the experiments, the type of assay used to

detect Zika virus or RNA in the mosquitos, if infection, dissemination, and transmission were appropriately measured, and the number of mosquitoes used to evaluate these parameters and if replicate experiments were conducted. Some items that comprised this portion of score had higher maximum points. The remaining items, not part of the methodology score emphasized proper reporting and included the term "vector competence" in the title, clear and accurate abstract and objectives, presentation of key study results in relation to study objectives, appropriate description and use of statistical analysis, transparent discussion of study limitations, and appropriate interpretation and generalizability of the study results. Two researchers (MB and ACM) scored each of the 45 included articles independently. If the scores did not match, all 38 scores were compared and where differences were observed discussed between the two researchers. For most scores that were not concordant, the article was reviewed to confirm that each researcher had correctly extracted the information necessary to provide a score (e.g., type of laboratory assay, mosquito colony history, virus passage history, number of virus or mosquito strains used). Scores were adjusted based on careful review of the manuscript and the score justification provided by each researcher. For some scores that were more subjective (e.g., clearly written abstract, adequate discussion of study limitation) the researchers discussed their scores, compared them to other articles to standardize their approach to each question. Each score was discussed until both researchers agreed on a single score. No studies were excluded after the quality assessment, but the analysis and the report of the results consider this quality assessment.

## Results

### Descriptive results

We retrieved 8,986 articles from four databases, one register, and one searchable website. We identified 6,637 as duplicates that were removed prior to screening (Fig 1). We screened the title and abstract of the remaining 2,349 articles and from those 2,246 were excluded based on the inclusion/exclusion criteria: of the 103 articles retrieved, 58 only evaluated *Ae. aegypti* or *Ae. albopictus*, finally leaving 45 papers that were included in our analysis of secondary vector species for ZIKV.

The 45 included peer-reviewed articles, summarized in Table 1, came from 20 journals, most with a focus on emerging infections (Emerging Infectious Diseases [3], Emerging Microbes and Infections [5], Eurosurveillance [3]), infectious and tropical diseases (American Journal of Tropical Medicine and Hygiene [2], BMC Infectious Diseases [1], Mem. Insti. Oswaldo Cruz [1], PLOS Neglected Tropical Diseases [9], Parasites and Vectors [4], Vector-borne and Zoonotic Diseases [1]), virology or microbiology (Frontiers in Microbiology [1], Journal of General Virology [1], Mbio [1], MDPI-pathogens [2], Virology Journal [1], Viruses [2]), entomology (Journal of the American Mosquito Association [1], Journal of Medical Entomology [3]), and three general journals (Nature [2], PeerJ [1], Proceedings B [1]).

### Quality grading tool assessment

Our quality grading tool scores (QAS) showed a positively skewed distribution ranging from 20 to 35 out of a possible 38 maximum points. The mean (± SD) score was 29.0 ± 4.2, with the lowest quartile representing scores from 20–25 and the top quartile ranging from 33–35 points out of a maximum of 38 points. Methodology scores (MS), a component representing 63% of the total score (TS) showed a similar distribution with a mean (± SD) score of 17.2 ± 3.2, with the bottom quartile scores ranging from 11–14 and top quartile from 21–22, out of a possible 24 methodology points. It is notable that the article scoring lowest was also the only article published before 2015 which does not include an infectious virus assay or assess transmission.

**Table 1. Evidence table, in chronological order from 2014 to 2021.** ZIKV = Zika virus, USUV = Usutu virus, VC = vector competence, VD = Virus/RNA detection in field collected mosquitoes, IT = Intrathoracic inoculations (examining upstream barriers to transmission), dpi = days post infection (includes manuscripts reporting dpe = days post exposure), RT = Room temperature. Infection Rate (IR, [#inf./#tested]) is defined as the percentage of mosquitoes containing virus in bodies or midguts (number positive/number tested). Terminology used for dissemination and transmission are expressed as cumulative (C) or stepwise (S) rates depending on the experimental design. For dissemination, we use cumulative dissemination rate (CDR, [#dissem./#tested]) or stepwise dissemination rate (SDR, [#dissem./#inf]), defined as the percentage of mosquitoes containing virus in head, legs+wings, or salivary glands/ovaries (number positive/number of engorged mosquitoes tested for infection [CDR] or number positive /number of infected mosquitoes [SDR]). For transmission, we use cumulative transmission rate (CTR, [#transm./#tested]) or stepwise transmission rate (STR, [#transm./#dissem.]), defined as the number of mosquitoes with virus in saliva or transmitting ZIKV to a mouse (number positive/number of engorged mosquitoes tested [CTR] or number positive/number disseminated infections [STR]). For authors that emphasized salivary gland testing, CSGR = cumulative salivary gland positivity rate is used. For the quality assessment TS = Total Score, MS = Methodology Score, QAS = percentile in quantile analysis among all articles scored.

| | Reference (Article Type) | Study Objectives | Mosquitoes/Virus/Temperature | Results | Quality Assessment (QAS) |
|---|---|---|---|---|---|
| [37] | **Lederman JP et al. PLoS NTD 2014** *doi:10.1371/journal.pntd.0003188* Aedes hensilli as a potential vector of Chikungunya and Zika viruses (*Research Article*) | VC: #inf./#tested (IR) #dissem/#inf. (SDR) VD Outbreak Investigation | Ae. hensilli (F12-15) MR766 4.9 log10 PFU/ml 5.7 log10 PFU/ml 5.9 log10 PFU/ml 28°C | Results of a late outbreak investigation on YAP island. Carried out 1 week of diurnal and nocturnal collections including larval surveys and adult mosquitoes using light and gravid traps, as well as household aspirations. Ae. hensilli was the most abundant species, followed by Cx. quinquefasciatus, but no virus was isolated from field caught mosquitos. A colony of Ae. hensilli was established. F12-15 mosquitoes were orally infected with MR766 ZIKV strain at three doses. Infection (bodies) and dissemination (heads) were evaluated at 8 dpi. IR at lowest dose was 7% (1/14) compared to higher doses was 84% (47/56). SDR was observed only at the two higher doses (19%, 9/47). | QAS: < 25% (TS = 20, MS = 11) Strengths: Dose Response evaluated, locally derived colony. Weakness Low sample sizes, only evaluated 8 dpi, rather than the standard 14 dpi. |
| [38] | **Diagne CT et al. BMC Infect Dis 2015;15: 492.** *doi:10.1186/s12879-015-1231-2* Potential of selected Senegalese Aedes spp. mosquitoes (Diptera: Culicidae) to transmit Zika virus (*Research Article*) | VC: #inf./#tested (IR) #dissem/#inf. (SDR) #transm./#dissem. (STR) Forest cycle | Ae. unilineatus, Ae. vittatus, Ae. luteocephalus Virus: ArD 128000, ArD 132912, ArD 157995, ArD 165522 (mosquito origin); HD 78788, MR766 (human origin) 6–7 log10 PFU/ml 27±1°C, 80±5% RH | Orally infected F1, Ae. aegypti, Ae. unilineatus, Ae. vittatus, and Ae. luteocephalus mosquitos with 5 ZIKV strains (3 and 2 of mosquito and human origin, respectively) at doses ranging from 6–7 log10 PFU/ml. IR (bodies), SDR (heads), and STR (saliva) were evaluated at 5, 10, and 15 dpi using qPCR. VC parameters varied with virus strain used for each species evaluated. Across strains and doses Ae. unilineatus (IR = 56/300, SDR = 3/56, STR = 0/3) was unable to transmit ZIKV whereas Ae. vittatus, (IR = 37/56, SDR = 10/37, STR = 2/10) and Ae. luteocephalus (IR = 45/60, STR = 19/45, STR = 50%) were able to transmit virus strains isolated from monkey/human sera (MR766, HD78788). | QAS: > 50% (TS = 32, MS = 20) Strengths: used F1 mosquitoes and multiple ZIKV strains, range of dpi evaluated. Weakness: Used PCR only to test mosquitoes. Saliva testing for Ae. luteocephalus not clear. |
| [39] | **Aliota MT et al. Emerg Infect Dis 2016;22:1857–1859.** http://dx.doi.org/10.3201/eid2210.161082 Culex pipiens and Aedes triseriatus mosquito susceptibility to Zika virus (*Letter to Editor*) | VC: #inf./#tested (IR) #dissem./#tested (CDR) #transm./#tested (CTR) Murine Model | Cx. pipiens, Ae. triseriatus, Ae. aegypti, Ae. albopictus Asian lineage PRVABC59 4.7 log10 PFU/ml 6.0 log10 PFU/ml 6.8 log10 PFU/ml No rearing temperature provided. | Laboratory studies, using ZIKV Ifnar-/- mice, Asian linage PRVABC59, at 14 dpi (3 replicates), Cx. pipiens, Ae. triseriatus, Ae. aegypti, Ae. albopictus (all from colonies, F>>10) collected bodies, legs, saliva, to evaluate IR, CDR, and CTR. Cx. pipiens was refractory to infection. Ae. triseriatus (IR6.8log10 PFU/ml = 4/13) became infected at high titers but showed no dissemination or transmission. Ae. albopictus infection, dissemination, and transmission was low compared to Ae. aegypti and dose dependent. Ae. aegypti evaluated at highest dose had IR = 17/17, CDR = 12/17, and CTR = 4/17. | QAS: < 25% (TS = 24, MS = 13) Strengths: All samples screened by plaque assay (infectious assay). Infected via animal feeding. Range of doses. Weakness: Laboratory mosquito strains (>15 yrs), single virus strain; short format. Rearing temperature not reported. |

(*Continued*)

Table 1. (Continued)

| | Reference (Article Type) | Study Objectives | Mosquitoes/Virus/Temperature | Results | Quality Assessment (QAS) |
|---|---|---|---|---|---|
| [40] | Amraoui F. et al. Euro Surveill 2016;21. doi:10.2807/1560-7917.ES.2016.21.35.30333 Culex mosquitoes are experimentally unable to transmit Zika virus. (Rapid Communication) | VC: #inf./#tested (IR) #dissem/#tested (CDR) #transm./#tested (CTR) Temperate species | Cx. quinquefasciatus (San Joaquin Valley, CA, 1950) Cx. pipiens (Tunisia, 2010) NC201-5132, New Caledonia 2014 $7.2 \log_{10}$ PFU/ml 28°C | VC assays using laboratory strains of Cx. quinquefasciatus and Cx. pipiens orally infected with single ZIKV strain at a single dose, evaluated at 3, 7, 14, and 21 dpi. The number of virus particles ingested were titrated in recently engorged mosquitoes were $4.8 \log_{10}$ PFU/ml for Cx. pipiens, and $5.0 \log_{10}$ PFU/ml for Cx. quinquefasciatus. IR for Cx. pipiens was 2% (1/48) at 3 dpi, 6% (3/47) at 7 dpi, 0% (0/47) at 14 dpi, and 13% (6/46) at 21 dpi. In contrast, for Cx. quinquefasciatus IR was 0% (0/42) at 3 dpi, 2% (1/47) at 7 dpi, 17% (7/41) at 14 dpi, and 13% (5/40) at 21dpi. Only Cx. quinquefasciatus was able to disseminate the virus ($CDR_{14dpi} = 1/41$, $CDR_{21dpi} = 3/40$) and neither species was able to transmit ZIKV up to 21 dpi. | QAS: 25% (TS = 26, MS = 14) Strengths: All samples screened by plaque assay (infectious assay); range dpi evaluated; intrathoracic injections provided additional support for Culex not being a vector. Weakness: Single virus strain, use of old laboratory colonies. No Ae. aegypti control |
| [41] | Boccolini D et al. Euro Surveill 2016;21. doi:10.2807/1560-7917.ES.2016.21.35.30328 Experimental investigation of the susceptibility of Italian Culex pipiens mosquitoes to Zika virus infection (Rapid Communication). | VC: #inf./#tested (IR) #dissem/#tested (CDR) #transm./#tested (CTR) Temperate species | Ae. aegypti Cx. pipiens ZIKV H/PF/2013 (Origin human French Polynesia 2013) $6.46 \log_{10}$ PFU/ml 26±1°C; 70% RH 14 h:10 h L:D | Culex pipiens, collected from Rome, Italy in 2015 (generation not provided) and Ae aegypti, from a laboratory colony from Reynosa, Mexico (1998, F>>10) were orally infected with ZIKV at a single dose and evaluated at 0, 3, 7, 10, 20, 24 dpi. Aedes aegypti, IR: 0 dpi = 8/8, 7 dpi = 6/12, 14 dpi = 4/8, 20d = 4/10; CDR: 7 dpi = 6/12, 14 dpi = 4/8, 20 dpi = 3/10; CTR: 7 dpi = 2/12, 14 dpi = 3/8, 20 dpi = 3/10. Cx. pipiens were 100% refractory. | QAS: 25% (TS = 26, MS = 14) Strengths: Evaluated range of dpi, used local Culex strain. Weakness: Used PCR only to test mosquitoes, single ZIKV strain and dose, single Culex strain. |
| [42] | Fernandes RS et al. PLoS NTD 2016 doi:10.1371/journal.pntd.0004993 Culex quinquefasciatus from Rio de Janeiro is not competent to transmit the local Zika virus (Research article) | VC: #inf./#tested (IR) #dissem/#inf. (SDR) #dissem./#dissem. (STR) #transm./#tested (CTR) Site with active ZIKV transmission | Ae. aegypti, Cx. quinquefasciatus Rio-U1 (KU926309), Rio-S1 (KU92630) (Local) 6 $\log_{10}$ PFU/ml 26°C, 70±10% RH 12 h:12 h L:D | Orally infected 4 strains (F1-F3) of Cx. quinquefasciatus from 4 districts, Rio de Janiero, Manguinhos (MAN), Triagem (TRI-colony), Copacabana (COP), Jacarepagua (JAC); Ae. aegypti Urca (URC), Paqueta (PAQ-F2), with 2 local Brazilian strains of ZIKV. Infection (bodies), dissemination (heads), and transmission (saliva) were evaluated at 7, 14, 21 dpi using plaque assay and qPCR. Aedes aegypti infection rates varied with mosquito strain. IR: 7 dpi = 80%, 14–21 dpi = 90–100%; SDR: 7 dpi = 40%, 14–21 dpi = 85–100%; $STR_{14dpi}$ = 72–97%, $CTR_{14dpi}$ = 61–93%. For all Culex strains, IRs were negligible to null, and no dissemination was observed. | QAS: ≥ 99% (TS = 35, MS = 22) Strengths: Used multiple mosquito and virus strains, infectious assay, and evaluated a good dpi range. Included Ae. aegypti control. Mosquitoes were recently collected from field. Weakness: none noted |

(Continued)

**Table 1.** (Continued)

| | Reference (Article Type) | Study Objectives | Mosquitoes/Virus/Temperature | Results | Quality Assessment (QAS) |
|---|---|---|---|---|---|
| [30] | **Guo XX et al. Emerg Microbes Infect 2016;5:e102.** *doi:* 10.1038/emi.2016.102 *Culex pipiens quinquefasciatus*: a potential vector to transmit Zika virus (*Research article*) | VC: #inf./#tested (IR) #SG+/#tested (CSGR) #transm./#dissem. (STR) #transm./#tested (CTR) Transmission to neonatal mice in addition to saliva testing | *Cx. quinquefasciatus* SZ01 (human traveler from Samoa 2016) 5.5 log$_{10}$ PFU/ml 26±1° C, 75±5% RH 14 h:10 h L:D | Orally infected *Cx. quinquefasciatus* laboratory strain (Hainan province 2014, generation unknown) with single low passage ZIKV strain at a single dose. Infection (midguts), dissemination (salivary glands/ ovaries), and transmission (ZIKV RNA + saliva over number of mosquitoes with disseminated infection), and CTR (ZIKV RNA + saliva over number of blood fed mosquitoes tested). ZIKV exposed mosquitoes allowed to feed on mice. Evaluations at 2,4,5,8,12,16,18 dpi using PCR. IR, 80% at 2 dpi, then oscillates between 10–40%. Salivary gland infection (CSGR) first observed at 2 dpi, peaks at 8 dpi, then decreases. Detection of ZIKV RNA in saliva peaks at 8 dpi, then decreases on 12 and 18 dpi. CTR, 6 dpi = 0%, 8 dpi = 80%, 12 dpi = 10%, 16 dpi = 0%. One day 10 post-exposure to ZIKV infected mosquitos, 8 of 9 infant mice had viral RNA in their brain. | QAS: < 25% (TS = 22, MS = 12) Strength: Transmission to mice is convincing but appears transient. Weakness: No infectious assay, colony material, single virus and mosquito strains. Poor discussion of possible explanation of results. No *Ae. aegypti* control. **Results are inconsistent with biology, looks like transient infections no persistent infection which would be a new paradigm.** |
| [43] | **Hall-Mendelin S. et al. PLoS NTD 2016; 10:e0004959** *doi:* 10.1371/journal.pntd. 0004959 Assessment of local mosquito species incriminates *Aedes aegypti* as the potential vector of Zika virus in Australia (*Research Article*) | VC: #inf./#tested (IR) #dissem./#inf. (SDR) #dissem/#tested (CDR) #transm./#dissem. (STR) #transm./#tested (CTR) Local species Australia | *Ae. aegypti, Ae. vigilax, Ae. procax Ae. notoscriptus, Cx. quinquefasciatus, Cx. annulirostris, Cx. sitiens* MR766 Dose: 6.5–6.9 log$_{10}$ TCID$_{50}$/ml 26° C, 12 h:12 h L:D | Vector competence studies on potential Australian ZIKV vectors: *Ae. vigilax, Ae. procax, Cx. annulirostris, Cx. sitiens* (F1), *Ae. aegypti* F4, Townsville; *Ae. notoscriptus, Cx. quinquefasciatus* F1 (All from Queensland, Australia). Carried oral infections with MR766 prototype virus at a single dose. *Ae. notoscriptus*, IR$_{7dpi}$ = 18/25, IR$_{14dpi}$ = 34/60, SDR$_{7dpi}$ = 2/18, SDR$_{14dpi}$ = 6/16, CDR$_{7dpi}$ = 2/ 25, CDR$_{14dpi}$ = 6/60. *Ae. procax*, IR$_{14dpi}$ = 2/6, SDR$_{14dpi}$ = 1/6, CDR$_{14dpi}$ = 1/2. *Ae. vigilax*, IR$_{14dpi}$ = 17/30, SDR$_{14dpi}$ = 8/30, SDR$_{14dpi}$ = 8/ 17. *Cx. quinquefasciatus*, IR$_{14dpi}$ = 2/30, SDR$_{14dpi}$ = 0/30. *Cx. annulirostris* and *Cx. sitiens* were completely refractory. No evidence of ZIKV transmission was observed in any species other than *Ae. aegypti*, the species with infected saliva. *Ae. aegypti* STR$_{10dpi}$ = 3/25, STR$_{14dpi}$ = 8/30, CTR$_{10dpi}$ = 3/7, STR$_{14dpi}$ = 8/12. | QAS: ≥ 75% (TS = 33, MS = 20) Strengths: Used F1 mosquitoes collected from the field, established infecting dose. Weakness: Used a single ZIKV strain and tested mosquitoes by PCR rather than infectious assay. |
| [44] | **Huang YJ et al. Vector Borne Zoonotic Dis 2016.** *doi:* 10.1089/vbz.2016.2058 *Culex* species mosquitoes and Zika virus (*Research Article*) | VC: #inf./#tested (IR) #dissem/#tested (CDR) | *Cx. pipiens, Cx. quinquefasciatus* (Colonized < 2yrs) PRVABC59 7.2 log$_{10}$ TCID$_{50}$/ml 28° C, 16 h:8 h L:D | Orally infected 2 laboratory strains of *Cx. pipiens* from California (F15) and New Jersey (F7) and one strain of *Cx. quinquefasciatus* (Vero Beach, F7) with ZIKV strain PRVABC59 (low passage) at a single dose. Evaluated infection (bodies) and dissemination (heads) at 7 and 14 dpi. Screened by TCID$_{50}$ followed by confirmation by PCR. All strains were refractory to infection. | QAS: < 25% (TS = 24, MS = 15) Strengths: Used low passage virus strain. Used infectious assay. Weakness: No *Ae. aegypti* control, mosquito colonies, single virus strain. |

(*Continued*)

Table 1. (Continued)

| | Reference (Article Type) | Study Objectives | Mosquitoes/Virus/Temperature | Results | Quality Assessment (QAS) |
|---|---|---|---|---|---|
| [45] | Richard V et al. PLoS NTD 2016;110: e0005024 doi:10.1371/journal.pntd.0005024 Vector competence of French Polynesian Aedes aegypti and Aedes polynesiensis for Zika Virus (Research Article) | VC: #inf./#tested (IR) #dissem/#tested (CDR) #transm./#tested (CTR) VD Evaluated EIP/Kinetics | Ae. aegypti Ae. polynesiensis PFI/251013, 3 passages 7 $\log_{10}$ TCID$_{50}$/ml 27°C, 75% RH 12 h:12 h L:D | Aedes aegypti and Ae. polynesiensis were suspected to be ZIKV vectors in addition of Ae. aegypti. Laboratory colonies (F16 to F18) were orally infected with a French Polynesian ZIKV strain at a single dose and evaluated at 2, 6, 9, 14, 21 d dpi. Infection (bodies), dissemination (legs), and transmission (saliva) evaluated by TCID$_{50}$ in C6/36 cells. Ae. aegypti, IR = 85–93% starting 6 dpi, CDR = 18% at 6 dpi, 75% 9 dpi, 85% 14 dpi and 93% at 21 dpi, CTR = 36% at 14 dpi, 73% at 21 d. Ae. polynesiensis, IR$_{6dpi}$ = 11% (10/95), IR$_{9dpi}$ = 20% (18/89), IR$_{9dpi}$ = 36% (24/66); CDR$_{9dpi}$ = 3% (3/89), CDR$_{14dpi}$ = 18% (12/66), CTR = 0%. Overall, Ae. aegypti was more susceptible to ZIKV and has more favorable kinetics than Ae. polynesiensis | QAS: > 50% (TS = 31, MS = 19) Strengths: Tested two possible vector strains from French Polynesian ZIKV outbreak, evaluated EIP/Kinetics. Weakness: Used a single virus strain |
| [46] | Weger-Lucarelli J et al. PLoS NTD 2016; 10:e0005101 doi:10.1371/journal.pntd.0005101 Vector competence of American mosquitoes for three strains of Zika virus (Research Article) | VC: #inf./#tested (IR) #dissem/#inf. (SDR) #transm./#dissem. (STR) #transm./#tested (CTR) Evaluation of freeze thaw cycles. In-Vitro replication, competitive fitness | Ae. aegypti, Cx. tarsalis, Cx. pipiens, Cx. quinquefasciatus PRVABC59 (4 passages) MR766 (149 passages) 41525 (isolated Aedes, 1 passage C6,36; 4 vero cells) 7.2 $\log_{10}$ PFU/ml 6.7 $\log_{10}$ PFU/ml 28°C, 70% RH 14 h:10 h L:D | Aedes aegypti Mexico (F11-13), Cx. quinquefasciatus (Sebring County FLA 1988), Cx. pipiens (Pennsylvania 2002), Cx. tarsalis (California 1953) were orally infected with the epidemic American strain PRVABC59, and evaluated at 7 and 14 d dpi. Plaque assay was used to detect virus. Compared frozen versus fresh virus, African strains had a fitness advantage in vitro, Freezing decreased IR, SDR, STR, and CTR in Ae. aegypti. In fitness experiments West (Senegal) and East (MR766) African strains out competed the American strain (PRVAC59). Except for infectious virus detected in one Cx. quinquefasciatus mosquito exposed to frozen virus, no Culex mosquito became infected. VC of Ae. aegypti varied by virus strain, with STR ranged from 60–80% and transmission efficiency (CTR) ranged from 20–70%. | QAS: ≥ 50% (TS = 30, MS = 18) Strengths: Multiple virus strains, infectious virus assays. Weakness: Used laboratory colonies and only a single virus infecting dose. |
| [47] | Dibernardo A et al. J Am Mosq Control Assoc. 2017;33:276–281 https://doi.org/10.2987/17-6664.1 Vector competence of some mosquito species from Canada for Zika virus. (Research Article) | VC: #inf./#tested (IR) #dissem/#inf. (SDR) #transm./#tested (CTR) IT—used to estimate transmission. Temperate species southern Manitoba, Canada | Ae. cinereus, Ae. euedes, Ae. fitchii, Ae. sticticus, Ae. vexans, Coq. perturbans, Cx. restuans, and Cx. tarsalis PRVABC59 KF993678 (Canadian traveler infected in Thailand) 5.4 $\log_{10}$ PFU/ml 25°C, 70–80% RH 18 h:6 h L:D | Field collected mosquitoes held at 25°C. Few of the mosquitoes became infected. Although none of the Ae. euedes, Ae. fitchii, Ae. sticticus, Cx. tarsalis, Coquillettidia perturbans contained ZIKV RNA after oral exposure. Ae vexans contained ZIKV RNA (IR = 4/131, SDR = 2/4). All other species tested were refractory. To detect the presence of a salivary gland barrier, that had either developed a disseminated infection after oral exposure or had been inoculated with ZIKV tested saliva. All the species tested had evidence of a salivary gland barrier. However, but low numbers of the Ae. vexans, Coq. perturbans and Cx. restuans had evidence of disseminated infection. Two Ae. vexans with disseminated infections also had ZIKV RNA + saliva. | QAS: ≥ 75% (TS = 33, MS = 20) Strengths: Use of field collected mosquitoes. Multiple virus strains. Confirmation of positive samples with infectious assay. Weakness: Single viral dose, reporting error in original manuscript for IR. Errors and inconsistencies in reported data. |

(Continued)

**Table 1.** (Continued)

| | Reference (Article Type) | Study Objectives | Mosquitoes/Virus/Temperature | Results | Quality Assessment (QAS) |
|---|---|---|---|---|---|
| [48] | Duchemin JB et al. Virol J. 2017;14 doi:10.1186/s12985-017-0772-y Zika vector transmission risk in temperate Australia: a vector competence study (Research Article) | VC: #inf./#tested (IR) #dissem./#tested (CDR) #transm./#tested (CTR) Local species Australia | Ae. (Och.) camptorhynchus, Ae. (Ram.) notoscriptus, Ae. aegypti Ae. albopictus, Cx. annulirostris Cx. quinquefasciatus Cambodia 2010 (GenBank KU955593) 5.8 $\log_{10}$ $\text{TCID}_{50}$/ml 25°C, 65% RH 14 h:10 h L:D | Field collected mosquitoes held at 25°C. IR (midguts) and DE (carcass containing ovaries and exoskeleton) were tested by PCR, whereas saliva was tested by $\text{TCID}_{50}$ at 14 dpi. Aedes aegypti was the most efficient vector (IR = 40/48), CDR = 39/47, CTR = 33/38). Aedes albopictus (IR = 19/26, CDR = 19/26, CTR = 20/26). Ae. notoscriptus (IR = 12/35; CDR = 2/59, CTR = 24/57). Ae. camptorhynchus: (IR = 5/18, CDR = 5/40, CTR = 5/37). Cx. quinquefasciatus (IR = 0/32, CDR = 032, CTR = 0/32) and Cx. annulirostris (IR = 0/20, CDR = 0/20, CTR = 0/20) were refractory to ZIKV. | QAS: ≥90% (TS = 34, MS = 21) Strengths: Used field collected mosquitoes, infectious assay to test saliva. Comparison to local Ae. aegypti and Ae. albopictus. Weakness. Used single strain of virus, and questionable method to assess dissemination. |
| [49] | Fernandes RS et al. Mem Inst Oswaldo Cruz. 2017;112: 577–579. doi: 10.1590/0074-02760170145 Culex quinquefasciatus from areas with the highest incidence of microcephaly associated with Zika virus infections in the Northeast Region of Brazil are refractory to the virus. (Short Communication) | VC: #inf./#tested (IR) #dissem./#tested (CDR) #transm./#tested (CTR) Site with active ZIKV transmission | Cx. quinquefasciatus F1 Recife, Campina Grande, and Rio de Janeiro >F10 Cx. quinquefasciatus and Ae. aegypti colonies. (Local Brazilian virus strains) ZIKVPE243 (6.4 $\log_{10}$ PFU/ml) ZIKVSPH (7.2 $\log_{10}$ PFU/ml) ZIKVU1 (6.6 $\log_{10}$ PFU/ml) No rearing temperature provided | Four Cx. quinquefasciatus strains were refractory to ZIKV regardless of 3 viral strains tested. Most mosquito-virus pairs evaluated at 7, 14, 21 dpi. Screening for ZIKV done with plaque assay then confirmed with PCR. Saliva was collected for testing if infection and dissemination was observed. Only one of 20 bodies of Cx. quinquefasciatus from Recife challenged with the ZIKV Rio-U1 was feebly positive at 7 dpi and the virus did not disseminate in this individual, as shown by the head repeatedly testing negative. As the virus did not disseminate in any Cx. quinquefasciatus, the saliva was not examined. For Ae. aegypti IR = 65–75% at 7 dpi and 68–100% at 14 dpi; CDR = 86–100% at 14 dpi. | QAS: ≤ 50% (TS = 30, MS = 18) Strengths: Multiple virus and mosquito strains tested from Brazil where ZIKV transmission occurred. Infectious Assay to detect virus. Weakness: Single viral dose, per strain. Did not test saliva of Ae. aegypti. Short format. Calculation of DE not clearly defined. Rearing temperature not included. |
| [50] | Gendernalik A et al. Am J Trop Med Hyg. 2017;96: 1338–1340 doi:10.4269/ajtmh.16-0963 American Aedes vexans mosquitoes are competent vectors of Zika virus. (Research Article) | VC: #inf./#tested (IR) #dissem./#tested (CDR) #transm./#tested (CTR) Temperate species | Ae. vexans F1 from N. Colorado PRVABC59 6.8 $\log_{10}$ PFU/ml 7.1 $\log_{10}$ PFU/ml 7.2 $\log_{10}$ PFU/ml 26°C, 80% RH, 16h: 8 h L:D | Conducted 3 biological replicates of mosquitoes. Infection (midguts), dissemination (legs) and transmission (saliva) were tested with Plaque assay and positives confirmed by PCR. IR ranged from 66–84% (118/148, mean 80%). CDR ranged from 3–25% (24/148, mean 16%). CTR ranged from 2–7% (7/148, mean = 5%) Wild-caught Ae. vexans from northern Colorado were highly susceptible to infection by ZIKV, however, dissemination and transmission were relatively low, indicating the existence of a moderately strong midgut escape and salivary gland barriers. | QAS: ≤ 50% (TS = 30, MS = 17) Strengths: Infectious assay, biological replicates, F1 mosquitoes. Weakness: single virus strain, single dpi evaluated and not clearly specified. |

*(Continued)*

**Table 1.** (Continued)

| Reference (Article Type) | Study Objectives | Mosquitoes/Virus/ Temperature | Results | Quality Assessment (QAS) |
|---|---|---|---|---|
| [31] Guedes RD et al. Emerg Microbes Infect. 2017;6: 1–11 https://doi.org/10.1038/emi.2017.59 Zika virus replication in the mosquito *Culex quinquefasciatus* in Brazil *(Research Article)* | VC: #inf./#tested (IR) #SG+/#tested (CSGR) VD Electron microscopy Site with active ZIKV transmission | *Ae. aegypti*, (F1-F2, colony 1996) *Cx. quinquefasciatus* (colony since 2009) ZIKV BRPE243/2015 $4 \log_{10}$ PFU/ml $6 \log_{10}$ PFU/ml $26\pm2^{\circ}$C, 65–85% RH 12 h:12 h L:D | Testing of midguts (infection), salivary glands (dissemination), and FTA cards (transmission), evaluated at 3, 7 and 15 dpi at two doses. PCR used to detect ZIKV RNA. At $6 \log_{10}$ PFU/ml, *Cx. quinquefasciatus*, IR at 7 dpi (10/12) and 15 dpi (7/18), CSGR at 7 dpi (12/12), at 15 dpi (5/18) compared to *Ae. aegypti* IR = at 7 dpi (18/20), at 15 dpi (6/16), CSGR at 7 dpi (12/20) and 15 dpi (6/16). At 6 $\log_{10}$ PFU/ml, *Cx. quinquefasciatus*, IR at 7 dpi (9/25) and 15 dpi (2/19), CSGR 7 dpi (2/25) and 15 dpi (0/19) compared to *Ae. aegypti* IR 7 dpi (9/20), 15 dpi (9/18), CSGR at 7 dpi (2/18), 15 dpi (9/18). To confirm ZIKV-infective particles in salivary glands, two *Ae. aegypti* RecLab and two *Cx. quinquefasciatus*-positive samples collected at 7 dpi were inoculated in VERO cells for 10 days. Virus particles in salivary glands were observed by Electron Microscopy. From 270 pooled samples of adult female *Cx. quinquefasciatus* and 117 pools of *Ae. aegypti* mosquitoes assayed by RT-qPCR, three *Cx. quinquefasciatus* and two *Ae. aegypti* pools were positive for ZIKV. | QAS: < 25% (TS = 27, MS = 17) Strengths: Tested two doses of virus and multiple dpi. Weakness: Used colonized mosquito lines, did not use infectious assays but only PCR, FTA card data presumable negative, low sample size. Electron microscopy showed virus particles but could not identify them. |
| [51] Hart CE et al. Emerg Infect Dis 2017; 23:559–560. http://dx.doi.org/10.3201/eid2303.161636 Zika Virus Vector Competency of Mosquitoes, Gulf Coast, United States *(Research Letter)* | VC: #inf./#tested (IR) #dissem./#inf. (SDR) #transm./#dissem. (STR) Compare artificial feeder and murine model | *Cx. quinquefasciatus* *Ae. taeniorhynchus* FSS13025 DAKAR41525 MEX1-7 MEX1-44 PRVABC59 $4$–$7 \log_{10}$ FFU/ml $27^{\circ}$C, 80% RH | Cohorts of 50 *Cx. quinquefasciatus* from laboratory colony (no history provided) and the field in Houston (F2) were infected by feeding on interferon type I receptor knockout mice ($4$–$7 \log_{10}$ FFU/ml, FSS13025). Experiment using artificial feeders infected *Cx. quinquefasciatus* and *Ae. taeniorhynchus* (>F10) with 3 strains of ZIKV (FSS13025 [2010 Cambodia related American strains], DAKR41525 [1985 Senegal], PRVABC59, MEX1-7 [2015 outbreak]) at a dose of 4–6 $\log_{10}$ FFU/ml. At 3, 7 14 dpi, heads and legs were tested and at day 7, 14 saliva was tested by FFA. Murine feeds at 4, 7, 6 FFU/ml tested bodies at day 7 and 14 and legs, saliva on day 14 dpi, *Ae. taeniorhynchus* fed 6 logs of Mex strain, salivary glands, legs, midguts dissected and screened by FFA was refractory to ZIKA virus. | QAS: < 25% (TS = 25, MS = 13) Strengths: used multiple virus strains, included field collected *Culex*, used infectious assay and a murine infection model. Weakness: Pulled multiple experiments together into a single paper, the methodology was not consistent. Did not include *Aedes* controls. |

*(Continued)*

**Table 1.** (Continued)

| Reference (Article Type) | Study Objectives | Mosquitoes/Virus/Temperature | Results | Quality Assessment (QAS) |
|---|---|---|---|---|
| [52] **Heitmann A et al. Euro Surveill. 2017;22.** *Doi*:10.2807/1560-7917.ES.2017.22.2.30437 Experimental transmission of Zika virus by mosquitoes from central Europe *(Research Article)* | VC: #inf./#tested (IR) #transm./#inf. (STR) Role of rearing temperature Temperate species | *Ae. aegypti* *Ae. albopictus* *Cx. p. molestus* *Cx. p. pipiens* *Cx. torrentium* FB-GWUH-2016 7 log₁₀ PFU/ml 18°C and 27°C | *Ae. aegypti* (Bayer company); *Cx. p. molestus* (Heidelberg, GER) in colony since 2011, *Cx p. pipiens* (F0, collected in Hamburg, Germany summer 2016), *Cx torrentium* (F0, Hamburg, GER), *Ae. albopictus* (F7, Freiburg GER), and *Ae. albopictus* (Calabria, Italy) were infected a single low passage ZIKV strain) and evaluated IR (bodies) and TR (saliva) at 14 and 21 dpi at 18°C and 27°C, by PCR with confirmation by Plaque Assay. *Cx. p. pipiens* IRs at 14 dpi were 12/41 at 18°C and 7/29 at 27°C and at 21 dpi were 2/32 at 18°C and 12/38 at 27°C. *Cx. p. pipiens* IRs at 14 dpi were 16/34 at 18°C and 3/37 at 27°C and at 21 dpi were 3/32 at 18°C and 0/35 at 27°C. *Cx. torrentium* IRs at 14 dpi were 11/35 at 18°C and 4/36 at 27°C and at 21 dpi were 1/38 at 18°C and 0/34 at 27°C. For *Ae. aegypti* and *Ae. albopictus* transmission (saliva) was observed only at 27°C ranging from 13–33%. No ZIKV was detected in any *Culex* species tested. | QAS: < 25% (TS = 24, MS = 14) Strengths: Evaluated temperature of incubation, multiple mosquito strains, recent field material included. Weakness: Selection of mosquito strains appeared to be fishing expedition. Transmission measured at # saliva+/i# infected. |
| [53] **Kenney JL et al. Am J Trop Med Hyg 2017; 96: 1235–1240.** *doi*:10.4269/ajtmh.16-0865 Transmission incompetence of *Culex quinquefasciatus* and *Culex pipiens pipiens* from North America for Zika virus *(Research Article)* | VC: #inf./#tested (IR) #dissem/#tested (CDR) #transm./#tested (CTR) IT Temperate species | *Cx. quinquefasciatus* *Cx. pipiens* *Ae. aegypti* (>>>F10) MR766 PRVABC59 R103451-Honduras 2016 4–7.1 log₁₀ PFU/ml 28°C, 70–75% RH | Vector competence studies for laboratory strains (>>>F10) of *Cx. quinquefasciatus* (Sebring, FL 1998), *Cx. pipiens* (Chicago 2010), and *Ae. aegypti* (Poza Rica, Mexico). Mosquitoes were orally infected with 3 strains of ZIKV (MR766, PRVABC59, and R103451-Honduras 2016) at doses ranging from 4–7.1 log₁₀ PFU/ml and held at 28°C, with infection (bodies), dissemination (legs +wings) and transmission (saliva) evaluated at 14 dpi using plaque assay and qPCR. IT inoculations and in vitro growth in mosquito cell lines were examined. *Ae. aegypti*, IR = 100%, CTR = 67%; *Cx. quinquefasciatus*, IR = 0–1% (1/108), CTR = 0%; *Cx. pipiens*, IR = 5–10% (5/58), CTR = 0. For IT inoculated mosquitoes, IR = 15–70% in *Cx. quinquefasciatus*, 61% in *Cx. pipiens*, and 100% in *Ae. aegypti*, but saliva was positive only for *Ae. aegypti* (67%), In vitro experiments showed significant growth restriction in *Culex* cells. | QAS: ≤ 50% (TS = 30, MS = 18) Strengths: IT inoculations provide additional evidence against *Culex* vectoring ZIKV. Infectious assay. In vitro studies. Inclusion of *Ae. aegypti*. Multiple virus strains. Weakness: Using mosquito laboratory colonies. |
| [54] **Liu Z et al. Emerg Infect Dis. 2017;23: 1085–1091** https://dx.doi.org/10.3201/eid2307.161528 Competence of *Aedes aegypti, Ae. albopictus*, and *Culex quinquefasciatus* mosquitoes as Zika virus vectors, China. *(Research Article).* | VC: #inf./#tested (IR) #dissem./#inf. (SDR) #transm./#dissem. (STR) #transm./#tested (CTR) Local species China | *Ae. aegypti* *Ae. albopictus* *Cx. quinquefasciatus* (Colonies > 20 years) Asian lineage (KU820899.2) Human origin China 2016 5.45 ± 0.38 log₁₀ copies/μl 27±1°C, 70–80% RH 16 h: 8 h L:D | 18–30 mosquitoes were examined at 0, 4, 7, 10, and 14 dpi. Tested midguts, heads, and salivary glands. *Cx. quinquefasciatus* IR was 22/138 (15.9%), with no dissemination or transmission; *Ae. aegypti* IR = 124/138 (90%), SDR = 91/124 (73.4%), STR = 78/124 (63%), CTR = 78/138 (57%); *Ae. albopictus* IR = 121/138 (88%), SDR = 51/121 (42%), STR = 29/121 (24%), CTR = 29/138 (21%). No transmission rates were calculated based on salivary glands incorrectly. SDRs were observed as early as 4 dpi, increasing rapidly to 100% by 7 dpi. For *Ae. albopictus* mosquitoes' dissemination was first detected at 7 dpi but was lower overall than for *Ae. aegypti* at the same time points. Zika virus was not detected in the head tissues of *Cx. quinquefasciatus* mosquitoes. | QAS: < 25% (TS = 25, MS = 14) Strengths: Monitored the kinetics of infection from 0 to 14 dpi. Weakness: Single virus strain, old laboratory mosquito strains, did not measure transmission appropriately directly testing salivary glands, tested only with PCR. |

*(Continued)*

**Table 1.** (Continued)

| Reference (Article Type) | Study Objectives | Mosquitoes/Virus/ Temperature | Results | Quality Assessment (QAS) |
|---|---|---|---|---|
| [55] **Lourenço-de-Oliveira R et al. J Gen Virol. 2018;99: 258–264.** *Doi:*10.1099/jgv. 0.000949 *Culex quinquefasciatus* mosquitoes do not support replication of Zika virus. *(Short Communication)* | VC: #inf./#tested (IR) #dissem/#inf. (SDR) #transm./#dissem. (STR) Role of Wolbachia endosymbionts on VC. | *Cx. quinquefasciatus* containing or free of *Wolbachia* NC-2014-5132 7 log$_{10}$ TCID$_{50}$/ml Rearing temperature not provided | All *Cx. quinquefasciatus* lines challenged with ZIKV were refractory to the virus whether they contained Wolbachia or not. Used plaque assays to detect virus. No infection, dissemination or transmission was detected in any of the mosquito lines at 7 and 14 dpi. ZIKV does not replicate to detectable levels in mosquito cells following a blood meal, in line with the infectivity data described above. | QAS: < 25% (TS = 25, MS = 13) *Strengths:* Used infectious assay. Weakness: Single virus strain, but authors were more oriented in finding how *Wolbachia* might account for refractoriness of *Culex*. Rearing temperature was not reported. |
| [56] **O'Donnell KL. Et al. J Med Entomol. 2017;54:1354–1359.** *Doi:*10.1093/jme/ tjx087 Potential of a Northern population of *Aedes vexans* (Diptera: Culicidae) to transmit Zika virus. *(Research Article)* | VC: #inf./#tested (IR) #dissem/#inf. (SDR) #dissem./#tested (CDR) IT Temperate species Upper Great Plains | *Ae. vexans* (F1), *Ae. aegypti* (F39) Unspecified from Puerto Rico 2016 Fresh: 5.3 log$_{10}$ PFU/ml Frozen: 7.0 log$_{10}$ PFU/ml 28°C, 16 h: 8 h L:D | Infection (bodies), dissemination (legs) and transmission, through IT inoculation were assessed. Mosquitoes were challenged with thawed frozen and fresh virus from cell culture and held at 28°C. For *Ae. vexans* challenged with thawed virus, IR = 29% (8/28) and CDR = 12% (1/8) compared to those fresh virus IR = 28% (9/32) and CDR = 3% (1/32). For *Ae. aegypti* challenged with thawed virus, the IR = 61% (11/18), SDR = 36% (4/11). Saliva tested in mosquitoes infected IT on 16–17 dpi. *Ae. aegypti* had significantly higher rates of viral infection and dissemination than *Ae. vexans*. All 47 inoculated *Ae. vexans* and 22 of 23 inoculated *Ae. aegypti* were positive for Zika virus. | QAS: > 50% (TS = 31, MS = 19) *Strengths:* Used field captured mosquitoes. Compare frozen and fresh virus preps. Weakness: PCR only for detection of viral RNA, single virus strain. |
| [57] **Calvez E et al. PloS NTD 2018;12: e0006637** https://doi.org/10.1371/journal.pntd. 0006637 Zika virus outbreak in the Pacific: vector competence of regional vectors *(Research Article)* | VC: #inf./#tested (IR) #dissem/#inf. (SDR) #transm./#dissem. (STR) #transm./#tested (CTR) Local species French Polynesia | *Ae. aegypti* (3 populations from French Polynesia, New Caledonia, Samoa (F1-F3) *Ae. polynesiensis* (2 populations from French Polynesia, Wallis and Futuna F1-F3) NC-2014-5132 7 log$_{10}$ TCID$_{50}$/ml 28°C, 80% RH, 16h: 8 h L:D | Examined distinct local *Ae. aegypti* and *Ae. polynesiensis* populations across the south pacific at 6, 9, 14, and 21 dpi. After oral challenge, mosquito infection (bodies), dissemination (heads) and transmission (saliva) were evaluated by plaque assay for ZIKV. For French Polynesian population of *Ae. polynesiensis*: IR 6 dpi (34/68), 9 dpi (64/86), 14 dpi (84/108), 21 dpi (71/92); SDR 6 dpi (6/33), 9 dpi (10/57), 14 dpi (32/71), 21 dpi (36/60); STR 14 dpi (1/32), 21 dpi (2/36); CTR at 14 dpi (1/72), 21 dpi (2/69). *Ae. aegypti* populations had significant heterogeneity in VC parameters and low competence overall among the three mosquito populations tested CTR was 0%, 6%, and 17%. | QAS: ≥ 99% (TS = 35, MS = 21) *Strengths:* Used Infectious assay for detection of ZIKV, provided raw data in supplementary data, geographically diverse mosquitoes strains recently from field, range of dpi studied. High sample sizes. Weakness: Single virus strain. |
| [58] **Dodson BL & Rasgon JL. Peer J 2017** doi: 10.7717/peerj.3096 Vector competence of *Anopheles* and *Culex* mosquitoes for Zika virus *(Research Article)* | VC: #inf./#tested (IR) #dissem./#tested (CDR) #transm./#tested (CTR) | *An. gambiae, An. stephensi Cx. quinquefasciatus* (>F10) PRVABC59 MR766 4.6–7.7 log$_{10}$ PFU/ml 27±1°C, 12 h: 12 h L:D | Vector competence studies carried out for laboratory strains (>F10) of *An. gambiae, An. stephensi* (Liston strain, Johns Hopkins University) and Cx. quinquefasciatus (Wadsworth Center, originally from Benzon Research) infected with ZIKV (MR766 Uganda prototype and PRVABC59) at 6 doses, by orally infecting mosquitoes on artificial feeder and human blood. Mosquitoes were held at 27°C. Infection (bodies), dissemination (legs), and transmission (saliva) were evaluated by Plaque assay at 5, 7, 14 dpi. No species were infected at any time point. | QAS: < 25% (TS = 27, MS = 15) *Strengths:* Used multiple virus strains and infectious doses and evaluated competence parameters at 3 time points. Weakness: Used highly passaged frozen virus, no *Ae. aegypti* control, and colonized mosquitoes. |

*(Continued)*

**Table 1.** (Continued)

| Reference (Article Type) | Study Objectives | Mosquitoes/Virus/ Temperature | Results | Quality Assessment (QAS) |
|---|---|---|---|---|
| [59] **Jansen S et al. Emerg Microbes Infect. 2018;7: 192** doi:10.1038/s41426-018-0195-x Experimental transmission of Zika virus by *Aedes japonicus japonicus* from southwestern Germany (*Research Article*) | VC: #inf./#tested (IR) #dissem/#inf. (SDR) #transm./#dissem. (STR) #transm./#tested (CTR) Temperate species Role of rearing temperature | *Ae. japonicus japonicus* (F1) ZIKV_FB-GWUH-2016 (KU870645) 7 $\log_{10}$ PFU/ml 21°C, 24°C, and 27°C, 80% RH | Groups of 20 females, previously screened by pan-Flavi-, pan-Bunya- and pan-Alphavirus by PCR were exposed to blood meal with ZIKV and bodies, legs, and saliva were evaluated at 14 dpi by PCR except for saliva, also tested at 14 dpi by plaque assay. IR = 10% (3/30) at 21°C, 24% (7/29) at 24°C, 66.7% (14/21) at 27°C. SDR, 0% at 21°C, ~26% at 24°C, ~10% at 27°C, but RNA copies were higher at 27°C. STR, 14% (2/14) at 27°C and CTR of 9.5% (2/21) at 27°C. | QAS: > 50% (TS = 31, MS = 19) Strengths: Mosquitoes from field, used infectious assay to assess transmission. Role of temperature. Weakness: single virus strain and titer. Difficult to extract metadata, especially positive legs. |
| [60] **Karna AK et al. Viruses. 2018; 10:434** doi:10.3390/v10080434 Colonized *Sabethes cyaneus*, a sylvatic New World mosquito species, shows a low vector competence for Zika virus relative to *Aedes aegypti*. (*Research Article*) | VC: #inf./#tested (IR) #dissem/#inf. (SDR) #transm./#dissem. (STR) Forest species Mouse model | *Sa. cyaneus* (since 1988), *Ae. aegypti* (F7) ZIKV MEX 1-7 (isolated from *Ae. aegypti* 2015) 5 $\log_{10}$ PFU/ml Infected infr/- mouse 27±1°C, 80% RH, 16 h: 8 h L:D | After oral exposure to virus infected blood, feeding rates for *Sa. cyaneus* were low, with 21% feeding on mice at 1 dpi and 28% feeding on mice at 2 dpi; in contrast, more than 85% of *Ae. aegypti* fed on mice on each day. Of 69 engorged *Sa. cyaneus*, ZIKV was detected in only one individual, albeit in all body compartments sampled (body, legs, and saliva) at 21 dpi This mosquito had fed on a mouse at day 2 dpi; titers increased in the mice by approximately tenfold between day 1 and day 2 dpi. In contrast, *Ae. aegypti* showed high levels of ZIKV infection, dissemination, and transmission. | QAS: < 50% (TS = 28, MS = 14) Strengths: Infectious assay to detect ZIKV, wide range of dpi, *Ae. aegypti* comparator. Weakness: Old colony, did not blood feed well sample size, single specimen became infected, showed dissemination and transmission. |
| [61] **Main BJ et al. PloS NTD 2018;12: e0006524** https://doi.org/10.1371/journal.pntd.0006524 Vector competence of *Aedes aegypti*, *Culex tarsalis*, and *Culex quinquefasciatus* from California for Zika virus. (*Research Article*) | VC: #inf./#tested (IR) #dissem/#tested (CDR) #transm./#tested (CTR) Temperate species California USA Mouse model | *Ae. aegypti (F6)*, Cx. *quinquefasciatus(F5)*, Cx. *tarsalis (>F10)* PRVABC59 (KX601168), MA66, P6-740 (KX601167.1), SPH2015 (KU321639); 5 $\log_{10}$ PFU/ml 26°C, 80% RH, 12 h: 12 h L:D (*Ae. aegypti* and Cx. *tarsalis*) 22°C, 33% RH (RT) (Cx. *quinquefasciatus*) | *Ae. aegypti* and Cx. *tarsalis* held at 26°C whereas Cx *quinquefasciatus* were held at 22°C and challenged with PR15 5.4–6.4 $\log_{10}$ PFU/ml. IR = 4% (2/46) at 14 dpi, 30% (6/20) at 21 dpi; CDR 4% (2/46) at 14 dpi, and 5% (1/20) at 21 dpi, CTR = 0% at 14 and 21 dpi. Cx. *quinquefasciatus* were refractory at 14 and 21 dpi. In contrast, *Ae. aegypti*: challenged with ZIKV MA66: IR = 86% (73/85) at 14 dpi and 96% (22/23) at 21 dpi; CDR = 79% (69/85) at 14 dpi and 91% (21/23) at 21 dpi; CTR = 53% (45/85) at 14 dpi and 87% (20/23). For ZIKV PR15, the infection, dissemination, and transmission rates on 14 dpi were 85%, 78%, and 65%, respectively. For ZIKV BR15 harvested 15 dpi had infection, dissemination, and transmission rates of 90%, 90%, and 75%, respectively. | QAS: ≥ 90% (TS = 34, MS = 21) Strengths: Cx. *quinquefasciatus* colony F5, *Ae. aegypti* comparator, confirmation of infectious virus. Range doses with Cx. *tarsalis*. Good sample size. Weakness: Cx. *tarsalis* old colony, single strain of virus, retrospective confirmation of infectious virus. |
| [32] **Smartt TC et al. Front Microbiol. 2018;9: 768.** doi: 10.3389/fmicb.2018.00768 *Culex quinquefasciatus* (Diptera: Culicidae) from Florida transmitted Zika virus. (*Research Article*) | VC: #inf./#tested (IR) Transmission verified through presence of ZIKV in saliva eluted from FTA cards | Cx. *quinquefasciatus* (>F10) Asian lineage, Gen Bank KU501215.1 5.7 $\log_{10}$ PFU/ml 28°C, 80% RH | IR (bodies) at 16 dpi revealed 9 female mosquito bodies with ZIKV RNA. Analysis of RNA in saliva eluted from the filter paper at 16 dpi revealed an average titer of 5.6 ± 4.5 $\log_{10}$ ZIKV PFU/ml per card and there was no significant difference in the titers per cage. The mosquitoes from the cages revealed positive bodies in cages 1 and 2 (IR = 55%). Plaque assays of the saliva samples eluted from the filter paper cards were positive for ZIKV infectious virus. | QAS: < 25% (TS = 21, MS = 12) Strength: Infectious virus confirmation of ZIKV in FTA card showing transmission. Weakness: Difficult to calculate rates, poor presentation of experiment metadata. |

(*Continued*)

**Table 1.** (Continued)

| Reference (Article Type) | Study Objectives | Mosquitoes/Virus/ Temperature | Results | Quality Assessment (QAS) |
|---|---|---|---|---|
| [62] **Ben Ayed W et al. J Med Entomol. 2019; 56:1377–1383.** *Doi:* 10.1093/jme/tjz067 A survey of *Aedes* (Diptera: Culicidae) mosquitoes in Tunisia and the potential role of *Aedes detritus* and *Aedes caspius* in the transmission of Zika virus. (*Research Article*) | VC: #inf./#tested (IR) #dissem/#inf. (SDR) #transm./#dissem. (STR) #transm./#tested (CTR) (Tunisia) Entomology survey | *Ae. caspius* (F1) *Ae. detritus* (F1) NC-2014-5132 $7.2 \log_{10}$ PFU/ml $28 \pm 1°C$, 80% RH, 16 h: 8 h L:D | *Aedes detritus:* Owing to the low survival rate in the laboratory and low feeding rates in BSL-3 conditions, few mosquitoes were examined at 14 dpi. Of these, IR = 75% (3/4), SDR = 0%, STR = 0%. *Aedes caspius:* IR 4% (1/24) and SDR = 0% at 7 dpi. At 14 dpi, IR = 10% (2/20), SDR = 100%l (2/2) and STR = 0%. Thus, neither species were competent to transmit ZIKV. | QAS: ≤ 50% (TS = 30, MS = 18) Strength: Infectious assay for bodies and heads. F1 mosquito strains. Weakness: Single virus strain, low numbers, PCR only to detect virus. |
| [63] **Elizondo-Quiroga D et al. Sci Rep 2019;9: 16955.** https://doi.org/10.1038/s41598-019-53117-1 Vector competence of *Aedes aegypti* and *Culex quinquefasciatus* from the metropolitan area of Guadalajara, Jalisco, Mexico for Zika virus. (*Research Article*) | VC: #inf./#tested (IR) #dissem./#inf. (SDR) #dissem./#tested (CDR) #transm./#dissem. (STR) #transm./#tested (CTR) Site with active ZIKV transmission | *Ae. aegypti* (F0) *Cx. quinquefasciatus.* (F0) 3 virus strains from Mexico $4.7 \log_{10}$, $5.2 \log_{10}$, $5.6 \log_{10}$, and $6.4 \log_{10}$ $TCID_{50}$/ml $28 \pm 1°C$, 80% RH, 12 h: 12 h L:D | Bodies, heads, and saliva evaluated at 14 dpi and different virus titers. *Cx. quinquefasciatus* were refractory at all virus concentrations. *Ae. aegypti* infection did not occur at low ($4.7 \log_{10} TCID_{50}$/ml) virus concentration, but at medium concentrations ($5.2–5.6 \log_{10} TCID_{50}$/ml), IR = 38%, SDR = 24%, STR = 32% and CTR = 7.7%. At high virus concentration ($6.4 \log_{10} TCID_{50}$/ml), IR = 93%, SDR = 72%, STR– 19.3%, and CTR = 14%. | QAS: ≥ 75% (TS = 33, MS = 19) Strength: Infectious assay, F1 mosquito strains, range of virus doses. Multiple virus strains. Weakness: Poor metadata on virus strain. |
| [64] **Fernandes RS et al. Sci Rep. 2019;9: 20151.** https://doi.org/10.1038/s41598-019-56669-4 Low vector competence in sylvatic mosquitoes limits Zika virus to initiate an enzootic cycle in South America. (*Research Article*) | VC: #inf./#tested (IR) #dissem./#inf. (SDR) #transm./#dissem. (STR) #transm./#tested (CTR) IT Forest species. | *Haemagogus leucocelaenus, Ae. terrens, Ae. scapularis, Sa. identicus, Sa. albiprivus* (field) Rio-U1 Rio-S1 Oral dose: $6.0 \log_{10}$ PFU/ml IT dose: $6.5 \log_{10}$ PFU/ml $28 \pm 1°C$, $80 \pm 10$% RH 12 h: 12 h L:D | 30 mosquitoes examined (bodies, heads, saliva) at 7, 14, and 12 dpi. *Hg. leucocelaenus* infected with ZIKV Rio-S1 had IRs at 7 dpi (4/20), 14 dpi (7/21), and 21 dpi (12/30) and SDR at 7 dpi (1/4), 14 dpi (1/7), and 21 dpi (1/12), but no transmission was observed. In contrast, when challenged with ZIKV Rio-U1 IR was lower at 14 dpi (4/30) and neither dissemination nor transmission was observed. *Sa. albiprivus* was refractory to ZIKV Rio-U1 at 7, 14, and 21 dpi, but with ZIKV Rio-S1 had low IR (1/32) only at 14 dpi with no infection at 7 or 21 dpi. No further dissemination was observed. *Ae. scapularis* challenged only with ZIKV Rio-S1 had low IR (1/42) but no further dissemination. *Ae. terrens* and *Sa. identicus* were completely refractory to ZIKV challenge. Transmission (virus present in saliva) was not detected in any mosquito orally challenged with ZIKV, regardless of viral isolate and incubation time. Dissemination and transmission were observed in *Hg. leucocelaenus, Sa. albiprivus, and Sa. identicus* after IT inoculation. | QAS: ≥ 75% (TS = 33, MS = 20) *Strengths:* Use of field collected mosquitoes; Use of infectious virus assay, >1 virus strain, infectious assays to detect ZIKV. Weakness: Low numbers for some species. |

(*Continued*)

Table 1. (Continued)

| Reference (Article Type) | Study Objectives | Mosquitoes/Virus/Temperature | Results | Quality Assessment (QAS) |
|---|---|---|---|---|
| [65] **Gutiérrez-López R et al. Emerg Infect Dis. 2019;25: 346–348** https://doi.org/10.3201/eid2502.171123 Vector competence of *Aedes caspius* and *Ae. albopictus* mosquitoes for Zika virus, Spain. (*Dispatches*) | VC: #inf./#tested (IR) #dissem/#inf. (SDR) #transm./#dissem. (STR) Temperate species (Spain) | *Ae. albopictus* (F2) *Ae. caspius* (F0) *Ae. aegypti* (F8) CAM (JN860885): PR (KU501215) 7.6 log₁₀ PFU/ml No rearing temperature provided | Bodies, legs, saliva tested by PCR at 7, 14, and 21 dpi. *Aedes caspius* had IRs at 7 dpi (21.4% [3/14]), 14 dpi (40% [10/25]), and 21 dpi (18.5% [5/27]), no virus dissemination or transmission was detected at any point with Puerto Rican Virus strain. *Aedes albopictus* Cambodia strain. IR = 90.5% at 7 dpi, 82% at 14 dpi and 94.4% at 21 dpi; SDR = 42% at 7 dpi, 82% at 14 and 21 dpi; STR = 10.5% at 7 dpi, 9% at 14 dpi, 23.6% at 21 dpi. Puerto Rico Strain: IR = 97% at 7 dpi, 93% at 14 dpi, 96% at 21 dpi; SDR = 31% at 7 dpi, 68% at 14, 96% at 21 dpi; STR = 0% at 7 and 14 dpi, 36% at 21 dpi. *Aedes aegypti* Cambodia, IR = 24% at 7 dpi, 23% at 14 dpi and 36% at 21 dpi; SDR = 75% at 7 dpi, 71% at 14 and 100% at 21 dpi; STR = 12.5% at 7 dpi, 14.3% at 14 dpi, 40% at 21 dpi. Puerto Rico Strain: IR = 62% at 7 dpi, 45% at 14 dpi, 56% at 21 dpi; SDR = 38% at 7 dpi, 78% at 14, 89% at 21 dpi; STR = 0% at 7, 16.7% at 14 dpi, 39% at 21 dpi. | QAS: ≥ 75% (TS = 33, MS = 21) *Strengths*: Mosquito strains from field or recently colonized, > 1 virus strain (only 1 for *Ae. caspius*). *Weakness*: Single virus strain for *Ae. caspius*, PCR only for virus detection. No rearing temperature reported. |
| [66] **Hery L et al. Emerg Microbes Infect. 2019;8:699–706.** https://doi.org/10.1080/22221751.2019.1615849 Transmission potential of African, Asian and American Zika virus strains by *Aedes aegypti* and *Culex quinquefasciatus* from Guadeloupe (French West Indies). (*Research Article*) | VC: #inf./#tested (IR) #dissem./#inf. (SDR) #dissem./#tested (CDR) #transm./#dissem. (STR) #transm./#tested (CTR) Site with active ZIKV transmission | *Ae. aegypti* (F1) *Cx. quinquefasciatus.* (F0) Senegal (KU955592) Martinique (KU647676) Malaysia (KX694533) 7 log₁₀ TCID₅₀/ml 27±1°C, 70% RH, 12 h: 12 h L:D | For *Cx. quinquefasciatus* no infection, nor dissemination nor transmission was detected for any of the ZIKV strains at 7, 14 or 21 dpi. For *Ae aegypti* Senegal strain, IR = 90% at 7 dpi, 92% at 14 dpi, and 84.6% at 21 dpi; CDR = 96.3% at 7 dpi, 91.3% at 14 dpi, 95.5% at 21 dpi, CTR 42.3% at 7 dpi, 62% at 14 dpi, 76.2% at 21 dpi. For *Malaysia* strain, IR = 23.3% at 7 dpi, 23.3% at 14 dpi, 16.7% at 21 dpi; CDR = 28.6% at 7 dpi, 71.4% at 14 dpi, 80% at 21 dpi. CTR 0% at 7 dpi, 20% at 14 dpi, 75% at 21 dpi. Martinique strain, IR = 23.3% at 7 dpi, 37% at 14 dpi, 27% at 21 dpi; CDR = 29% at 7 dpi, 54.5% at 14 dpi, 62.5% at 21 dpi, TR 0% at 7 and 14 dpi, 80% at 21 dpi. | QAS: ≥ 90% (TS = 34, MS = 22) Mosquito strains from field, multiple virus strains, range of dpi, infectious assays. Weakness: Single virus dose. |
| [67] **Núñez AI et al. Parasit Vectors. 2019;12: 363.** https://doi.org/10.1186/s13071-019-3620-7 European *Aedes caspius* mosquitoes are experimentally unable to transmit Zika virus. (*Research Article*) | VC: #inf./#tested (IR) #dissem/#inf. (SDR) #transm./#dissem. (STR) #transm./#tested (CTR) Temperate species (Spain) | *Ae. caspius* (F0) *Ae. aegypti* (since 1994) Suriname (EVAg no. 011V-01621; Asian lineage), MR766 (African I lineage) 7 log₁₀ TCDI₅₀/ml 26/22°C (day/night), 80% RH 14 h: 10 h L:D | After virus challenge, infection (bodies), dissemination (legs), and transmission (saliva) evaluated at 7, 14, and 21 dpi. Screened by PCR, confirmation with plaque assay. *Aedes caspius* Suriname, 0% IR, SDR, and STR at all time points. ZIKV RNA was detected at low levels at 14 and 21 dpi. MR766 (African I lineage) 0% IR, SDR, and STR at all time points, also with low levels of ZIIKV detection by RT-qPCR. *Aedes aegypti* Suriname, IR = 85% at 7 dpi, 100% at 14 dpi, 95% at 21 dpi; SDR = 45% at 14 dpi, 84% at 21 dpi, STR = 33% at 14 dpi, 6.2% at 21 dpi; MR766, IR = 10% at 7 and 14 dpi, 5.2% at 21 dpi, but no dissemination or transmission | QAS: ≤ 50% (TS = 29, MS = 18) *Strengths*: Use of field collected mosquitoes; Use of infectious virus assay, >1 virus strain. Weakness: Low numbers for DR and TR assessments, single virus strain. |

(Continued)

**Table 1.** (Continued)

| | Reference (Article Type) | Study Objectives | Mosquitoes/Virus/Temperature | Results | Quality Assessment (QAS) |
|---|---|---|---|---|---|
| [68] | **Abbo SR et al. PloS NTD 2020; 14: e0008217.** *Doi:* https://doi.org/10.1371/journal.pntd.0008217 The invasive Asian bush mosquito *Aedes japonicus* found in the Netherlands can experimentally transmit Zika virus and Usutu virus. *(Research Article)* | VC: #inf./#tested (IR) #dissem./#tested (CDR) #transm./#tested (CTR) IT RNA replicative intermediates Entomology survey Temperate species | *Ae. japonicus* (F0) Ae. aegypti (control, Rockefeller) Suriname 2016 USUV (MH891847.1) 7.2 $\log_{10}$ TCID$_{50}$/ml 28° C, 12 h: 12 h L:D | Assessed infection (bodies), dissemination (legs+wings), and transmission (saliva) at 14 dpi at 28° C for both ZIKV and Usutu (USUV). IT inoculations were also performed. For ZIKV-blood fed mosquitoes, IR = 10% (6/62), CDR = 8% (5/62), and CTR = 3% (2/62). For USUV-blood fed mosquitoes, IR = 13% (4/30), CDR = 13% (4/30), and CTR = 13% (4/30). Of the intrathoracically injected mosquitoes, 96% (ZIKV) and 88% (USUV) showed virus-positive saliva at 14 dpi. Small RNA deep sequencing of orally infected mosquitoes confirmed active replication of ZIKV and USUV, as demonstrated by potent small interfering RNA responses against both viruses. Additionally, *de novo* small RNA assembly revealed the presence of a novel narnavirus in *Ae. japonicus. Ae. aegypti* controls IR = 100%, CDR = 100%, CTR = 80%), | QAS: > 50% (TS = 32, MS = 21) *Strengths:* Use of field collected mosquitoes; Intrathoracic injections to identify mechanistic barriers to transmission. Use of infectious virus assay. Weakness: No statistical analysis or description or group replicates. |
| [69] | **Abbo SR et al. Viruses. 2020;12;659** *doi:*10.3390/v12060659 Forced Zika virus infection of *Culex pipiens* leads to limited virus accumulation in mosquito saliva. *(Research Article)* | VC: #inf./#tested (IR) #transm./#tested (CTR) IT Temperate species | *Cx. p. molestus, Cx. p. pipiens (colony)* Suriname 2016 USUV (MH891847.1) 7.0 $\log_{10}$ TCID$_{50}$/ml 28° C | Infection (bodies) and transmission (saliva) determined at 14 dpi. ZIKV IR for *Cx. pipiens* (2/133) but no transmission (0/133) observed after an infectious blood meal with ZIKV titers in *Cx. pipiens* observed to be low. ZIKV IR for *Ae. aegypti* was 100% (121/121) and CTR was 65% (79/121). *Cx. molestus* was refractory to ZIKV. The infection and transmission of potential of ZIKV-injected *Cx. pipiens* was dependent on the viral dose provided. Viral dissemination into the saliva of *Cx. pipiens* does not always correlate with a high viral titer in the mosquito body. *Cx. pipiens* is an inefficient vector for ZIKV. | QAS: < 25% (TS = 27, MS = 17) Strengths: Infectious assay to detect ZIKV. Weakness: Used colonized mosquito lines with poorly described history. |

*(Continued)*

**Table 1.** (Continued)

| | Reference (Article Type) | Study Objectives | Mosquitoes/Virus/ Temperature | Results | Quality Assessment (QAS) |
|---|---|---|---|---|---|
| [70] | **Blagrove MSC et al. Proc Biol Sci 2020;287:2020019** http://dx.doi.org/10.1098/rspb.2020.0119 Potential for Zika virus transmission by mosquitoes in temperate climates. *(Research Article)* | VC: #inf./#tested (IR) #transm./#tested (CTR) Temperate species Role of rearing temperature Modeling | *Ae. albopictus* (Verano F3) *Ae. detritus* (F0) (reported as *Oc. detritus* in original article) PE243 (Brazil) 6 log$_{10}$ PFU/ml 17°C, 19°C, 21°C, 24°C, 27°C and 31°C 70% RH, 12 h: 12 h L:D | Mortality and competence of wild-obtained *Ae. detritus* and colony *Ae. albopictus* were tested at six different temperatures: 17°C, 19°C, 21°C, 24°C, 27°C and 31°C. Adult females were sacrificed at eight time points: 0, 5, 7, 10, 14-, 17-, 21- and 28-days dpi. Zika RNA was detected in the saliva of both *Ae. albopictus* (Verano colony) and *Ae. detritus* at all temperatures from 19°C to 31°C, as early as 7 dpi at 31°C for *Ae. albopictus* and 10 dpi at 27°C and 31°C for *Ae. detritus*. ZIKV titers in *Ae. albopictus* saliva were 3.8x higher than in *Ae. detritus*. *Ae. detritus* detected starting at 7 dpi and 19°C until 28 dpi and 31°C (IR at 19°C: 14 dpi = 1/12, 17 dpi = 2/19, 21 dpi = 3/16, 28 dpi = 3/18; CTR at 19°C: 17 dpi = 1/19, 28 dpi = 1/18; IR 21°C: 14 dpi = 5/20, 17 dpi = 4/15, 21 dpi = 3/16, 28 dpi = 4/16; CTR at 21°C: 14dpi = 4/20, 17 dpi = 1/15, 21 dpi = 1/16, 28 dpi = 2/16; IR 24°C: 10 dpi = 1/9, 14 dpi = 1/17, 17 dpi = 3/21, 21 dpi = 5/16, 28 dpi = 3/17; CTR at 24°C: 14dpi = 1/17, 17 dpi = 1/21, 21 dpi = 3/16, 28 dpi = 3/17; IR 27°C: 10 dpi = 2/10, 14 dpi = 3/21, 17 dpi = 3/17, 21 dpi = 2/15; CTR at 27°C: 10 dpi = 1/10, 14dpi = 2/21, 17 dpi = 3/17, 21 dpi = 2/15; IR at 31°C: 10 dpi = 2/9, 14 dpi = 2/13, CTR at 31°C: 10 dpi = 1/9, 14 dpi = 2/13. *Ae. albopictus* detected starting 7 dpi. | QAS: > 25% (TS = 27, MS = 16) Strengths: Examined a range of temperatures. Weakness: Used PCR assays to detect ZIKV RNA. Single dose and virus strain. |
| [71] | **Chan KK et al. Parasit Vectors 2020; 13:188.** https://doi.org/10.1186/s13071-020-04042-0 Vector competence of Virginia mosquitoes for Zika and Cache Valley viruses. *(Research Article)* | VC: #inf./#tested (IR) #dissem./#tested (CDR) #transm./#tested (CTR) IT Temperate species (Virginia) | Field collected (F1) *Cx. pipiens*, *Cx. restuans*, *Ae. albopictus*, *Ae japonicus*, *Ae. triseriatus*. *Ae. aegypti* (Colony) PRVABC59 Oral dose (*Aedes*) 6.5–7.7 log$_{10}$ PFU/ml IT dose (*Aedes*) 4.7–5.3 log$_{10}$ PFU/ml Oral dose (*Culex*) 6.7–7.5 log$_{10}$ PFU/ml 24°C, 75% RH, 16h: 8 h L:D | IR (bodies), CDR (legs and wings), and CTR (saliva) determined at 14 dpi by plaque assay. Mosquitoes reared at 24°C. *Ae. aegypti:* IR = 17/25, CDR = 15/25, CTR = 12/25. *Ae. albopictus:* IR = 18/37, CDR = 15/37, CTR = 9/37. *Ae. japonicus:* IR = 15/73, CDR = 7/73, CTR = 2/73). *Ae. triseriatus:* IR = 7/28, CDR = 0/28, CTR = 0/28. *Cx. pipiens* and *Cx. restuans* were completely refractory to ZIKV infection. Transmission rates after IT inoculation: *Ae. albopictus* (63%, 12/19), *Ae. japonicus* (19%, 4/21), *Ae. aegypti* (71%, 15/21) No virus detected in the saliva of *Ae. triseriatus* from either orally (0/28) or parenterally (0/23) infected groups. Study also examined VC to Cache Valley virus (CVV). CVV was detected in the saliva of *Ae. albopictus* (high titer: 68%, low titer: 24%), *Ae. triseriatus* (high titer: 52%, low titer: 7%), *Ae. japonicus* (high titer 22%, low titer: 0%) and *Ae. aegypti* (high titer: 10%, low titer: 7%). *Culex pipiens* and *Cx. restuans* were also refractory to CVV. | QAS: > 50% (TS = 32, MS = 19) Strengths: Infectious assay to detect ZIKV. Mosquitoes from field Weakness: Single virus strain, experimental metadata difficult to extract. |

*(Continued)*

Table 1. (Continued)

| | Reference (Article Type) | Study Objectives | Mosquitoes/Virus/Temperature | Results | Quality Assessment (QAS) |
|---|---|---|---|---|---|
| [72] | **Fernandes SR et al. Pathogens. 2020; 9:** 575. Doi: 10.3390/pathogens9070575 Vector competence of *Aedes aegypti, Aedes albopictus* and *Culex quinquefasciatus* from Brazil and New Caledonia for three Zika virus lineages. (*Research Article*) | VC: #inf./#tested (IR) #dissem/#inf. (SDR) #transm./#dissem. (STR) #transm./#tested (CTR) Sites with active ZIKV transmission | 2 strains *Ae. aegypti* (F1) *Cx. quinquefasciatus* (F0– New Caledonia) 5 strains *Ae. aegypti* (F2–F4– Brazil), *Ae. albopictus* (F2–F4 Brazil) MRS_OPY_Martinique_PaRi_2015, DAK 84, MASS 66 7 $\log_{10}$ TCID$_{50}$/ml 28±1°C, 70–80% RH 12 h: 12 h L:D | After mosquito were challenged with ZIKV infected blood IR (bodies), dissemination (heads), and transmission (saliva) at 7, 14, and 21 dpi tested by Plaque assay, comparing *Ae. aegypti, Ae. albopictus,* and *Cx. quinquefasciatus* with 3 strains of ZIKV. *Cx. quinquefasciatus* only at Dumbea, were refractory. *Ae. aegypti*: At 7 dpi IR = 30–100%, SDR = 0–100%, STR = 0–85%, CTR = 0–85%; at 14 dpi IR 60–100%, SDR = 25–100%, STR = 14–100%, CTR = 3–100%; at 21 dpi, IR = 45–100%, SDR = 19–100%, STR = 0–96%, CTR = 0–90%. *Aedes albopictus*: At 7 dpi IR = 7–73%, SDR = 0–41%, STR = 0–57% CTR = 0–13%; at 14 dpi IR 7–80%, SDR = 0–83%, STR = 0–100%, CTR = 0–50%; at 21 dpi, IR = 7–80%, SDR = 10–100%, STR = 0–100%, CTR = 0–67%. There was high variability in VC based on virus strain-mosquito combinations. | QAS: > 50% (TS = 32, MS = 19) Strengths: Infectious assay to detect ZIKV. Comparing VC numerous sites. Multiple virus strains. Mosquitoes from field Weakness: Single site for *Cx. quinquefasciatus*, low sample sizes for dissemination and transmission experiments. |
| [73] | **Glavinic U et al. Parasit Vectors 2020;** **13:479.** *Doi:* https://doi.org/10.1186/s13071-020-04361-2 Assessing the role of two populations of *Aedes japonicus* for Zika virus transmission under a constant and a fluctuating temperature regime. (*Research Article*) | VC: #inf./#tested (IR) #dissem/#inf. (SDR) #transm./#dissem. (STR) #transm./#tested (CTR) Including fluctuating vs constant temperature rearing schemes. | *Ae. japonicus* (2 sites) Zurich Steinbach DAK84 Senegal 7.1 $\log_{10}$ TCID$_{50}$/ml Constant temperature: 27°C, 85% RH Fluctuating temperature: 14–27°C (mean = 23°C), 45–90% RH 16h: 8 h L:D | Infection (Abdomen/thorax), dissemination (heads), and transmission (saliva) were determined by plaque assay at 7, 14, and 21 dpi in two mosquito strains (Zurich, Steinbach) reared at constant and fluctuating temperature. For the Zurich strain held at a constant temperature: IR at 7 dpi (25/30), at 14 dpi (28/30), at 21 dpi (26/30), SDR at 7 dpi (10/25) at 14 dpi (9/28), 21 dpi (3/26), CTR at 7 dpi (7/30), at 14 dpi (6/30), 21 dpi (2/30). For the Steinbach strain held at a constant temperature: IR at 7 dpi (20/30), at 14 dpi (30/30), at 21 dpi (20/30), SDR at 7 dpi (8/20) at 14 dpi (6/30), 21 dpi (5/20), CTR at 7 dpi (2/30), at 14 dpi (3/30), 21 dpi (2/30). For the Zurich strain held at a fluctuating temperature: IR at 7 dpi (14/30), at 14 dpi (5/30), at 21 dpi (23/30), SDR at 7 dpi (10/14) at 14 dpi (3/5), 21 dpi (6/23), CTR at 7 dpi (3/30), at 14 dpi (0/30), 21 dpi (3/30). For the Steinbach strain held at a fluctuating temperature: IR at 7 dpi (22/30), at 14 dpi (18/30), at 21 dpi (14/30), SDR at 7 dpi (7/22) at 14 dpi (11/18), 21 dpi (9/14), CTR at 7 dpi (3/30), at 14 dpi (8/30), 21 dpi (1/30). SDR decreased over time regardless of the incubation temperature regimes. Saliva was positive at 7 dpi for all populations independent of incubation conditions. | QAS: < 50% (TS = 29, MS = 17) Strengths: Infectious assay to detect ZIKV. Field collected mosquitos. Comparison of different mosquito populations. Impact of fluctuating temperature. Weakness: Single virus strain and dose. |

(Continued)

**Table 1.** (Continued)

| Reference (Article Type) | Study Objectives | Mosquitoes/Virus/Temperature | Results | Quality Assessment (QAS) |
|---|---|---|---|---|
| [74] Gomard Y et al. Parasit Vectors 2020;13: 392. https://doi.org/10.1186/s13071-020-04267-z Contrasted transmission efficiency of Zika virus strains by mosquito species *Aedes aegypti*, *Aedes albopictus* and *Culex quinquefasciatus* from Reunion Island. (*Research Article*) | VC: #inf./#tested (IR) #dissem/#inf. (SDR) #transm./#dissem. (STR) Sites with active ZIKV transmission. | *Ae. albopictus* (2 strains, F1) *Ae. aegypti* (1 strain- F27) *Cx. quinquefasciatus* (2 strains F0, F1) Dak84 ($10^{7.4}$ PFU/ml) PaRi_2015 ($10^{5.8}$ PFU/ml) MAS66 ($10^{6.9}$ PFU/ml) 26±1°C, 80% RH 12 h: 12 h L:D | For the two *Cx. quinquefasciatus* strain after challenge with all three virus strains, this species was found to be completely refractory, with no virus detected in bodies, heads, or saliva by plaque assay at either 14 or 21 dpi. For *Ae. albopictus*, one specimen out of 64 individuals from the location Sainte-Marie was able to be infected, to disseminate and to transmit MAS66, but was refractory to PaRi_2105. The *Ae. aegypti* strain also showed a low susceptibility to MAS66 as infectious viral particles were only detected in the body of 1/32 individuals at 14 dpi, but a limited number of specimens had disseminated PaRi_2015 strain. The highest VC parameters were observed for the African ZIKV strain Dak84 from *Ae. albopictus* and *Ae. aegypti*. | QAS: ≥ 90% (TS = 34, MS = 22) Strengths: Infectious assay to detect ZIKV. Field collected mosquitoes. Multiple virus strains. Weakness: Experimental metadata difficult to extract. |
| [75] Li C-X et al. PloS NTD 2020; 14: e0008450. https://doi.org/10.1371/journal.pntd.0008450 Susceptibility of *Armigeres subalbatus* Coquillett (Diptera: Culicidae) to Zika virus through oral and urine infection. (*Research Article*) | VC: #inf./#tested (IR) #dissem/#tested (CDR) #SG+/#tested (CSGR) #transm./#tested (CTR) Transmission to neonatal mice and urine infection in larvae | *Ar. subalbatus* Coquillett (Colony) SZ01 (KU866423) 6.5 log$_{10}$ PFU/ml 26±1°C, 75±5% RH 14 h: 10 h L:D | Midgut, salivary gland, ovary, and saliva tested by PCR at 4, 7, 10 dpi. At 4 dpi, IR for midgut (53% [8/15]), salivary gland (13% [2/15]), ovary (7% [1/15]) and saliva (7% [1/15]. At 7 dpi, IR for midgut (32% [8/25]), salivary gland (16% [4/15]). ovary (12% [3/25]) and saliva (12% [3/25]. At 10 dpi IR for midgut (36% [9/25]), salivary gland (28% [7/25]), ovary (16% [4/25]) and saliva (8% [2/25]. 90% of the infant mice had viral RNA in their brain after being bitten by infectious mosquitoes. ZIKV RNA was detected in 100% of the 4$^{th}$ instar larvae, but not confirmed by infectious assay. After the mosquitoes emerged as adults, the quantitative PCR results were all negative. Similarly, infectious virus was not detected from the saliva of any adult. | QAS: < 25% (TS = 24, MS = 14) Strengths: Transmission to neonatal mice is convincing. Weakness: Used infectious assays retrospectively indicating much of results may be residual RNA and not represent infectious virus. VC parameters not properly defined, and dpi range is short. |
| [13] MacLeod HJ, Dimopoulos G. Mbio. 2020;11. Doi:10.1128/mBio.01765-20 Detailed analyses of Zika virus tropism in *Culex quinquefasciatus* reveal systemic refractoriness. (*Research Article*) | VC: #inf./#tested (IR) #transm./#tested (CTR) IT Retest HAI strain previously found to transmit ZIKV. Invitro studies | *Ae. aegypti* (Rockefeller) *Cx. quinquefasciatus* JHB (South Africa) HAI (China) FSS13025 (Cambodia, pre-epidemic) 6.9 log$_{10}$ and 9.3 log$_{10}$ PFU/ml (oral); 5.5 log$_{10}$ PFU/ml (IT) 27°C, 80% RH 12 h: 12 h L:D | Midguts tested at 7 dpi, salivary glands, and saliva at 14 dpi. No positive midgut samples at 7 dpi in *Cx. quinquefasciatus* JHB or HAI strains, but 11% of JHB and 23% of HAI strain midguts had indeterminant titers with ZIKV-Cambodia, which likely did not reflect infectious virus. No positive salivary glands for either of the *Cx. quinquefasciatus* strains, but 12% of JHB and 51% of HAI salivary gland samples were of indeterminate infectious status. Intrathoracically: 8% of JHB salivary glands were positive and no saliva samples were positive for infectious ZIKV by titer; 44% were negative, and 56% were of indeterminate status. Evidence suggest virus can enter midgut but not replicate but that other barriers exist in *Cx. quinquefasciatus*. | QAS: ≤ 50% (TS = 30, MS = 17) Strength: Study focused on specific mechanisms to identify the mechanisms resulting incompetence of *Cx. quinquefasciatus*. Focused on possible mechanisms for previous evidence for VC of species. Weakness: Not a natural system. |

(*Continued*)

**Table 1.** (Continued)

| Reference (Article Type) | Study Objectives | Mosquitoes/Virus/ Temperature | Results | Quality Assessment (QAS) |
|---|---|---|---|---|
| [76] **Uchida L et al. Pathogens 2021;10:938** https://doi.org/10.3390/ pathogens10080938 Zika virus potential vectors among *Aedes* Mosquitoes from Hokkaido, Northern Japan: Implications for potential emergence of Zika disease (*Research article*) | VC: #inf./#tested (IR) #dissem/#inf. (SDR) Temperate species (Japan) Entomology survey | *Ae. japonicus, Ae. punctor, Ae. galloisi* PRV ABC59 (KU501215) 5 $\log_{10}$ FFU/ml 6 $\log_{10}$ FFU/ml 26±2°C, 65–85% RH 12 h: 12 h L:D | No evidence of infection or dissemination of ZIKV was found for *Ae. punctor*. For *Ae. galloisi*, 71% (5/7) of tested abdomens contained ZIKV RNA at 5–10 dpi, but by FFA IR was 25% (1/4) at 10 dpi, the latter specimen's head and thorax was also positive, however titers were low. For *Ae. japonicus*, ZIKV RNA was detected in 22% of abdomens at 0 and 10 dpi. IR at 10 dpi was 10% (1/10) but no additional evidence of dissemination or transmission was observed. | QAS: < 25% (TS = 21, MS = 11) Strengths: Field collected mosquitoes. Weakness: No transmission assessment. Low sample sizes and cutoffs for assay not clear. Data in supplementary tables inconsistent with text. Poor study design. |
| [77] **Zimler RA et al. J Med Entomol. 2021; 58: 1405–1411.** *Doi:* 10.1093/jme/tjaa286 Transmission potential of Zika virus by *Aedes aegypti* (Diptera: Culicidae) and *Ae. mediovittatus* (Diptera: Culicidae) populations from Puerto Rico. (*Research article*) | VC: #inf./#tested (IR) #dissem/#tested (CDR) #transm./#tested (CTR) | *Ae. aegypti* (F2) *Ae. mediovittatus* (F1) PRV ABC59 (KU501215.1) 7 log10 PFU/ml, 6 log10 PFU/ml 5 log10 PFU/ml 4 log10 PFU/ml 28±1°C, 80% RH 15 h: 9 h L:D | Mosquitoes were tested at 15 dpi. *Aedes aegypti* females were approximately twice as susceptible to Zika infection as *Ae. mediovittatus* and infection rates increased with viral dose. IRs ranged 70 to 85% at (7$\log_{10}$ PFU/ml) compared to 0–10% for 4 $\log_{10}$ PFU/ml). *Aedes aegypti* had 5-fold higher disseminated infection than *Ae. mediovittatus*. Significant differences were observed between *Ae. aegypti* and *Ae. mediovittatus* at 6 and 7 $\log_{10}$ PFU/ml, with rates of disseminated infection being 5-fold to 22-fold higher for *Ae. aegypti* than *Ae. mediovittatus*. Only two saliva samples were available for *Ae. mediovittatus*, and not analyzed. Saliva infection was 14.81% at 7 $\log_{10}$ PFU/ml. The remaining lower viral doses had 0% saliva infection for *Aedes. aegypti.* | QAS: ≥ 75% (TS = 33, MS = 20) Strengths: Field collected mosquitoes., multiple viral doses. Weakness: No infectious assay (PCR only). Only 15 dpi. Difficult to extract experimental metadata. |

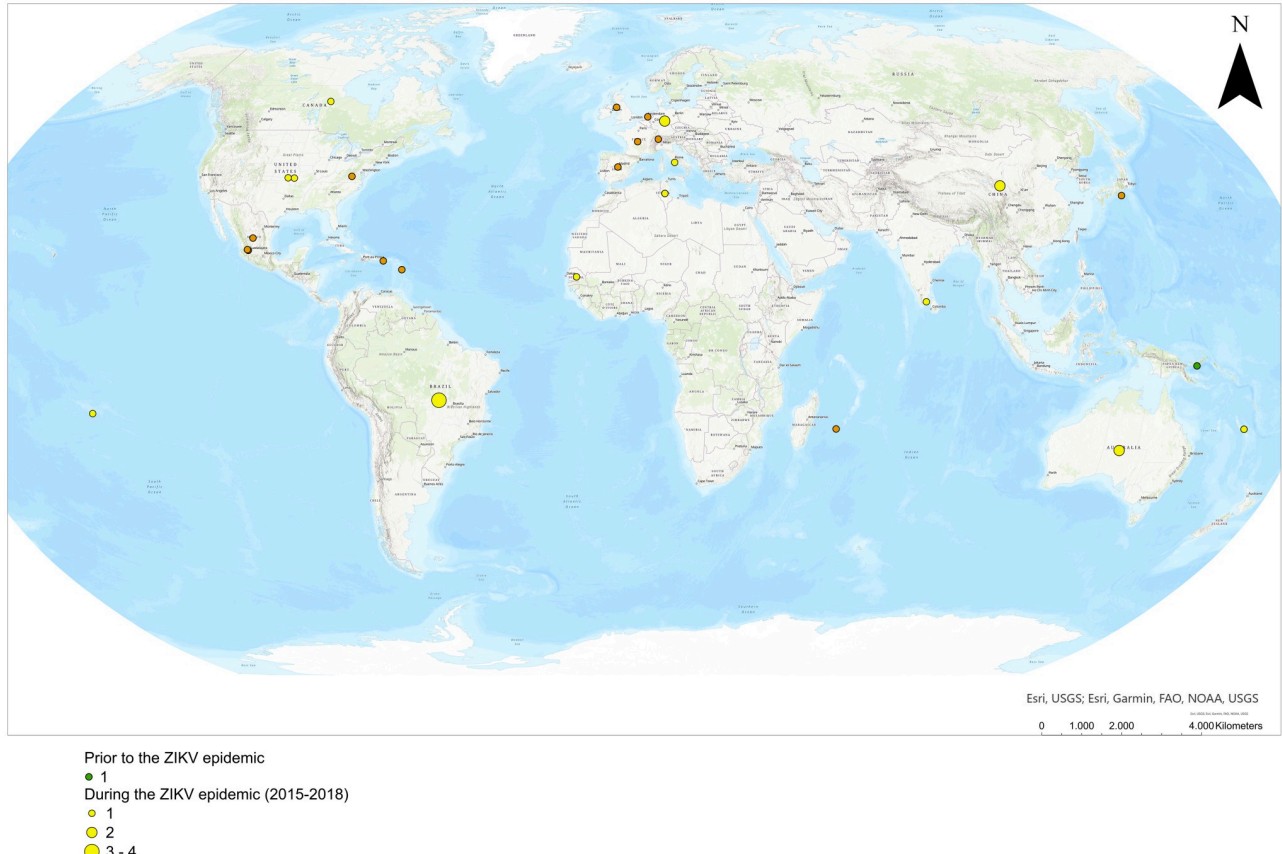

**Fig 2. Geographic and temporal distribution of 45 vector competence studies of mosquito species other than** *Aedes aegypti* **and** *Aedes albopictus* **for Zika virus transmission included from an electronic search finalized on 15 March 2022 [78].** Figure created in ARCGIS using the following map which the information in the links claim is open source. https://www.arcgis.com/home/item.html?id=30e5fe3149c34df1ba922e6f5bbf808f https://services.arcgisonline.com/ArcGIS/rest/services/World_Topo_Map/MapServer/0 https://www.arcgis.com/apps/mapviewer/index.html?layers=30e5fe3149c34df1ba922e6f5bbf808f.

Four of five of the lowest scores were from papers where no infectious assay was used. Interestingly, of the seven publications in short format (letters, dispatches, short communications), five (71%) scored in the lowest 25%, in large part because important metadata was excluded.

## Description of the included studies

**Time and geographical clustering of studies.** The included papers could be categorized in three-time periods representing the period prior to, during, and after the 2015–2018 ZIKV epidemic in the western hemisphere. Among our selected articles, one was published before 2015, 27 were published between 2015 and 2018, and 17 published after 2018 (Fig 2). The earliest study was on *Ae. hensilli*, the most abundant *Aedes* species found on the island of Yap, in the Federated States of Micronesia, the location of the only described ZIKV outbreak reported prior to 2015, prompting interest in the vector capacity of this species [37].

Among the papers published during the 2015–2018 ZIKV epidemic, the majority (21 of 27) were laboratory studies principally from the USA and Europe that examined 1) laboratory colony mosquitoes that were rapidly available for VC studies; or 2) assessed local mosquito

species in those areas that could represent risk of transmission to populations in those regions. For example, laboratory colonies of *An. gambiae*, *An. stephensi*, and *Cx. quinquefasciatus* were tested for competence for ZIKV [58]. Although the laboratory methodology was sound, they did not include an *Ae. aegypti* comparator, and this study is illustrative of the rush to publish this type of work at the time. There were also articles from Brazil [31] and China [30] that argued the role of *Cx. quinquefasciatus* as a possible secondary vector of Zika virus, especially in Brazil, where its role in the ongoing and devastating outbreak was of great public health concern.

Starting in 2019, the rational for published studies continued with the evaluation of local species [62,65,67,68,70,71,73,76,77] but also expanded to examining the role of sylvatic species [64], clarify the role of *Cx. quinquefasciatus* [13,63,66,72,74], and ask more specific questions about the mechanisms of ZIKV transmission within their respective vectors [69,71].

Of the total number of the articles included in this review, 38% (17 articles) were focused in North and Central America, and the Caribbean, (Panama, Puerto Rico, Mexico, Guadeloupe, Canada and USA), 10 papers focused on Europe (Switzerland-France, Netherlands, Spain, Germany, Italy, United Kingdom and Reunion Islands), five on Asia (China and Japan), four on Oceania (French Polynesia- New Caledonia—Samoa-Wallis and Futuna, Papua New Guinea, Australia and French Polynesia), four on South America (Brazil), two on Africa (Tunisia and Senegal), and one was not identified. Two papers tested mosquitoes from different continents: one from Oceania (New Caledonia) and South America (Brazil) and another from North America (USA) and Africa (Tunisia).

## Mosquito species tested for vector competence

Table 2 summarizes the mosquito species undergoing VC assays for ZIKV. A total of 27 *Aedes* species other than *Ae. aegypti* and *Ae. albopictus* were studied. Seven, two, and one species from the genus *Culex*, *Anopheles*, and *Coquillettidia* were studied, respectively. Finally, Li et al. [75] examined the vector competence of *Armigeres subalbatus* as a mosquito species associated with human waste pools and potential infection through exposure to urine. *Culex quinquefasciatus* was the focus of almost half of VC examining species other than *Ae. aegypti and Ae. albopictus* (22 articles). Other species studies by multiple authors were *Cx. pipiens* (9 articles), *Ae. japonicus* (5 articles), and *Ae. vexans*, *Ae. caspius*, and *Cx. tarsalis* (3 articles each), and *Ae. triseriatus*, *Ae. polynesiensis*, and *Ae. notoscriptus* (2 articles each).

## Mosquitoes used in vector competence experiments

Mosquitoes were infected orally with an artificial feeder in 41 of the 45 studies, whereas four articles used interferon deficient mice infected with ZIKV to infect *Cx. pipiens* and *Ae. triseriatus* [39], *Cx. quinquefasciatus* [51,61], *Cx. tarsalis* [61] and *Sabethes cyaneus* [60]. The dose used to infect mosquitoes ranged from 4 to 8 $\log_{10}$ PFU/ml. Vector competence studies are best done with field derived material rather than strains that have been colonized for many generations. Twenty-four studies included mosquitoes captured from field populations recently from generations $F_1$-$F_3$ and three studies from $F_4$-$F_9$. Of these, 10 studies also tested some colonized strains. Eighteen studies only used mosquitoes colonized for more than 10 generations and up to 50 years. In most cases, readers were able to infer or distinguish between if colonized or field derived mosquitoes were used, but many articles failed to clearly indicate the generations the mosquitoes had been in the laboratory.

**Table 2. Summary of species studied country of origin.**

| Continent | Country | Species (no of reference) |
|---|---|---|
| Africa (n = 4) | Senegal | *(Aedes unilineatus, Ae. vittatus, Ae. luteocephalus)* [38] |
| | Tunisia | *(Ae. caspius, Ae.detritus)* [62]; *(Culex quiquefaciatus, Cx. pipiens)* [40] |
| | Reunion Islands* | *Cx. quinquefasciatus* [74] |
| Asia (n = 5) | China | *Cx.* quinquefasciatus [13,30,54]; *Armigeres subalbatus Coquillett* [75] |
| | Japan | *(Ae. japonicus, Ae. punctor, Ae. galloisi)* [76] |
| Europe (n = 9) | Italy | *Cx. pipiens* [41] |
| | Germany | *(Cx. pipiens, Cx. torrentium)* [52]; *Ae. japonicus* [59] |
| | Spain | *Ae. caspius* [65,67] |
| | Netherlands | *Cx. pipiens* [68,69] |
| | Switzerland/ France | *Ae. japonicus* [73] |
| | UK | *Ae. detritus* [70] (reported as *Ochlerotatus detritus*) |
| North- Central America and the Caribbean (n = 17) | USA | *Cx. quinquefasciatus* [32,40,44,46,51,53,58,61]; *Cx. pipiens* [39,40,44,46,53]; *Cx. tarsalis* [46,61]; *Ae. triseriatus* [39,71]; *Ae. vexans* [50,56]; *Ae. taeniorhynchus* [51]; *Ae. japonicus* [71]; *(Anopheles gambiae, An. stephensi)* [58] |
| | Canada | *(Ae. cinereus, Ae. euedes, Ae. fitchii, Ae. sticticus, Ae. vexans, Coquillettidia perturbans, Cx. restuans, Cx. tarsalis)* [47] |
| | Mexico | *Cx. quinquefasciatus* [63] |
| | Panama | *Sabethes cyaneus* [60] |
| | Guadeloupe | *Cx. quinquefasciatus* [66] |
| | Puerto Rico | *Ae. mediovittatus* [77] |
| Oceania (n = 6) | Papua New Guinea | *Ae. hensilli* [37] |
| | French Polynesia | *Ae. polynesiensis* [45,57] |
| | Australia | *Ae. vigilax* [43]; *Ae. procax* [43]; *Ae. notoscriptus* [43,48]; *Ae. camptorhynchus* [48]; *Cx. annulirostris* [43,48]; *Cx. sitiens* [43]; *Cx. quinquefasciatus* [43,48] |
| | New Caledonia | *Ae. polynesiensis* [57]; *Cx. quinquefasciatus* [72] |
| | Samoa, Wallis, Futuna | *Ae. polynesiensis* [57] |
| South America (n = 5) | Brazil | *Cx.. quinquefasciatus* [31,42,56,72],; *(Ae. terrens, Ae. scapulatus, Haemagogus leucocelaenus, Sa. identitus)* [64] |

*Administratively belongs to France, and not part of WHO AFRO Member states but geographically in the Africa Region.

## Virus strains used

Studies using well described viral strains, including strain names, source and passage history were considered of high scientific quality. Study design elements including testing >1 strain, use of low passage strains (≤ 20 passages), and inclusion of strains from epidemiologically relevant endemic regions of the world were all items in our quality grading tool. Among the 45 articles included in our systematic review, we identified 27 ZIKV strains used in experiments (Table 3). The most used strain was PRVABC59 (GenBank accession numbers KU501215 or KX601168) isolated in 2015 used in 14 of the studies, followed by MR766 the prototype strain originally isolated from a sentinel monkey during YF surveillance studies in 1947 was included in seven studies. Virus strains used by researchers appear driven mostly by access, but details provided by investigators was not standardized. For example, not all investigators included GenBank accession numbers or detailed information on passage histories. It was clear, however, in 30 of 45 articles (67%) if the virus used had a low or high passage history, and 26 of 45

**Table 3. Virus strains used in 45 vector competence studies examining secondary mosquito vectors for Zika virus.** GenBank accession numbers were not included unless specified in original article.

| Virus Strain | GenBank No.[#] (reference) | Country of origin | Year of isolation | Host | References |
|---|---|---|---|---|---|
| MR766 (prototype) | AY632535 [46] | Uganda | 1947 | Sentinel Monkey | [37,38,43,46,53,58,67] |
| MAS66<br>MYS/P6-740 | KX694533 [66,74]<br>KX601167 [61] | Malaysia | 1966 | *Aedes aegypti* | [61,66,72,74] |
| DAK41525 | KU955591 | Senegal | 1984 | *Ae. africanus* | [46,51] |
| DAK84 | KU955592 [66,73] | Senegal | 1984 | *Ae. taylori* | [66,72–74] |
| ArD128000 | | Senegal | 1997 | *Ae. luteocephalus* | [38] |
| ArD132912 | | Senegal | 1998 | *Ae. dalzieli* | [38] |
| ArD157995 | | Senegal | 2001 | *Ae. dalzieli* | [38] |
| ArD165522 | | Senegal | 2002 | *Ae. vittatus* | [38] |
| HD78788 | | Senegal | 1991 | Human patient | [38] |
| Cambodia 2010<br>FSS13025 | KU955593 [48]<br>JN860885 [13,65] | Cambodia | 2010 | Human patient | [13,48,51,65] [48] |
| H/PF13<br>PF13/251013-18 | KX369547<br>KJ776791 [41] | French Polynesia | 2013 | Human patient | [41,45] |
| PLCal_ZV | KF993678 | Thailand* | 2013 | Human patient | [47] |
| NC-2014-5132 | SRR5309452 [57] | New Caledonia | 2014 | Human patient | [40,55,57,62] |
| MRS_OPY_Martinique_PaRi_2015 | KU647676 [66,74] | Martinique | 2015 | Human patient | [66,72,74] |
| MEX1-7 | KX247632 [60] | Mexico | 2015 | *Ae. aegypti* | [51,60] |
| MEX1-44 | | Mexico | 2015 | *Ae. aegypti* | [51] |
| Pariaba_01 | KX280026 | Pariaba, Brazil | 2015 | Human patient | [13] |
| PRVABC59 | KU501215 [32,39,46,47,50,65,71,76,77]<br>KX601168 [61] | Puerto Rico | 2015 | Human patient | [32,39,46,47,50,51,53,58]<br>[44,61,65,71,76,77] |
| PE243 | KX197192 [31] | Recife, Brazil | 2015 | Human patient | [31,49,70] |
| SPH2015 | KU321639 | Sao Paulo, Brazil | 2015 | Human patient | [13,49,61] |
| SZ01 | KU866423 | China | 2016 | Human patient*** | [30] |
| ZJO3 | KU820899 | China | 2016 | Human patient | [54] |
| FB-GWUH | KU870645 | Guatemala** | 2016 | Fetal brain | [52,59] |
| HND/R103451/2015 | KX694534 | Honduras | 2016 | Placenta**** | [53] |
| Unspecified Puerto Rico Strain | | Puerto Rico | 2016 | Human patient | [56] |
| Rio-S1 | KU926310 | Rio de Janeiro, Brazil | 2016 | Human patient | [42,64] |
| Rio-U1 | KU926309 [42,64] | Rio de Janeiro, Brazil | 2016 | Human patient | [42,49,64] |
| EVAg Ref-SKU 011V-01621 | KU937936 [68,69] | Suriname | 2016 | Human patient | [67–69] |

[#]GenBank Accession numbers were not reported by all authors, we include the references when reported.

*Individual with travel history to Thailand;

**Individual with travel history to Guatemala;

***Individual with travel history to American Samoa;

****Individual with travel to Honduras in 2015.

studies (58%) included a low passage virus in their experiments. Of the articles providing passage numbers, for most specific values that were generally < 10 passages for "low" passage, apart from one research group in Senegal reporting the use of 20 passages for strain MR766 [38]. This latter prototype strain was reported to have between 146–150 passages for most authors who used it in their experiments [37,43,46,53,78]. No recombinant viruses were used for experiments in the studies reviewed. Additionally, 56% (25 of 45) included clear descriptions of both virus and mosquito strains used.

## Vector competence parameters

The principal objective of the 45 studies evaluated was a straightforward evaluation of the ability of one or more mosquito species to become infected, disseminate, and transmit ZIKV at different viral titers and rearing temperatures. There were only a few exceptions where the authors objectives were focused on understanding the biology behind VC patterns observed [13,55]. Although, the main VC parameters (infection, dissemination, and transmission) were universally recognized, terminology used to describe these parameters was variable and more alarming, inconsistent; the terms dissemination rate (DR) and transmission rate (TR) had distinct definitions depending on the article. Fortunately, most authors provided definitions for the terms used, but unless readers are meticulous, mistakes in interpretation are inevitable. All 45 articles estimated infection rates either by testing mosquito bodies (usually the abdomen/thorax) or midguts for Zika virus or RNA. The denominator used was the total number mosquitoes tested which represented mosquitoes that were blood fed and held for 7–21 days. Dissemination rates were estimated in 37 of the 45 studies through testing heads, wings, legs, and in one case salivary glands and another ovaries/exoskeleton. Either the number of infected mosquitoes (stepwise approach), or alternatively, all mosquitoes tested (cumulative approach) were used in the denominator. The most relevant parameter, transmission was estimated in 39 of the articles. Two papers used salivary glands as a surrogate for virus transmission [31,54] whereas 37 articles tested saliva. In two articles, infected mosquitoes were observed to transmit Zika virus to neonatal mice [30,75]. As with dissemination, the denominator used to estimate transmission was the number of mosquitoes with disseminated infections (stepwise) or all the total number of mosquitoes tested (cumulative). Some authors use the term transmission efficiency and more recently dissemination efficiency [79] to make the distinction between stepwise and cumulative rates. Articles using the term "efficiency" distinguish between stepwise rates (e.g., dissemination rate [DR], transmission rate [TR]) from cumulative rates (e.g., dissemination efficiency [DE], transmission efficiency [TE]). The major cause of confusion is that among the 45 articles in our review, 22 used DR and TR to signify cumulative rates, whereas 17 articles used the same terminology, "DR" and "TR" to describe stepwise rates while sometimes also presenting cumulative rates using the terms "DE" and "TE". Other authors avoided these terms and used their own well-defined definitions. Other terms used are dissemination infection rate (DIR) and transmission infection rate (TR[D]) to contrast these two different VC parameters.

## Laboratory assays used to detect ZIKV or ZIKV RNA

Protocols employed to test mosquitoes for ZIKV or RNA varied across studies, but there is a clear distinction between studies that only used PCR to detect RNA as opposed to the more labor intensive and technical assay that directly measured infectious virus (Table 4). About 43% of the studies only used PCR assays to measure infection and dissemination, but for transmission 29% of the studies relied solely on PCR. For the identification of the transmission

**Table 4. Laboratory Assays used to detect Zika virus or RNA for infection, dissemination, and transmission assays.** Number of studies (%).

| Assay | Description | Infection (n = 45) | Dissemination (n = 40) | Transmission (n = 41) |
|---|---|---|---|---|
| PCR | Both RT and qRT PCR | 10 (22%) | 17 (42.5%) | 12 (29%) |
| Infectious virus | Plaque, TCID$_{50}$ or Fluorescent Focus assay using Vero or C6/36 cells, | 16 (36%) | 15 (37.5%) | 20 (49%) |
| Combination | Varied protocols, some screening with infectious assay followed by titration with qPCR or PCR screen and confirmation of all or subset of samples by infection assay. | 19 (42%) | 8 (20.0%) | 9 (22%) |

status of the tested mosquitoes a variation of laboratory assays was used. About 20% of the studies used a combined strategy of screening by either PCR or an infectious assay, followed by confirmation or titration with the other type of assay.

## Studies using mice

In six studies, mice were utilized (either neonatal or genetically modified) to infect mosquitoes in a more natural and efficient manner or to confirm transmission of virus from the mosquito to a new host. Vector competence parameters were higher for mosquitoes fed on mice than those using artificial feeding strategies or direct testing of saliva, consistent with the observation that blood from viremic animals are typically more infectious for mosquitoes than artificial meals [80,81]. Four studies used interferon deficient mice infected with ZIKV to infect *Cx. pipiens* and *Ae. triseriatus* [39], *Cx. quinquefasciatus* [51], *Cx. tarsalis* [61] and *Sa. cyaneus* [60] mosquitoes experimentally, compared to *Ae. aegypti*. All species infected using mice were unable to transmit ZIKV, except for *Sa. spp.*, where a single mosquito (11%) that fed on a mouse with 6.8 $\log_{10}$ PFU/ml after being held for 21 days tested positive for ZIKV by plaque assay, compared to 70% of *Ae. aegypti*.

Two studies evaluated transmission in parallel with saliva testing for *Cx. quinquefasciatus* [30] and *Ar. subalbatus* [75]. The former study continues to be cited as the only credible evidence for the competence of *Cx. quinquefasciatus* for ZIKV transmission. Ten days after being fed on by orally infected mosquitoes, the brains of eight of nine neonatal mice tested positive for ZIKV RNA. Similarly, 8-day post exposure 80% of 10 mosquitoes had ZIKV RNA positive saliva, but only 1 of 3 mosquitoes >12 d post exposure had RNA positive saliva. Biologically, we would expect that once saliva tests positive, it would remain so; low sample numbers made clear evaluation difficult, but the infection of the neonatal mice is difficult to interpret as anything other than evidence of virus transmission. Using almost identical methodology for *Ar. subalbatus*, saliva tested positive for ZIKV RNA in 7–12% of the mosquitoes exposed and nine of ten of the neonatal mouse brains tested positive for RNA. Although both studies presented virus titers observed in the saliva samples, these were generated from standard curves developed in the laboratory, not based on the mosquito samples themselves. In both studies, the failure to use and infectious virus assay to test for ZIKV prevented unequivocal conclusions about each species as a vector.

## Review of evidence for individual vector species other than *Ae. aegypti* and *Ae. albopictus*

Existing evidence for VC of vector species other than *Ae. aegypti* and *Ae. albopictus* for ZIKV can be divided into the following relevant epidemiological contexts: 1) African [38] and South American [60,64] forests cycles; 2) local vector species from areas directly affected by ZIKV transmission generally associated with Micronesia/Polynesia in island settings [37,45,57]; 3) local vector species in normally temperate areas where risk of ZIKV transmission was evaluated for outbreak readiness which included species in Australia [43,48], Mediterranean region [62,65,67], Europe [40,41,52,68,69], Canada [47], USA [39,50,51,56,61,71,77] and the examination of one cosmopolitan species *Ae. japonicus* [59,76]; and 4) in settings similar to endemic DENV/CHIKV transmission where *Cx.* species are potentially involved. The latter category resulted a considerable focus on *Cx.* species in areas with and without ZIKV transmission. Table 5 summarizes the existing evidence, by species and the four categories above, of VC by mosquito species.

**Table 5. Vector competence (VC) parameters of species investigated as possible vectors of Zika virus.** Infection Rate (IR, [#inf./#tested]) is defined as the percentage of mosquitoes contain virus in bodies or midguts (number positive/number tested). Terminology used for dissemination and transmission are expressed as cumulative (C) or stepwise (S) rates depending on the experimental design. For dissemination, we use cumulative dissemination rate (CDR, [#dissem./#tested]) or stepwise dissemination rate (SDR, [#dissem./#inf]), defined as the percentage of mosquitoes containing virus in head, legs+wings, or salivary glands/ovaries (number positive/number of engorged mosquitos tested for infection [CDR] or number positive /number of infected mosquitoes [SDR]). For transmission, we use cumulative transmission rate (CTR, [#transm./#tested]) or stepwise transmission rate (STR, [#transm./#dissem.]), defined as the number of mosquitos with virus in saliva or transmitting ZIKV to a mouse (number positive/number of engorged mosquitoes tested [CTR] or number positive/number disseminated infections [STR]). NT = not tested. Raw data presented in S1 Data.

| Type | Species | IR | SDR | CDR | STR | CTR |
|---|---|---|---|---|---|---|
| Forest Africa | *Ae. unilineatus* (5 ZIKV strains) [38] | 19% | 5% | 1% | 0% | 0% |
| | *Ae. vittatus* (5 ZIKV strains) [38] | 14% | 27% | 4% | 20% | 1% |
| | *Ae. luteocephalus* (5 ZIKV strains) [38] | 75% | 42% | 32% | 50% | ~17%[a] |
| Forest South America | *Ae. terrens* (21 dpi) [64] | 0% | 0% | 0% | 0% | 0% |
| | *Ae. scapularis* [64] | 2% | 0% | 0% | 0% | 0% |
| | *Sa. identicus* [64] | 0% | 0% | 0% | 0% | 0% |
| | *Sa. albiriprivus* (2 strains) [64] | 1% | 0% | 0% | 0% | 0% |
| | *Hg. leucocelaenus* (7,14,21 dpi) [64] | 27% | 11% | 3% | 0% | 0% |
| | *Sa. cyaneus* (21 dpi) [60] | 1% | 100% | 1% | 100% | 1% |
| Local Polynesia | *Ae. polynesiensis* (2 studies) [45,57] | 36;87% | 50;45% | 18;39% | 0;3% | 0;1%[b] |
| | *Ae. henselli* (4.9-5.9 $\log_{10}$ PFU/ml) [37] | 69% | 69% | 13% | NT | NT |
| Puerto Rico | *Ae. mediovittatus* (4-7 $\log_{10}$ PFU/ml) [77] | 29%[c] | 2% | 1%[c] | NT | NT |
| Australia | *Ae. camptorhynchus* [48] | 28% | 14% | NT[d] | 14% | <1%[d] |
| | *Ae. notoscriptus* (2 studies) [43] | 57% | 21% | 12% | 0% | 0% |
| | [48] | 34% | 3% | 1% | 50% | <1%[d] |
| | *Ae. vigilax* [43] | 57% | 47% | 27% | 0% | 0% |
| | *Ae. procax* [43] | 33% | 50% | 17% | 0% | 0% |
| China | *Armigeres subalbatus* (10 dpi) [75] | 36% | 78% | 28%(SR) | 29% | 8%[e] |
| Local | *Refractory species* | 0% | 0% | 0% | 0% | 0% |
| | *Ae. cinereus* [47], *Ae. punctor* [76], *Ae. galloisi* [76], *Cx. annulirostris* [43,48], *Cx. sitiens* [43], *An. gambiae* [58], *An. stephensi* [58] | | | | | |
| Widespread Temperate | *Ae. vexans* (3 studies) | | | | | |
| | $10^{5.8}$ PFU/ml, 25°C [47] | 13% | 47% | 6% | NT | NT |
| | $10^{6.8-7.2}$ PFU/ml, 27°C [50] | 80% | 20% | 16% | 29% | 5% |
| | $10^{5.3}$ PFU/ml, 28°C [56] | 28% | 11% | 3% | 5%[f] | 1%[f] |
| | *Ae. triseriatus* (2 studies) [39,71] | 12;25% | 0% | 0% | 0% | 0% |
| | *Ae. taeniorhynchus* ($10^{6.8}$ PFU/ml)[g] [51] | 0% | 0% | 0% | 0% | 0% |
| | *Ae. caspius* (3 studies) | | | | | |
| | (PCR only) [62, 65] | 10;40% | 0% | 0% | 0% | 0% |
| | (Plaque assay) [67] | 0% | 0% | 0% | 0% | 0% |
| | *Ae. detritus* (2 studies) | | | | | |
| | $10^{7.2}$ PFU/ml, 14 dpi [62] | 75% | 0% | 0% | 0% | 0% |
| | $10^{6}$ PFU/ml, 14-28 dpi, 19°C [70] | 14% | NT | NT | NT | 3% |
| | $10^{6}$ PFU/ml, 14-28 dpi, 21°C [70] | 24% | NT | NT | NT | 9% |
| | $10^{6}$ PFU/ml, 14-28 dpi, 24°C [70] | 17% | NT | NT | NT | 10% |
| | $10^{6}$ PFU/ml, 14-28 dpi, 28°C [70] | 15% | NT | NT | NT | 13% |
| | $10^{6}$ PFU/ml, 14 dpi, 31°C [70] | 15% | NT | NT | NT | 15% |
| | *Ae. japonicus* (5 studies) [76] | 10% | NT | NT | NT | NT |
| | $10^{6.5-7.7}$ PFU/ml, 24 °C [71] | 20% | 47% | 9% | 29% | 3% |
| | $10^{7.2}$ TCID$_{50}$/ml, 28 °C [68] | 10% | 83% | 8% | 40% | 3% |
| | 14,21 dpi, constant 27°C [73] | 87% | 22% | 19% | 57% | 11% |
| | 14,21 dpi, fluctuating 21-27°C [73] | 50% | 48% | 24% | 41% | 10% |
| | 14 dpi, 21°C [59] | 10% | NT | 0% | 0% | 0% |
| | 14 dpi, 24°C [59] | 24% | NT | ~26% | 0% | 0% |
| | 14 dpi, 27°C [59] | 67% | NT | ~13% | 14% | 10% |
| | *Cx. pipiens* [40, 41,44,46,71] | 0% | 0% | 0% | 0% | 0% |
| | 18-27°C, 14,21 dpi [52] | 16% | NT | NT | 0% | 0% |
| | $10^{4-7.1}$ PFU/ml, 28°C [53] | 9% | 0% | 0% | 0% | 0% |
| | $10^{7}$ TCID$_{50}$/ml, 28°C [69] | 47% | NT | NT | NT | 3% |
| | *Cx. molestus* [69] | 0% | 0% | 0% | 0% | 0% |
| | 18 and 27°C, 14,21 dpi [52] | 24% | NT | NT | 0% | 0% |
| | *Cx.. tarsalis* [46,47] | 0% | 0% | 0% | 0% | 0% |
| | 14,21 dpi, 26°C [61] | 12% | 38% | 5% | 0% | 0% |
| | *Cx. torrentium* 14,21dpi 18°,27°C [52] | 11% | NT | NT | 0% | 0% |

(*Continued*)

**Table 5.** (Continued)

| Type | Species | IR | SDR | CDR | STR | CTR |
|---|---|---|---|---|---|---|
| Widespread Temperate | *Refractory species*<br>*Ae. euedes* [47], *Ae. fitchii* [47], *Ae. strictus* [47],<br>*Cx. restuans* [47,71], *Coquillettidia perturbans* [47] | 0% | 0% | 0% | 0% | 0% |
| Worldwide | *Cx. quinquefasciatus* | 0-2%[h] | 0%[h] | 0%[h] | 0%[h] | 0%[h] |

[a]TR was reported as 50% representing denominator of 19, but text states that 27 saliva specimens were tested.

[b]At 21 dpi CTR = 3%.

[c]Estimated IR from a logistical model at 4 titers (18%). The range for IR at lower titers was 0–75% increasing with dose. Dissemination was observed only at the highest titers between 4–5%.

[d]Dissemination measured as dissected positive carcasses including ovaries and exoskeleton over number mosquitoes exposed and transmission had distinct denominator from IR, dissemination, and transmission experiments. It is impossible to extract the number exposed and denominators are higher for dissemination and transmission experiments. Because denominators are independent, we present as SDR and STR.

[e]90% of neonatal mice that were fed on by infected mosquitoes had ZIKV RNA detected in their brains.

[f]34% transmission rate after IT inoculation, used to estimate TR/TE.

[g]Infected from Ifnar-/- mice with at 4.7 $\log_{10}$ and 6.8 $\log_{10}$ PFU/ml, only infected at higher titer.

[h]18/22 studies

## Forest cycles

Laboratory incrimination studies for African forest vectors was limited to a single study conducted in Senegal [38]. The only forest species with laboratory evidence incriminating them as a ZIKV vectors were *Ae. vittatus* and *Ae. luteocephalus*, both able transmit virus with CDR of 1 and 17%, respectively. The same study did not find *Ae. unilineatus* to be competent, and the author's pointed out the need for further studies on other important forest species that could be involved in transmission. For South American forest species, *Ae. terrens*, *Ae. scapularis*, *Sa. identicus* were completely refractory to infection, *Sa. albiprivus* had 0.5% infection rate, but *Haemagogus leucocelaenus* had infection rates of 14.8 to 40%, dissemination rates of 2.2 to 5% depending on day post infection but showed no evidence of transmission [64]. A laboratory study from a long-term colony of *Sa. cyaneus* had one mosquito with evidence of infection, dissemination, and transmission out of 69 tested.

## Vectors associated with Micronesia/Polynesia in Island settings

Evidence for *Ae. hensilli* and *Ae. polynesiensis* as vectors for ZIKV was evaluated because of their high abundance at the time of ZIKV outbreaks. Infection of *Ae. hensilli* was demonstrated but no evaluation of transmission (saliva or salivary gland testing) was conducted [37]. *Aedes polynesiensis* showed disseminated infections with ZIKV under laboratory conditions but did not transmit the virus [45]. Dissemination was far less efficient than observed for *Ae. aegypti*. In contrast, a later study showed two populations of *Ae. polynesiensis* from French Polynesia and Wallis and Futuna had high infection rates (71%) that were less variable than *Ae. aegypti* populations from the same region and showed high dissemination rates (SDR = 45%, CDR = 39% and low transmission rates (STR = 3%, CDR = 1%) [57].

## Local vectors not found to be competent to transmit Zika Virus

In Australia, there are many local *Aedes* species examined for their ability to transmit ZIKV [43]. *Ae. vigilax*, *Ae. procax*, and *Ae. notoscriptus* showed infection and dissemination rates consistent with *Ae. aegypti*, but all failed to transmit the virus. In contrast, a later study [48]

found some transmission (<1%), but lower infection and dissemination rates than the previous study for both *Ae. notoscriptus* and *Ae. camptorhynchus*.

In Canada, of the three *Culex* species examined, *Cx. sitiens* and *Cx. annulirostris* were completely refractory to infection and only 2 out of 30 *Cx. quinquefasciatus* were infected but none developed disseminated infections. Similarly, in Europe, no evidence was found for transmission in *Cx. pipiens pipiens* or *Cx. pipiens molestus* [52,69], although both species did accumulate virus in saliva after intrathoracic injection. North American species *Ae. triseriatus* [39,71] and *Ae. taeniorhynchus* [51] were both found to be refractory to ZIKV infection. *Ae. mediovittatus* from Puerto Rico was studied [77] and found to be half less susceptible to oral infection than *Ae. aegypti*, indicating a more effective midgut infection barrier in *Ae. mediovittatus*. Dissemination rates were < 5% compared to 40–95% in *Ae. aegypti*. Insufficient saliva samples were obtained to evaluate transmission for this species. A single study [58] found laboratory colonies of *An. gambiae* and *An. stephensi* to be refractory to ZIKV infection.

One study from China investigated *Ar. subalbatus*, a species of interest because it developed in human waste lagoons where ZIKV could be shed in urine. Average infection rate for this species was 43% (measured in midguts) and 90% of the infant mice that were bitten by infectious mosquitoes had viral RNA in their brain although was not detected in mosquito saliva. Furthermore, *Ar. subalbatus* larvae reared in water containing ZIKV and human urine did not result in any adult mosquitoes with detectable ZIKV RNA in saliva, providing no evidence of transmission via this route. *Ae. galloisi* and *Ae. punctor* were evaluated in Japan, showing very low rates of susceptibility to ZIKV infection but dissemination and transmission were not evaluated. These species provide a good example of the methodological difficulties associated with evaluation when blood feeding rates are low, and the numbers of infected mosquitoes are so low a true evaluation of dissemination and transmission is difficult.

## Species with evidence of very low transmission potential

Species with evidence of transmission, albeit at very low levels, were *Ae. vexans*, *Ae. detritus*, and *Ae. japonicus* (Table 5). Three studies indicated that, *Ae. vexans* is a competent vector for ZIKV [47,50,56] and while infection rates were high in some of these studies, all of them reported low dissemination and transmission rates. In Canada, of the 4 (13%) *Ae. vexans* positive for ZIKV RNA, 2 (50%) had ZIKV RNA detected in their legs for an overall dissemination rate of 6%. Twenty-three percent of *Ae. vexans* infected by intrathoracic injection had ZIKV RNA detected in their saliva. Evidence for *Ae. vexans* transmission was also found in Colorado, USA, with relatively high infection rates ranging from 66–91%, dissemination rates from 3–25%, but low transmission rates from 2–7%. Consistent with these finding was another study [56] from the great plains region that had high transmission rates in *Ae. vexans*, after intrathoracic infection, suggesting that the primary barrier in this species is midgut escape barrier. In the United Kingdom *Ae. detritus* (reported as *Ochlerotatus detritus* in original article) became infected and showed infectious virus in saliva at rates lower than *Ae. albopictus*, but saliva positivity increased at higher temperatures [70]. In the Mediterranean Region neither *Ae. detritus* nor *Ae. caspius* showed the ability to disseminate or transmit ZIKV [62,65,67]. Additionally, *Ae. caspius* had low infection rates. Multiple studies have shown that *Ae. japonicus* is a competent vector for ZIKV. However, the infection, dissemination, and transmission rates of the virus were found to be temperature-dependent, especially at temperatures above 27˚C [59,68,71,73].

## Possible role of *Culex species*

A total of 24 of the 45 studies identified in our systematic review tested *Culex* species; of these 22 included *Cx. quinquefasciatus*, and 14 used mosquito strains recently brought from the field

to laboratory. Thus, the overwhelming weight of evidence is that *Culex* species are unable to transmit ZIKV. *Culex quinquefasciatus* species were refractory in 10 laboratory studies using mosquitoes from colonies (n = 5) and field (n = 5). Of the remaining 11 studies infection rates were very low, dissemination rare, and one report on infectious saliva in one mosquito infected at very high virus titer. Three studies make claims that *Cx. quinquefasciatus* can transmit ZIKV. Guedes et al. [31] showed viral particles in salivary glands by electron microscopy, suggesting this was evidence of transmission, but was not able to demonstrate infectious virus in saliva. A study from China [30] showed transmission of ZIKV from laboratory infected mosquitoes to mice, however saliva from the same mosquitoes, tested by PCR only, showed presence of RNA at 8 dpi (80%) and 12 dpi (10%) but not 16 dpi, a result consistent with the observed detection of decreasing levels of noninfectious pieces of RNA rather than infectious virus. Finally, infectious virus was detected on FTA cards exposed to laboratory-infected field mosquitoes, evidence of transmission [32]. In summary, the studies evaluating *Culex*. laboratory competence indicated that *Culex* species has poor VC for ZIKV overall, but the possibility of geographically isolated strains that are competent must be considered. Macleod and Dimopoulos [13] provide a comprehensive review and interpretation of the data available for *Cx. quinquefasciatus* VC to date.

### Role of temperature

Four of the reviewed articles conducted VC experiments that asked specific questions about the impact of rearing temperatures on European *Culex* species [52], *Ae. detritus* [70] and *Ae. japonicus* [59,73]. Transmission increased with rearing temperature for both *Aedes* species tested and when *Ae. japonicus* was reared at more realistic fluctuating temperature scheme (daily variation between 14°C and 27°C compared 27°C) there was no impact of fluctuating temperatures on VC (Table 5).

## Discussion

As of March 2022, our systematic review confirms that, *Ae. aegypti* and *Ae. albopictus* are by far the most significant vectors of ZIKV worldwide, but that some temperate *Aedes* species, principally *Ae. japonicus*, *Ae. vexans* and to some degree *Ae. detritus*, are capable of transmitting ZIKV efficiently at higher temperatures. As climate change increases average temperatures in temperate regions of the world, these species could grow in importance as secondary vectors of ZIKV [82], if the behavioral characteristics of the vectors and humans favor contact between them. Although, *Cx. quinquefasciatus* received considerable interest and investigation (22 articles) as possible vector for ZIKV, the weight of the evidence suggests it is not important but recognizes the possibility that this species could be relevant for a limited number of localized genetic mosquito vector-virus strain combinations.

Our systematic review illustrates the inherent difficulties in the synthesis and interpretation of these kinds of studies because of the heterogeneities in experimental design and data presentation across them. A recent effort to address this issues published after the completion of our systematic review, provided guidance on minimum data reporting standards for VC studies highlighting this problem, stating "that the complexity of these experiments (vector competence), and the variety of conditions under which they are conducted, make it difficult to meticulously share (and synthesize) all relevant meta data, especially with consistent enough terminology to compare results across studies" [7,82].

Although, we developed and applied a quality assessment tool as part of our systematic review, its application proved cumbersome. First, not all the reviewed manuscripts provided data sets either posted online or in a data repository. New studies should be required to do so,

and this requirement will help overcome problems associated with inconsistent reporting of study metadata. The purpose of our QAS tool was not to rank articles, but to clearly delineate minimum standards, and distinguish between different levels of evidence quality. As an analogy, for field trials, data from randomized controlled trials is seen as the gold standard, followed by non-randomized controlled trials, and finally observational studies [84]. For VC studies, those using infectious laboratory assays, with higher sample sizes (including replication and statistical analysis), a wider range of mosquito and virus strains, and clear measure of transmission represent higher quality data than those that only detect RNA, have smaller samples, and are limited by less natural conditions using colonized mosquito strains and heavily passaged viral strains, but these later studies can be informative is reporting is transparent and study limitations are understood. Our grading tool could not distinguish between well designed and executed studies where the reporting was limited (e.g., short format articles) and poorly designed studies. We encourage subject area experts to modify this tool if it is to be applied for future systematic reviews.

## Research interest and study design often driven by outbreaks and financial considerations

Interest in mosquito vectors of ZIKV, in particular secondary vectors, has been driven by outbreak response with limited publications until 2016, about one year into the ZIKV outbreak in the Western Hemisphere, especially its association with fetal abnormalities.

In the following paragraphs we first describe the characteristics of ZIKV VC studies conducted on 1) forest species, 2) in response to the Yap Island outbreak in 2014, 3) rapid publications during 2015–2017 ZIKV outbreak, and 4) 2018 to present where study quality has improved, and research has been more systematic with the aim to better understand and predict ZIKV disease transmission [7].

Prior to 2007, sylvatic YF surveillance studies isolated ZIKV from wild-caught African forest species, but laboratory-based VC studies to properly assess the competence of these species was not a research priority and limited to a single study included in our systematic review [38]. These studies identified several mosquito species of interest for further study. Unfortunately, technical and logistical challenges for forest species that cannot survive or feed on blood in laboratory conditions in sufficient numbers required to conduct VC studies represented a significant barrier to VC evaluation. These limitations, extend to two studies conducted on forest species from the western hemisphere [60,64].

A second torrent of research on mosquito vectors occurred between 2007–2014 in response to ZIKV outbreaks in the South Pacific, but studies were quite limited overall, one conducted as part of the outbreak response did not include evaluation of transmission [37]. Only two studies on the potential secondary vector *Ae. polynesiensis* showed mixed transmission results [45,57]. This observation highlights the potential for large genetic and VC differences among vector populations of the same species. Calvez et al. [57] did evaluate three *Ae. aegypti* and two *Ae. polynesiensis* populations with a single virus strain from New Caledonia. Ideally, more strains from more locations with distinct virus strains could be evaluated, but these experiments are labor intensive and costly.

At the initiation of the South American ZIKV outbreak, VC studies appeared to be designed based on what mosquito colonies and virus strains could be obtained rapidly. Many of these initial publications, were short communications, and in many ways appeared rushed. There was considerable concern in Europe and the US, that ZIKV could potentially be transmitted by local species, leading to a higher proportion of studies from these countries. Although assessing the risk of potential vectors in these locations made sense, early studies used

mosquito strains that had been in colony often for decades, transitioning into more studies that collected mosquitoes directly from the field that could be brought back to the laboratory. Also, reading between the lines were studies that appeared to retrospectively verify they were detecting infectious virus rather than relying exclusively on PCR to detect ZIKV RNA. Thus, these initial publications often lacked detailed metadata that are important for future meta-analyses or full assessments of a particular species' vector potential. These studies did provide needed quick assessments for making Public Health decisions. ZIKV represents a prototype of what occurs during the initial response to emerging Public Health Emergency, in which an unprecedented amount of research was published very quickly [83].

Research priorities should include a more systematic and comprehensive approach to understanding forest species. ZIKV has been isolated from 15 *Ae.* species as well as a few species from *Culex*, *Eretmapodites*, *Mansonia*, and *Anopheles* genera [4,5,23] where no laboratory studies have been conducted. Virus isolation from field-collected mosquitoes does not constitute evidence for their role as a vector, only that the mosquito fed on an infected host, or contamination occurring within a trap. Laboratory VC studies are required for vector implication to justify further studies on the role of these species in potential spill-over events as well as species whose ecology has changed and could have important implications for ZIKV transmission. For example, *Ae. africanus* and *Ae. furcifur* became important vectors of YF in villages where these species moved freely between the forest canopy and villages nearby where *Ae. aegypti* was absent. Theoretically, the same is possible for ZIKV.

There was significant proliferation of studies on *Cx. quinquefasciatus* based on a premature press release [85] and data from a single study [30], despite significant evidence otherwise. It was not until 2020, that a clear effort was made to replicate the results of the original study from China and evaluate the existing evidence for competence of this species [13]. At the time we had completed our search in March 2022, many groups had evaluated this species as well as many other species within the *Culex* genera. Many of these studies were a part of coordinated research effort by the Zika Alliance. Obadia et al. [79], a study published a few months after concluding our search, provides an illustration of a coordinated research effort using a unified protocol across a wide geographic distribution using strains of *Ae. albopictus* and *Ae. japonicus* directly from the field involving multiple research groups. These kinds of efforts allow experimental studies that would be otherwise logistically impossible for a single research group. A collaborative approach where multiple research groups evaluate one or two local mosquito strains and if possible, an epidemiologically relevant virus strain (e.g., in sites with endemic transmission a strain from the same location), provides more credible and robust conclusions. We now have interesting candidates for future study: *Ae. vexans*, *Ae. japonicus*, and *Ae. detritus* for more comprehensive studies examining the role of ambient temperature.

## Variability in laboratory methodologies

VC studies were generally of high quality methodologically, but have highly variable study designs, using different mosquito and ZIKV virus strains (appropriate), different blood feeding techniques (different artificial feeders, animal blood sources, murine models), virus preparations (dose, frozen versus fresh virus), methods for evaluating infection (testing mosquito bodies versus midguts), dissemination (testing mosquito heads, legs, wings, or salivary glands), and transmission (direct saliva testing, saliva amplified in cell culture, saliva amplified in mosquitoes by intrathoracic inoculations, with mice to confirm transmission). This variability makes comparison across studies difficult, although not impossible. Unfortunately, details on experimental designs were inconsistent across the articles reviewed.

Recently, a minimum data standard for VC experiments was proposed [7] which suggested metadata reporting for mosquito and virus strains used and experimental details and outcomes. Most articles provided most of the necessary experimental metadata, but experimental outcomes were less consistent overall. Mosquito and virus strain metadata varied across the publications, and for some articles the reader was left to infer the age of mosquito colonies and virus passage histories. Some authors provided this information clearly in tables with clear virus passage histories and the number of generations a mosquito strain had been in the laboratory, whereas other articles implied that a strain was local and "recent" but did not provide this information. The best example of this was the HAI strain of *Cx. quinquefasciatus* from China originally found to transmit ZIKV to neonatal mice [30] and later evaluated by MacLeod and Dimopoulous [13]. Neither publication provides precise information on when that colony was established other than the year 2014.

During the review, our team attempted to extract all the metadata mentioned and would recommend that authors include it in a tabular format in the main body of the publication, rather than placing these details in supplementary information or referring readers to a previous publication. Even if "none" or "not known" is included for mosquito colony generations or virus passage history, excluding this information leaves the perception of a lack of transparency. In the following sections, we highlight the more prominent issues associated with interpreting the reviewed articles, but that apply to the VC literature in general.

## Definitions of Infection, Dissemination, Transmission, and other terms

When assessing VC, the following barriers have been described: midgut infection and escape barriers, and the salivary gland infection and escape barriers. Infection implies that virus has passed the midgut infection barrier, whereas dissemination -where virus is found throughout the body of the mosquito- suggests the virus has made it at a minimum through the midgut escape barrier [6,7,82]. Virus in saliva means virus has escaped from the salivary glands. Understanding where the barriers to VC exist for individual species is important to delineate biological mechanisms underlying the ability of virus to make it from the midgut to the saliva at titers sufficient to infect another host. These processes are essential for predicting the potential for emergence events where critical mutations in either the virus or vector would result in a virus-vector pair switch from incompetent to competent. Thus, presentation of stepwise rates has value, but in the context of a Public Health Emergency, like observed with ZIKV, the critical question is "can this species serve as a vector?". To answer this later question, cumulative rates are the most informative since they are using the total number of mosquitoes exposed and tested in the denominator. There is no technical reason, that both stepwise and cumulative rates cannot be presented side by side, and many of the manuscripts did this, but the terms used to present these rates were not consistent. A major concern remains that the terms dissemination rate (DR) and transmission rate (TR) have different definitions depending on the article. One alternative has been the introduction of the terms dissemination efficiency (DE) and transmission efficiency (TE), terminology used in 11 of the reviewed articles but clearly distinguishes between stepwise and cumulative rates [79]. Moving forward, our recommendation is the universal adoption of the terminology for cumulative and stepwise rates as we have presented in Table 5.

Both definitions are presented clearly in Wu et al. [7], who make the additional recommendation that authors include raw data from their experiments in a supplementary appendix so that those calculations can be made by anyone reading the manuscript. They point out the "original raw data may never be reported and is often impossible to reconstruct from provided bar or line charts" and "derived quantities often follow different calculations, with (usually

intentional but) very different biological meaning (e.g., the difference between 'dissemination rate and 'disseminated infection rate' are often used interchangeably)" [7]. Although, most articles clearly defined their parameters, the choice on how to present data was often driven by the desire to tell a story. Interpretation of these parameters are also hindered by the inevitable problem of low sample sizes for dissemination and transmission assessments. When infection and dissemination rates are low, transmission rates presented as a percentage are often based on 1–2 mosquitoes.

Finally, in the reviewed articles the number of days after mosquitoes were fed blood infected with ZIKV was reported in days post-infection (dpi) for most articles and as days post-exposure (dpe) in a few. Although less problematic than the terms dissemination and transmission rate having distinct definitions, the term dpe is more appropriate and should be adopted to standardize terminology in the VC literature.

## Methods that dissect out body parts versus testing bodies, legs, wings, and heads compared to murine models

Although, most published articles use the methodology of testing mosquito bodies (abdomens) to indicate infection, either wing, legs, heads, thoraces, and sometimes other organs to indicate dissemination, there is at the very least qualitative differences when comparing to papers testing dissected midguts and salivary glands. Experiments directly comparing these methodologies would be helpful to the field. The use of genetically modified murine models showed higher infection, dissemination, and transmission rates than experiments using artificial feeding methods. The observation that using a viremic animal to infect mosquitos represents a more realistic model than artificial blood feeding methods is not a new concept [80,81,86]. The ability to implement more realistic and appropriate study designs for VC studies in low- and middle-income countries where ZIKV and other arboviruses are endemic may be limited due to financial and logistical restraints. Thus, more detailed head-to-head comparisons of results from studies using artificial blood feeding with those using animal models or direct human feeds need to be conducted. The same is true for testing transmission from mosquitoes to a new host, that is—head-to-head comparisons of virus detection in saliva to infection of neonatal mice, especially under circumstances when mice become infected in the absence of a clear presence of infectious virus in saliva.

## Minimum quality standards for vector competence studies and how to evaluate them

There are many challenges to the implementation of minimal quality standards for VC studies and there is significant recognition that their absence is a significant problem, but at the same time reluctance to impose undue burden on investigators, especially in locations with limited resources and facilities [7,82]. Our strongest recommendation is the transparent and clear reporting of metadata associated with VC experiments, as suggested by Wu et al. [7], and reflected by our quality grading scale. We strongly support the recommendation to provide access to raw data sets. There is strong consensus that presentation of experimental results needs to clearly report total counts of tested and positive mosquitoes at each stage evaluated (infection, dissemination, and transmission). Also critical is that transmission competence should only indicate the presence of virus in saliva, and not just positive salivary glands [83]. Reliance on PCR assays only, with no component to verify that there is infectious virus present is becoming unacceptable, with the minority of studies in the review not including some recognition of this.

At present, there have been efforts or suggestions for more standardized reporting of experimental data, but not for experimental design or preferred study laboratory assays. Vector competence studies require a balance between well controlled experimental conditions with approximation of natural conditions which are often at odds with each other [7,82]. We found the application of our quality grading tool difficult because much of the metadata especially on virus and mosquito strains, denominators used for competence rate calculations are incomplete or difficult to extract. Also apparent was the lack of experimental replication, which can only be accomplished if sufficient insectary space, mosquitoes, and labor are available to run replicates at the same time, all of which presents numerous technical challenges of rearing enough mosquitoes of the same generation to conduct these experiments. Laboratory conditions cannot be 100% duplicated, running multiple replicates for distinct species/virus or mosquito strain may be less of a priority that other components on the quality grading list. Related to this issue, however, is the observation that few manuscripts provided measures of variability or conducted statistical analyses, a more serious issue. Because our review focused on secondary vectors, the use of multiple virus and mosquito strains was limited, but either additional geographical isolates or studies that include a wide range of geographic mosquito strains together with multiple virus strains will provide convincing evidence of a species VC [79].

We agree with Azar and Weaver [83] that adherence to reporting guidelines such as those suggested by Wu et al. [7], would improve our ability to conduct meta-analyses and draw conclusions from reviews including systematic reviews, but also point out the need for establishing some experimental standards as well. Their recommendations including the use of viral doses that are consistent with viremias observed in human patients that may be easier to achieve using animal models which are not available for ZIKV except for knockout mice with their innate immunity altered for which many research teams may not have access. Insect specific viruses (ISVs) and vector microbiome also play a role in VC, but neither of these parameters were mentioned in the 45 articles we included in our review with one exception.

MacLeod and Dimopolous [13] hypothesized that the virus particles observed in *Cx. quinquefasciatus* mosquitoes challenged with ZIKV [31] represent ISVs rather than ZIKV particles which would be consistent with the absence of viral RNA found in mosquito saliva. We also argue that future systematic reviews potentially exclude articles where no infectious assays are included. We recognize there may be limited circumstances where publication of studies that rely on PCR assays to detect viral RNA are justified, the reliability of this evidence is greatly reduced. Another area that requires input from subject matter experts is minimum number of mosquitoes per stage the require evaluation to have confidence in rate estimates.

Our quality grading tool scored the title of each article, providing a point if the "vector competence" appeared in the title. We do not recommend this moving forward but the appropriate use of key parameters such as susceptible, refractory, transmit deserve discussion. Another item in our grading tool was the issue of sample size. We set 30 mosquitos per virus-mosquito pair evaluated in our quality grading tool as a minimum standard, but this is another experimental component where guidelines would be appropriate. This would be particularly important for reporting results for species deemed negative. We included publications reporting results for refractory species, but this may be an underestimate since there is often a bias against publishing negative results. We recognize, along with other authors [7,82] that complete standardization of VC studies is not possible, but the next logical extension toward this goal is developing quality guidelines, like our quality grading tool to help evaluate the relative quality of manuscripts for inclusion in future metanalyses which are evidently needed to clearly define vector competent species. We view our tool as a good start that needs additional revision by experts before widespread application.

## Conclusions

As of March 2022, secondary vectors of ZIKV that show evidence of transmission, albeit at lower rates than primary vectors are *Ae. japonicus*, *Ae. detritus* and *Ae. vexans*, at higher temperatures, as well as local Australian species *Ae. notoscriptus* and *Ae. camptorhynchus*. There is ample evidence for *Cx. quinquefasciatus* not being an efficient vector of ZIKV. There is a strong need for future research that establishes what are significant differences between experimental approaches such as the use of dissected body parts (midguts, salivary glands) compared to whole body parts (heads, legs+wings, bodies) to evaluate infection and dissemination, and the impact of using murine models as well as other artificial feeding systems on VC parameters, and importantly, to develop large collaborative multi-country projects possibly taking advantage of research networks (eg. ZikaAlliance) [79], Centers for Research on Emerging Infectious Diseases [87] to conduct large scale simultaneous evaluations of specific mosquito species with common protocols to appropriately address the inherent geographic variation in both mosquito and virus strains.

## Supporting information

**S1 Table. Grading tool developed for quality assessment of VC studies using Reporting of Observational Studies in Epidemiology (STROBE) and Strengthening of the Reporting of Molecular Epidemiology for Infectious Diseases (STROME-ID) criteria.**
(DOCX)

**S1 Data. Excel spreadsheet contain raw data extracted from manuscripts to calculate the infection rate (IR), stepwise dissemination rate (SDR), cumulative dissemination rate (CDR), stepwise transmission rate (STR) and cumulative transmission rate (CTR) presented in Table 5.**
(XLSX)

## Acknowledgments

We thank Evangelia Zavitsanou for contribution to the development of Fig 2. Additionally, we are grateful to Mike Turell for technical advice and lively discussion about the issues highlighted in the manuscript.

## Author Contributions

**Conceptualization:** Carlos Alberto Montenegro-Quinoñez, Olaf Horstick, Pablo Manrique-Saide, Silvia Runge-Ranzinger, Amy C. Morrison.

**Data curation:** Marina Bisia, Carlos Alberto Montenegro-Quinoñez, Amy C. Morrison.

**Formal analysis:** Marina Bisia, Carlos Alberto Montenegro-Quinoñez, Valérie R. Louis, Amy C. Morrison.

**Funding acquisition:** Olaf Horstick, Pablo Manrique-Saide, Silvia Runge-Ranzinger, Amy C. Morrison.

**Investigation:** Marina Bisia, Amy C. Morrison.

**Methodology:** Marina Bisia, Carlos Alberto Montenegro-Quinoñez, Amy C. Morrison.

**Supervision:** Amy C. Morrison.

**Writing – original draft:** Marina Bisia, Amy C. Morrison.

**Writing – review & editing:** Marina Bisia, Carlos Alberto Montenegro-Quinoñez, Peter Dambach, Andreas Deckert, Olaf Horstick, Antonios Kolimenakis, Valérie R. Louis, Pablo Manrique-Saide, Antonios Michaelakis, Silvia Runge-Ranzinger, Amy C. Morrison.

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
