## [Decision Letter · Decision Letter 0]

13 Apr 2023

Dear Dr. Morrison,

Thank you very much for submitting your manuscript "Secondary vectors of Zika Virus, a systematic review of laboratory vector competence studies" for consideration at PLOS Neglected Tropical Diseases. As with all papers reviewed by the journal, your manuscript was reviewed by members of the editorial board and by several independent reviewers. In light of the reviews (below this email), we would like to invite the resubmission of a significantly-revised version that takes into account the reviewers' comments. 

We cannot make any decision about publication until we have seen the revised manuscript and your response to the reviewers' comments. Your revised manuscript is also likely to be sent to reviewers for further evaluation.

Sincerely,

Andrea Morrison, Ph.D.

Academic Editor

Andrea Marzi

Section Editor

Reviewer's Responses to Questions

**Key Review Criteria Required for Acceptance?**

**Methods**

-Are the objectives of the study clearly articulated with a clear testable hypothesis stated?

-Is the study design appropriate to address the stated objectives?

-Is the population clearly described and appropriate for the hypothesis being tested?

-Is the sample size sufficient to ensure adequate power to address the hypothesis being tested?

-Were correct statistical analysis used to support conclusions?

-Are there concerns about ethical or regulatory requirements being met?

Reviewer #1: No, please see comments 9 and 17 in my Summary and General Comments

Reviewer #2: -Yes

-Yes

-Yes

-N/A

-N/A

-No

Reviewer #3: See the attached review report

**Results**

-Does the analysis presented match the analysis plan?

-Are the results clearly and completely presented?

-Are the figures (Tables, Images) of sufficient quality for clarity?

Reviewer #1: I don't think that Figure 2 adds any value to the manuscript and should be deleted.

Reviewer #2: -Yes

-Yes

-Yes

Reviewer #3: See the attached review report

**Conclusions**

-Are the conclusions supported by the data presented?

-Are the limitations of analysis clearly described?

-Do the authors discuss how these data can be helpful to advance our understanding of the topic under study?

-Is public health relevance addressed?

Reviewer #1: No, again, please see comments 9 and 17 in my Summary and General Comments

Reviewer #2: -Yes

-Yes

-Yes

-Yes

Reviewer #3: See the attached review report

**Editorial and Data Presentation Modifications?**

Reviewer #1: There are numerous minor editorial errors in capitalization, spelling, abbreviations, etc. Please see my Summary and General Comments

Reviewer #2: Minor Revision

Reviewer #3: See the attached review report

**Summary and General Comments**

Reviewer #1: General Comment: Zika virus (ZIKV) created a major panic back in 2016-2018 when it caused millions of cases of disease in the Americas. As with the other anthroponotic arboviruses, e.g., yellow fever, dengue, and chikungunya viruses, Ae. aegypti is the principal vector. Yes, Ae. albopictus has been involved in a few outbreaks, but see comment 1 below. Because these anthroponotic viruses require the same mosquito to take two separate blood meals on a human, only anthropophilic mosquitoes that preferentially feed on humans are likely to be involved in the transmission of these viruses. This should be emphasized more in the paper. My biggest concern, as mentioned in comments 9 and 17, is that because authors use completely different definitions of a transmission rate, simply reporting the rate that these authors claimed is very misleading and may make an essentially incompetent mosquito appear highly efficient.

I understand that for most of the authors, English is a second language, but there were a lot of minor inconsistencies. Editors, reviewers, and readers can’t see how the study was done, but if the writing is careless (see numerous comments below), they are less likely to believe that the authors were careful in how they did their study. 

Specific Comments: 

1. Line 46 and numerous comments throughout the entire manuscript. Yes, Ae, albopictus can readily transmit ZIKV and yes, Ae. albopictus readily feeds on humans. However, how important is Ae. albopictus in the transmission of ZIKV, chikungunya virus, or dengue virus? After their introduction into the Americas beginning in 2013, about 10,000 cases of chikungunya and Zika were diagnosed and reported in the U.S. (given the inefficiency in diagnosis and reporting, this was a severe underreporting of cases). Despite many, many reported imported cases of chikungunya and Zika in the southeastern U.S., where Ae. albopictus is one of the principal pest mosquitoes, not one case of locally transmitted chikungunya or Zika was reported from any location that did not also have a significant Ae. aegypti population. Because they are not as anthropophilic as Ae. aegypti, a single Ae. albopictus is not likely to take multiple blood meals from a human, and Ae. albopictus are not as likely to be involved in significant transmission of anthroponotic pathogens such as dengue, chikungunya, Zika, and yellow fever as Ae. aegypti. Note, in the outbreak on the islands around the Indian Ocean, there were very few other animals present to feed on, so the Ae. albopictus did take multiple blood meals from humans, and the same thing happened in a few locations in Italy.

2. Lines 64-66: Why are the use of murine models and infectious virus assays critical gaps? To me, it is critical that the studies use infectious virus assays as qRT-PCR may detect some lingering, non-infectious pieces of RNA remaining from the blood meal in an uninfected mosquito resulting in it inappropriately being called positive. Similarly, the use of artificial blood meals often produce lower infection rates than when mosquitoes ingest the same virus titer from a viremic host. Therefore, it is not immunocompromised murine models that are a critical gap, but rather the lack of use of these models. 

3. Lines 86-87: I found this confusing. Testing mosquito bodies for infection is MUCH more efficient than testing midguts. Similarly, testing salivary glands for virus dissemination is very inefficient compared to testing either legs or heads. Note, I have seen papers that test heads to test the presence of virus in salivary glands, but in a mosquito, the salivary glands are in the thorax, not the head.

4. Line 110-120: Very well said!

5. Line 295: Yes, “Days post infection (DPI)” is commonly used, but shouldn’t it be “Days post exposure (DPE)” as many of the mosquitoes did NOT become infected and in some cases, none became infected, so how can it be days post INFECTION?

6. Line 296: What is a Transmission rate versus a Transmission efficiency? I have seen these terms misused numerous times in recent years, and they need to be defined in this paper. Because people define these terms differently, it is important that the authors first define how they are being used in this paper and then correct the citations so that the cited paper are using the same definition, i.e., if the paper states that it is using transmission efficiency, but it is using the definition of a transmission rate (as defined in this review), then the paper should be cited as using a transmission rate despite claiming to have used a transmission efficiency.

7. Line 297 (Table 1): I found this table to be very confusing. Obviously, the various “Results” were written by different people with different formats, i.e., summary first or methods first. These all need to be rewritten to be in a similar format. Either is fine, but they need to be consistent. There are also a bunch of minor errors which I will include here, rather than in the section below on Minor Comments. It is important that the writing be consistent so that the reader can easily go from one paper to the other and compare the results. Someone needs to go through ALL of these and make sure that they are consistently presented.

 a. Reference 68 and 69: What is USUV? It is not established in the paper. I know that it is Usutu virus, but will the readers know this? Also, ZIKV is used in many of the papers. However, ZIKV was not established in the heading for this table and ALL abbreviations need to be established in figures and tables.

 b. The titers of the infectious blood meal are listed in numerous different ways. For example, it is reported as “log10 PFU/ml” (most of the papers) or as “logs” (see paper 39 and others), or “3x10(5) PFU (See paper 30), or 105.4 PFU (note, the 5.4 was in a subscript) (see paper 47), or 2,530 PFU/ML (see paper 40), or merely as particles (see reference 40). Again, the authors need to be consistent. All of these “PFU” titers should be in the same format, either X.X log10 PFU/ml (note, the “10” should be a subscript) or 10x.x PFU/ml (note, the x.x should be a superscript). Minor, but reporting a titer to two decimal places is reporting data beyond the level of accuracy and should be rounded to the nearest tenth of a log (see reference 44).

 c. Some of the English usage is a bit confusing. For example, in paper 45, it states, ‘Aedes aegypti and Ae. polynesiensis in ZIKV spread was also suspected in addition of Ae. aegypti, supported by its ability to transmit laboratory colonies (F16 to F18) were orally infected with a French Polynesian strain) at a 7 log10 TCID50/ml and evaluated at 2, 6, 9, 14, 21 d DPI.” This, and many of the others, needs to be completely rewritten. Note, why was it “…, 21 d DPI.?”

 d. Why is “Hemotek” listed under “Mosquitoes/Virus” for reference 37? The method of virus exposure is not list for others under this column, but rather under results for some of the references.

 e. Check spacing. There were numerous places where a space was left out. For example, in paper 57, “at 9dpi,”should be at “9 dpi,” (note, see above and this should be at 9 dpe). Likewise, in paper 58, “log10PFU/ml” should be “log10 PFU/ml,…”

 f. Why are the infectious titers given in the “Mosquito/Virus” column for some papers, but in the “Results” column for other papers. You need to be consistent. Putting these critical elements in different places makes it much more difficult for the reader. 

 g. Also, is it “dpi” as in paper 57 and others or “DPI” as in paper 54 and others? Likewise, is it “ml” as used in most papers, “ML” as used in paper 37, or “mL” as used in paper 77?

 h. What is a “PFUe” in paper 77?

 i. Why in paper 77 does it have titers expressed as “log10” with only the “0” as a subscript?

8. Line 310 (Figure 2): I’m not sure what this adds to the paper. Yes, according to the criteria selected by some of the authors, some of the papers were stronger than others, but which papers received high scores and which received low scores, so that we know which papers’ results/conclusions are more believable, is already indicated in Table 1. 

9. The difference in how the studies defined these rates is CRITICAL! For example if study A defined the Transmission Rate (TR) as the number of mosquitoes transmitting virus divided by the number of mosquitoes with a disseminated infection tested and study B defined the TR as the number of mosquitoes transmitting virus divided by the number tested (this is what is said on lines 418-420), and the actual numbers were 100 tested, 50 infected, 3 with a disseminated infection, and 2 transmitting, then in study A, the TR was 67%, yet in study B, it was 2%. How will the reader know which it really was? Again, you need to state one definition and adjust the rates so that the studies are comparing the same thing. As written, the species tested in study A would appear to be a highly effective vector, while those in study B would appear to be very poor vectors. This can be VERY misleading as if in study A, the numbers were above for mosquito species X, but in study B, the numbers were 100 tested, 50 infected, 20 with a disseminated infection, and 20 transmitting for mosquito species Y, so if study B used the definition for study B, it would report a TR of 20% for species Y, so in your paper, you would report that species X has a TR of 67%, while species Y has a TR of only 20%, despite the fact that the actual TR of species Y was 10-fold higher than species X. If you report the rates from the various studies, which this paper needs to do, you NEED to state and use a consistent definition of that rate, so if the TR is the percentage of all mosquitoes tested (which is the classical definition), then the rate for study A needs to be changed to 2%, even if they reported 67%. That allows the reader to compare the results of the various studies. If this is not done, then this paper is completely misleading and should not be published!

10. Line 433 (Table 4): How can 19 studies in the “Infection” column be only “20%?”

11. Line 478 (Table 5): See comment 9 above. It can be extremely misleading if the data presented here are using different definitions of transmission. These rates need to be adjusted to a single definition. Note, using the percentage of mosquitoes with a disseminated infection is itself very misleading as mosquitoes with a very low dissemination rate may appear to have a very high transmission rate. Look at the table, how can a mosquito have a 43% infection rate and a 90% transmission rate?

12. Lines 508-509: Again, how can you have “significant transmission of ZIKV” with only a 42% infection rate?

13: Lines 529, 531: First of all, it is Ochlerotatus, not Ochelerotatus. More importantly, is it Ochlerotatus detritus or Ae. detritus? The genus Ochlerotatus was briefly establish for some of the Aedes species a number of years ago, but nearly everyone has gone back to using Aedes. Note, if you use Ochlerotatus, then it should also be Oc. taeniorhynchus, which I do not recommend.

14. Line 535: This is the second time that the authors mention that one of the reasons that Ar. subalbatus was tested was because they develop in human waste lagoons where ZIKV could be shed in urine. I believe that the study found no risk of the larvae becoming infected despite exposure to very high titers of ZIKV in a water/urine suspension in the laboratory. Repeating the statement above, without clarification that they didn’t find that risk, is misleading.

15. Lines 575-579: Will warmer temperatures really affect ZIKV transmission? Remember, yellow fever and dengue were once common diseases in the United States, including Philadelphia, New York, and Boston. It was not warmer temperatures, but social economic status that allowed for the transmission of viruses by Ae. aegypti. However, warmer temperatures will allow for the greater and more efficient transmission of zoonotic viruses.

16. Lines 642-644: Yes, the mere isolation of a virus from a field-collected species does not mean that it is involved in the transmission of that virus. Even if a species is completely incompetent for a particular virus, if it had recently fed on a viremic host, it would contain virus. Similarly, when mosquitoes are sorted for virus detection, legs of numerous specimens break off and may adhere to the body of a mosquito of a different species. If that leg came from an infected mosquito, then whatever species it adhered to would appear to be positive. You might want to add a little about why the mere detection of virus from a species does not make it a vector.

17. Lines 703-708: Despite what was said in this paragraph, the use of the number of disseminated mosquitoes as the denominator is a relatively new use, and as I explained in comment 9, may be a very misleading number. If only 1/100 mosquitoes has a disseminated infection and that one transmits virus, then if only the disseminated mosquitoes are counted, the transmission rate for that species is 100%, making it appear that the species is very important, while if in another species, in which 100% develop a disseminated infection and only 20% transmit, that species would have a 20% transmission rate despite have 20-fold higher vector competence. This new definition came out of the Institute Pasteur, and I believe that they introduced it to make their work look more important, i.e., in the example above, a 100% transmission rate is much more publishable than a 1% one. From a public health standpoint, we need to know how efficiently a particular mosquito species is as a vector, saying that the species above has a 100% transmission rate might lead to the control of the wrong species. You sort of indicate this on line 717-718.

18. Line 733: What do you mean by “traditional methods?” the use of animal models and checking for infectious virus are the traditional methods.

19. Lines 734-736: Various studies have shown that the amount of virus injected when feeding on a vertebrate host is significantly higher as compared to the amount of virus injected during artificial methods to collect saliva. Similarly, transmission rates are usually higher to a vertebrate than when collecting saliva and testing it. Many laboratories don’t want to recognize this as it is difficult, or even impossible, for them to maintain the necessary animal colonies. Therefore, they use the less scientifically efficient artificial techniques.

20. Lines 769: Unfortunately, using viral doses that are consistent with viremias observed in humans in an artificial blood meal will result in significantly lower infection rates in mosquitoes than if they ingested the same dose of virus from a viremic animal (i.e., human). Therefore, studies using those doses would greatly underestimate the importance of that mosquito species.

21. Lines 794-796: Yes, if the study only tested 15 mosquitoes and did not find transmission, that really doesn’t mean that the mosquito tested was incompetent. However, if they tested 10 specimens and eight of them transmitted virus by bite, then I think that they have clearly shown that this species may be very important. 

22. Line 812: I would be very surprised is there is any difference in infection rates determined by testing bodies or midguts, but it is a lot more labor intensive to test midguts. 

23. Lines 812-813: Heads and salivary glands are testing two different things. It is possible to have a disseminated infection, i.e., positive head and the salivary glands be negative. Likewise, a positive salivary gland does not mean that there will be virus in the saliva.

24. References: These need to be formatted properly.

 a. Only the first word and proper nouns in a reference title should be capitalized. See references 13, 36, 39, and others. 

 b. References should not be in all caps. See reference 20. 

 c. Shouldn’t the journal for reference 28 be “Trans R Soc Trop Med Hyg” instead of “Transactions of The Royal Society of Tropical Medicine and Hygiene?”

Minor Comments:

25. Lines 49-50: Should “…identify knowledge key gaps” be “…identify key knowledge gaps?”

26. Line 59: Why is there a “comma” after “quinquefasciatus?” 

27. Line 98: “Aedes” needs to be written out the first time it is used in the body of the manuscript.

28. Line 103: Traditionally, numbers less than 10 are written out, so this should be “eight genera…” Note, it appears that there is an extra space after “genera.” In contrast, all numbers used in measurement are given as numerals. Therefore, on line 149, “five years…” should be “5 years…” Please check the rest of the manuscript and correct these.

29. Line 165: Why is there a space after the “)” and before the “comma?”

30. Line 169: Should “WHOLIS” be “(WHOLIS)?”

31. Line 191: Should “wild mosquitoes” be “field-collected mosquitoes?”

32. Line 203: “n= 8.986)…” should be “(n= 8,986)…” Note that a comma was correctly used elsewhere.

33. Line 291: “Medical Entomology [3] and” should be “Medical Entomology [3]), and” with both a close parenthesis and a comma.

34. Line 330 and many others: Why write out “Zika virus” when “ZIKV” was established and used many times earlier in the manuscript?

35. Line 371: Why are Aedes, Culex, Anopheles, etc. written out each time they are used in this table? They should only be written out the first time that they are used.

36. Line 377: Why is the “10” in “log(10)” in parentheses? 

37. Line 380: “…ten also…” should be “…10 also…”

38. Line 393: “7 studies” should be “seven studies.” See also lines 452, 475, and others.

39. Lines 440-441: Why is “(Coffey et al. 2018)” listed here? Should this be assigned a reference number? Note, all of the reference numbers may need to be adjusted.

40. Line 441: Why is “Sabethes” written out throughout the manuscript? You don’t write out Aedes or Culex? These should all be “Sa. species” as Sabethes was established on line 376. The same is true for Armigeres.

41. Line 443: Again, reporting a titer to two decimal places is reporting data to a level that is not accurate. Yes, I know that some papers do this, but that does not make it right, and you should not make the same error. On this line, simply report the titer as “6.8 log10 PFU/ml.: Note, elsewhere in the paper, PFU is in all caps. Please be consistent.

42. Line 484: Change “vector were…” to “vectors were…”

43. Line 501: What does “more consistent infection rates” mean? Dose that mean that the rates were consistent, i.e., 5%, 6%, 4% in three separate trials or that they were consistently higher than those for Ae. aegypti tested from that area?

44. Line 515: Why is there a “comma” after “injection,?”

45. Line 516: “Colorado USA…” should be “Colorado, USA…”

46. Line 520: “Ae. triaseriatus” should be “Ae. triseriatus”

47. Line 521: Why is “(Hart et al. 2017)” listed as a reference by name rather than by reference number 51?

48. Line 531: Change “or Ae.” to “nor Ae.”

49. Line 624: Why “(2018)” rather than “(57)?”

50. Line 766: Why “(2019)” rather than “(79)?”

Reviewer #2: In the review “Secondary vectors of Zika Virus, a systematic review of laboratory vector competence

studies” Bisia and colleagues have conducted a thorough and transparent review of the existing Zika virus vector competence literature with respect to laboratory studies conducted on potential secondary vectors. As a laboratory construct, vector competence represents the research community’s best method to examine a complex interplay of factors that contribute to a given mosquito species or populations ability to serve as vector for a given agent. That said, the methodology/discipline suffers from a pervasive lack of unity with respect to methodologies, definitions and other critical parameters. In conducting these analyses, Bisia et al are able to illustrate this issue alongside their primary objective of identifying potential secondary vectors for ZIKV. Given the increased incidence in the emergence of zoonotic diseases, including arboviruses, increasing the rigor and reproducibility in this discipline is paramount. This manuscript illustrates this in the context of the ZIKV outbreak in the Americas from 2015-2018. 

In total the manuscript is well-written, articulates its methodologies clearly and transparently, identifies promising avenues for further development, and draws appropriate and scientifically sound conclusions. Publication of this manuscript is therefore highly recommended albeit with the following suggestions and comments put forward for consideration. 

Major Points

1. Lines 141-142 and throughout: Something to note; the connection between Culex quinquefasciatus and ZIKV was NOT initially facilitated by peer-reviewed research. A group that was affiliated with public health agency FioCruz in Recife issued a press release (e.g. https://www.theguardian.com/world/2016/mar/03/zika-virus-carried-more-common-mosquito-scientists-say) in the absence of public data a full year prior to the publication of their manuscript. 

2. In the text and in the grading tool (Supplemental Table 1), the authors refer to the use of “low passage virus” as a parameter. While considering colonization of mosquito populations, authors define the association between the point value and the number of generations in the laboratory. I feel that a defined passage number should be indicated in the table, as “low passage” in the absence of a defined number is ambiguous.

a. Related; do the authors distinguish between passaged primary isolates or recombinant viruses in these experiments? Can this be noted in the text either way?

3. Lines 760-761: The authors note the lack of experimental replication in the vector competence literature. Although the authors note that this is understandable due to the labor cost of these experiments, it also worth noting that from a strict standpoint such replication is not technically possible. Exposure of a given laboratory population is associated with a given generation of colonization (e.g. F3, F4, etc). In the case of mosquitoes that do not lay eggs that can undergo desiccation and storage such as Culex spp, it is logistically impossible to perform an exact replica of an experiment with the same generation of mosquito from the same population. While such experimental replicates ARE technically feasible in the case of some mosquito species, it also requires substantial up-front effort to rear large quantities of mosquitoes and store substantial cohorts of eggs to allow for generation matching.

Minor Points

1. Line 138: This may be pedantic; however, I believe this sentence requires clarification. Specifically, the phrase “much of our knowledge comes from…” seems inappropriate as our current knowledge base far surpasses what was known solely based on the YFV surveillance. 

2. Line 483: The word “both” on this line appears to be misplaced as there is no secondary factor after “laboratory evidence”.

3. Line 491: The correct spelling is Sabethes cyaneus

4. Lines 495 and 497: The authors here refer to Aedes polynensis, which is a misspelling of the species name Aedes polynesiensis.

5. In line 576, the authors identify Ae. japonicus, Ae. vexans, “and to some degree” Ae. detritus as potential secondary vectors given higher temperatures. In lines 659-661 the authors note that Ae. vexans, Ae. japonicus, and Ae. polynesiensis as possible candidates for follow up studies. Please correct for consistency.

Reviewer #3: See the attached review report

PLOS authors have the option to publish the peer review history of their article (what does this mean?). If published, this will include your full peer review and any attached files.

Reviewer #1: No

Reviewer #2: No

Reviewer #3: Yes: Thomas Obadia
---

## [Decision Letter · Decision Letter 1]

10 Jul 2023

Dear Dr. Morrison,

Thank you very much for submitting your manuscript "Secondary vectors of Zika Virus, a systematic review of laboratory vector competence studies" for consideration at PLOS Neglected Tropical Diseases. As with all papers reviewed by the journal, your manuscript was reviewed by members of the editorial board and by several independent reviewers. The reviewers appreciated the attention to an important topic. Based on the reviews, we are likely to accept this manuscript for publication, providing that you modify the manuscript according to the review recommendations. 

Thank you for your resubmission! I would highly recommend reviewing the manuscript for remaining grammatical issues as indicated by the reviewers. Additionally, please pay close attention to Reviewer 1's comments regarding transmission rate.

Sincerely,

Andrea Morrison, Ph.D.

Academic Editor

Andrea Marzi

Section Editor

Thank you for your resubmission! I would highly recommend reviewing the manuscript for remaining grammatical issues as indicated by the reviewers. Additionally, please pay close attention to Reviewer 1's comments regarding transmission rate.

Reviewer's Responses to Questions

**Key Review Criteria Required for Acceptance?**

**Methods**

-Are the objectives of the study clearly articulated with a clear testable hypothesis stated?

-Is the study design appropriate to address the stated objectives?

-Is the population clearly described and appropriate for the hypothesis being tested?

-Is the sample size sufficient to ensure adequate power to address the hypothesis being tested?

-Were correct statistical analysis used to support conclusions?

-Are there concerns about ethical or regulatory requirements being met?

Reviewer #1: See general comments.

Reviewer #2: Accept

Reviewer #3: Yes

Yes

Yes

N/A

N/A

No

**Results**

-Does the analysis presented match the analysis plan?

-Are the results clearly and completely presented?

-Are the figures (Tables, Images) of sufficient quality for clarity?

Reviewer #1: See general comments.

Reviewer #2: Accept

Reviewer #3: Yes

Yes

Yes

**Conclusions**

-Are the conclusions supported by the data presented?

-Are the limitations of analysis clearly described?

-Do the authors discuss how these data can be helpful to advance our understanding of the topic under study?

-Is public health relevance addressed?

Reviewer #1: There are some issues. Please see general comments.

Reviewer #2: Accept

Reviewer #3: Yes

Yes

Yes

Yes

**Editorial and Data Presentation Modifications?**

Reviewer #1: There were numerous minor grammatical issues. I pointed many of these out in the general comments.

Reviewer #2: Accept

Reviewer #3: (No Response)

**Summary and General Comments**

Reviewer #1: General Comment: This version is much improved. However, I am still a little concerned about how transmission rate is used in this paper. The use in Table 5 is fine but its use in the text can be misleading (see comment 8 below). Also, there were numerous minor editorial errors (multiple uses of a numeral, at random, for numbers less than 10). To me, this illustrates some carelessness. More importantly, many of the rates given in Table 5 are wrong (see comment 5 below). 

Specific Comments: 

1. Line 124: Vector competence studies generally do NOT include “inoculation” or 

‘intrathoracic inoculation,” but may use that technique to examine either midgut escape or salivary gland barriers. Inoculated mosquitoes cannot be used to determine a transmission rate as this technique bypasses two critical barriers, midgut escape and salivary gland barriers.

2. Table 1, Guo reference: As they did not check for actual, infectious virus, isn’t it much more likely that they were detecting lingering small pieces of noninfectious RNA, rather than a transient infection, which as you indicate would be totally new and unexplained paradigm?

3. Lines375-383: I added up the numbers for the various locations, 16, 10, 5, 4, 4, 2, 1 (not identified), and 2 (multiple continents), and the total was 44.Where was the missing study done? Based on Table 2, shouldn’t the 16 for the Americas be 17?

4. Lines 388-389: Why is Armigeres not listed on line 388? Li et al. did test the vector competence of this species for ZIKV.

5. Table 5: Much better, but there are still some problems.

 a. For Ae. luteocephalus, how is it possible for the CDR to be 42% if the SDR was only 37%? By definition, the SDR has to be greater than or equal to the CDR.

 b. For Ae. vigilax, how is it possible for the CDR to be 27% if the SDR was only 10%?

 c. For Aedes vitalis, how can the CDR be 17% if only 14% of the mosquitoes were infected?

 d. I am a little confused about the calculations for Ae. japonicus. If the IR was 87%, and the CDR was 19%, should the SDR have been 22%, not 27%. Alternatively, if the SDR was 27%, shouldn’t the CDR have been 22%. There is the same problem with the TRs. If the SDR was 27 and the STR was 57%, then the CTR should have been 15%, not 11%. However, if the SDR was 22% (see above) and the STR was 57%, then the CTR should have been 12.5% (note, the 1% difference might be due to the decimal points not shown in the original data. 

 e. As illustrated by the above four examples, there are some errors in the calculations of these rates. Please go back and confirm all of the rates as I found several others that are also wrong. 

 f. The importance of clarifying the transmission rate definitions is illustrated by Sa. cyaneus. If the paper defined TR as the number transmitting/number tested, then the TR was 1%. However, if they used the inappropriate definition of number transmitting/number disseminated, it was 100%. In the abstract of the paper, it would have said, the TR for Sa. cyaneus was 1% (or 100%) depending on the definition used in the paper. Unfortunately, most people only read the abstract to get this information. 

6. Lines 549-551: As illustrated by the example of Sa. cyaneus in 5(f) above. The definition of transmission rate is critical. If you used the STR, the transmission rate would be 100%, i.e., more efficient that Ae. aegypti, but if you used the CTR, then it would be 1% and extremely inefficient. You need to explain which you are using, and if using the STR, then some extremely inefficient vectors would be classified as extremely efficient, and this is not only misleading, but public health people might focus their control on an extremely inefficient species and ignore more important vectors, resulting n more disease.

7. Lines 568: Again, you need to be clear as to which definition of dissemination you are using for the various studies. Species with a relatively low infection rate may appear to have a high dissemination rate, making them look much more efficient.

8. Lines 574-576: This really illustrates the problem. If only 34% of the Ae. notoscriptus became infected (and only 3% had a disseminated infection), how could they have a 42% transmission rate when only about 1% of the virus-exposed mosquitoes actually transmitted ZIKV? Obviously, this species is a very inefficient vector of ZIKV. However, someone reading this sentence would think that it was an efficient vector. This is very misleading. 

9. Lines 632-634: You might want to add that noninfectious pieces of RNA have been shown to persist for long periods of time in a mosquito, and thus, this study detected these noninfectious pieces of RNA, but the amount of these noninfectious pieces would decline with time providing the results reported.

10. Lines 810-814: This is important.

11. Lines 817-820: Yes. If someone reads the paper, they can understand how the authors defined a TR. However, many of these papers simply state in the abstract that the TR for species X was 50% and most readers do not read the paper. However, if they had used a stepwise rate and only two of the 100 mosquitoes tested had a disseminated infection, and one of these transmitted, then instead of being an efficient vector, as indicated in the abstract, it would be a very inefficient vector. As indicated in the paragraph, this could have serious public health liabilities. We need to be consistent in our definition of a transmission rate, and if the purpose is to identify potential vectors, then it needs to be the number transmitting/number tested.

12. Lines 835-839: Thanks for pointing this out.

13. Lines 855-858: I’ve played with this quite a bit. I have often found that a single mosquito bite on the tip of a mouse’s tail usually contains >1,000 PFU of virus, but a saliva sample rarely contained mor than 100 PFU, and a mosquito that had transmitted to a rodent often contained no virus in its saliva sample. While it is easier and less costly to simply collect saliva and test it, it tends to underestimate the actual TR. This is consistent with what you are reporting.

Minor Comments:

14. Line 98: “VC” needs to be established in the main body text.

15. Line 119: “vector capacity…” should be “vectorial capacity…”

16. Line 148: What is “transmission efficiency?”

17. Line 163: This does not read well. Should “where” be added so that it is now “…and a period where VC studies…?”

18. Line 319: Why is “Days” post exposure capitalized, while “days” post infection is not?

19. Line 319: “contain…” should be “containing…”

20. Line 334: Why establish “MS” if it is not used again?

21. Line 335: Why establish “TS” if it is not used again?

22. Line 388: I don’t think that you need both “genus” and “genera” in this sentence. That is repetitive.

23. Line 400: “4 articles…” should be “four articles …” 

24. Line 413: “using well described the viral strains…” should be “using well described viral strains…”

25. Line 469: Why establish “DIR” or “YR(D)” if they are not used again?

26. Line 493: Why report a titer to a hundredth of a log? This is not accurate. I don’t care if that was reported in the paper cited, it should be 6.8 logs.

27. Line 501: “ten mosquitoes…” should be “10 mosquitoes…”

28. Line 663: “these kind studies…” should be “these kinds of studies…”

29. Line 702: Why establish “YF” if it is not used again?

30. Line 760: “1 or 2 local…” should be “one or two local…”

31. Line 929: Shouldn’t “rates primary vectors…” be “rates than primary vectors…?

32. References

 a #16 (Cao-Lormeau): Shouldn’t “Polynesia” be capitalized?

 b. #18: Why is “Transactions of The Royal Society of Tropical Medicine and Hygiene” written out. Check others.

 c. #37: Why is “chikungunya” capitalized?

Reviewer #2: In the revision of the review “Secondary vectors of Zika Virus, a systematic review of laboratory vector competence

studies” Bisia and colleagues have addressed and/or justified their rationale for comments and concerns expressed by myself and the other reviewers. As such, I support the publication of the manuscript in its current form.

Reviewer #3: Following my initial review and that of the other two anonymous reviewers, the authors have substantially improved their manuscript and I can only commend them for going through what must have been some very tedious modifications and proof-reading.

My main criticism at first submission was the long delay between the end date for including papers in this systematic review and timing for submission (more than a year). This resulted, among other things, in reviewing that paper after I had myself published what the authors were calling for as “future efforts” in the discussion. In an effort for transparency, I decided it was only fair that I disclosed my name during the review process, and that hopefully it did not make me sound like an author wanting his word credited. I appreciate that for any systematic review, and end date must be set and respected. It is unfortunate that publication of this paper will happen more than a year and a half later, though we can all agree that “publishing science takes time” and sometimes delays out of authors’ control contribute to lengthy publication process.

Among the strongly positive points now made by the authors is the call for harmonized reporting standards (e.g. clearly defining stepwise/cumulative rates), a major point raised by anonymous Reviewer #1. I can only hope that future VC work will build upon that terminology. The QAS, though remaining of high relevance, also feels better now that it is introduced as “guidelines” towards standardization rather than a rewarding metric.

I now only have minor comments that, once addressed, will make that paper acceptable for publication. Upon reading that R1 revision, I consider my major comments addressed and can recommend this article for publication PLoS Neglected Tropical Diseases pending corrections for remaining typos and few syntactical errors.

Minor comments

L155: yellow fever is first mentioned here, then again at L420, then at L702 (where it is abbreviated YF) and L745-746 (“yellow fever” again). The abbreviation may come at L155 and YF be used later for clarity and consistency.

L295: “considers” should probably not have an “s” here.

L539+: consider mentioning somewhere that the now harmonized “NT” stands for “Not tested”.

L725: “accessing” should probably be “assessing”. This was actually a minor suggestion from my first review and marked as “done” by the authors, could you double-check that other small typoes highlighted earlier were not corrected then wrongly rejected/reverted ?

L762: Missing an “s” at “provides”.

L801: accessing/assessing, again

L851-853: the sentence does not make sense as is though I think understand what the authors meant. Consider inverting the beginning so that it’s clear that these constraints dictate what can and can’t be reasonably expected from low/middle-income countries in terms of lab capacity and experiments.

L889: Shouldn’t “geographically” be “geographical”?

PLOS authors have the option to publish the peer review history of their article (what does this mean?). If published, this will include your full peer review and any attached files.

Reviewer #1: No

Reviewer #2: No

Reviewer #3: Yes: Thomas Obadia

Figure Files:

Data Requirements:

Reproducibility:

References

---

## [Decision Letter · Decision Letter 2]

10 Aug 2023

Dear Dr. Morrison,

Thank you very much for submitting your manuscript "Secondary vectors of Zika Virus, a systematic review of laboratory vector competence studies" for consideration at PLOS Neglected Tropical Diseases. As with all papers reviewed by the journal, your manuscript was reviewed by members of the editorial board and by several independent reviewers. The reviewers appreciated the attention to an important topic. Based on the reviews, we are likely to accept this manuscript for publication, providing that you modify the manuscript according to the review recommendations. 

We appreciate the edits you have made in response to the reviewers' comments and think that this would be a valuable publication. We request a few additional edits be made in response to reviewer #1's comments as we believe they will even further improve upon the manuscript and provide the final details needed.

Sincerely,

Andrea Morrison, Ph.D.

Academic Editor

Andrea Marzi

Section Editor

Reviewer's Responses to Questions

**Key Review Criteria Required for Acceptance?**

**Methods**

-Are the objectives of the study clearly articulated with a clear testable hypothesis stated?

-Is the study design appropriate to address the stated objectives?

-Is the population clearly described and appropriate for the hypothesis being tested?

-Is the sample size sufficient to ensure adequate power to address the hypothesis being tested?

-Were correct statistical analysis used to support conclusions?

-Are there concerns about ethical or regulatory requirements being met?

Reviewer #1: Methods are fine and no concerns

Reviewer #2: Meets all review criteria and no ethical/regulatory issues noted

Reviewer #3: (No Response)

**Results**

-Does the analysis presented match the analysis plan?

-Are the results clearly and completely presented?

-Are the figures (Tables, Images) of sufficient quality for clarity?

Reviewer #1: please see General comments

Reviewer #2: The results meet all the aforementioned criteria

Reviewer #3: (No Response)

**Conclusions**

-Are the conclusions supported by the data presented?

-Are the limitations of analysis clearly described?

-Do the authors discuss how these data can be helpful to advance our understanding of the topic under study?

-Is public health relevance addressed?

Reviewer #1: please see General comments

Reviewer #2: The conclusions meet all the aforementioned criteria

Reviewer #3: (No Response)

**Editorial and Data Presentation Modifications?**

Reviewer #1: please see General comments

Reviewer #2: Accept

Reviewer #3: (No Response)

**Summary and General Comments**

Reviewer #1: General Comment: This version is much improved. However, the are still a few issues that need to be corrected. Please see my comments below. 

Specific Comments: 

1. Line 406: “ten generations…” should be “10 generations…”

2. Table 5

 a. What does “Raw data presented in” at the end of the table heading mean?

 b. For Ae. notoscriptus, you report IR- 29%, SDR-21%, CDR-10%. However, if 29% are infected, and 21% of those have a disseminated infection, then only 6% of the total would be infected. On the other hand, if the CDR was 10% and 29% were infected, the SDR would be 34%

 c. The results for Ae. vigilax IR- 34%, SDR-47%, CDR-27% ref43 and Ae. procax IR- 57%, SDR-50%, CDR-17% ref 43 seem to be partially crossed. The SDR-50% and CDR-17% reported for Ae. procax work for Ae. vigilax, and the SDR-47% and CDR-27% reported for Ae. vigilax work for Ae. procax. Alternatively, the two infection rates were switched.

3. Line 550: What does, “with both laboratory evidence” mean?

4. Line 558: Yes, it was only one mosquito that became infected and transmitted ZIKV. However, the sentence gives no indication of the sample size, so if they had only tested that one Sa. cyaneus, it would appear to be a very efficient vector. The authors might want to modify the sentence to, “…had one mosquito out of 60 tested with…

5. Line 583: Why use the word “Important” for those two species, particularly as they were refractory. Delete “Important.”

6. Line 674: I agree that that data sets should be provided. However, how are the authors going to guarantee that they “will be required.” Shouldn’t this be, “New studies should be required to do so…”

7. Line 824: Shouldn’t “Both these definitions…” be “Both of these definitions…?

8. Lines 852-858: You might also want to cite Turell MJ. Reduced Rift Valley fever virus infection rates in mosquitoes associated with pledget feedings. Am J Trop Med Hyg. 1988 Dec;39(6):597-602 that clearly showed that when feeding mosquitoes (both Culex and Aedes) on blood from a viremic hamster or on the hamster itself, the infection rate in the mosquitoes was significantly higher if they had fed on the hamster although the amount of virus ingested was identical.

9. Line 949: Thanks

Reviewer #2: In the revision of the review “Secondary vectors of Zika Virus, a systematic review of laboratory vector competence

studies” Bisia and colleagues have further addressed concerns, in particular in regards to Table 5, and the definitional consideration between stepwise vs cumulative dissemination/transmission rate. In revising, the authors have strengthened an already phenomenal manuscript. Given that, I would wholeheartedly endorse the acceptance and publication of this work.

Reviewer #3: My minor concerns mentioned after Revision 1 have been addressed. 

The authors now provide Table 5, which raised numerous concerns (most of these detected by the very thorough anonymous Reviewer #2), has now been derived from an Excel spreadsheet with automated calculations. This makes sure that IR >= CDR >= CTR. The formulas have not been applied to all rows, presumably to avoir '#ERROR' values due to some divisions by zero, and instead value was forced to zero in these cases. I am honestly not sure which is best since 0/0 is also not equal to 0, though readership will eventually be using "0" for any dissemination/transmission rate if infection was completely unsuccessful. So, this is fine by me.

There are rows where formulas do not reference existing cells: 

- Some with hard-coded values that were derived based on extrapoliting percentages in original articles; this is fine by me

- Two have hard-coded mathematical operations, this is slightly worse because one has to make sure the numbers do match those from columns B, C, D and E. This applies to row 20 (I20 and J20)

Since this supplementary file is used as a basis for generating Table 5, it would be good that at least these two hard-coded mathematical operations are corrected to reference the corresponding denominators from columns B and D. The results are, nonetheless, valid.

I am fine with proceeding for publication (or asking the authors for that very minor modification to Table S2 and then only undergo editorial validation).

PLOS authors have the option to publish the peer review history of their article (what does this mean?). If published, this will include your full peer review and any attached files.

Reviewer #1: No

Reviewer #2: No

Reviewer #3: Yes: Thomas Obadia

Figure Files:

Data Requirements:

Reproducibility:

References

---

## [Editor Report · Decision Letter 3]

14 Aug 2023

Dear Dr. Morrison,

We are pleased to inform you that your manuscript 'Secondary vectors of Zika Virus, a systematic review of laboratory vector competence studies' has been provisionally accepted for publication in PLOS Neglected Tropical Diseases.

Best regards,

Andrea Morrison, Ph.D.

Academic Editor

Andrea Marzi

Section Editor

---

## [Editor Report · Acceptance letter]

21 Aug 2023

Dear Dr. Morrison,

We are delighted to inform you that your manuscript, "Secondary vectors of Zika Virus, a systematic review of laboratory vector competence studies," has been formally accepted for publication in PLOS Neglected Tropical Diseases.

Best regards,

Shaden Kamhawi

co-Editor-in-Chief

Paul Brindley

co-Editor-in-Chief
